# Transfer of modified gut viromes improves symptoms associated with metabolic syndrome in obese male mice

Xiaotian Mao[1], Sabina Birgitte Larsen[1], Line Sidsel Fisker Zachariassen[2], Anders Brunse [3], Signe Adamberg [4], Josue Leonardo Castro Mejia[1], Frej Larsen [1], Kaarel Adamberg [4], Dennis Sandris Nielsen [1], Axel Kornerup Hansen [2], Camilla Hartmann Friis Hansen[2] & Torben Sølbeck Rasmussen [1] ✉

Metabolic syndrome encompasses amongst other conditions like obesity and type-2 diabetes and is associated with gut microbiome (GM) dysbiosis. Fecal microbiota transplantation (FMT) has been explored to treat metabolic syndrome by restoring the GM; however, concerns on accidentally transferring pathogenic microbes remain. As a safer alternative, fecal virome transplantation (FVT, sterile-filtrated feces) has the advantage over FMT in that mainly bacteriophages are transferred. FVT from lean male donors have shown promise in alleviating the metabolic effects of high-fat diet in a preclinical mouse study. However, FVT still carries the risk of eukaryotic viral infections. To address this, recently developed methods are applied for removing or inactivating eukaryotic viruses in the viral component of FVT. Modified FVTs are compared with unmodified FVT and saline in a diet-induced obesity model on male C57BL/6 N mice. Contrasted with obese control, mice administered a modified FVT (nearly depleted for eukaryotic viruses) exhibits enhanced blood glucose clearance but not weight loss. The unmodified FVT improves liver pathology and reduces the proportions of immune cells in the adipose tissue with a non-uniform response. GM analysis suggests that bacteriophage-mediated GM modulation influences outcomes. Optimizing these approaches could lead to the development of safe bacteriophage-based therapies targeting metabolic syndrome through GM restoration.

Obesity is a severe metabolic disease that affects millions of individuals worldwide due to high-calorie foods and a sedentary lifestyle[1]. It is also a risk factor for the later development of other metabolic comorbidities, such as type-2 diabetes and metabolic dysfunction-associated fatty liver disease (MAFLD)[2,3]. While many patients with obesity can

achieve significant weight loss, only a few can maintain their long-term body weight. It is increasingly recognized that a dysbiotic gut microbiome (GM) plays a crucial role in the persistence of obesity[2,4]. Therefore, targeting the GM for treating metabolic syndrome and restoring its long-term balance is an attractive strategy. Fecal

[1]Section of Food Microbiology, Gut Health, and Fermentation, Department of Food Science, University of Copenhagen, Frederiksberg, Denmark. [2]Section of Preclinical Disease Biology, Department of Veterinary and Animal Sciences, University of Copenhagen, Frederiksberg, Denmark. [3]Section of Comparative Pediatrics and Nutrition, Department of Veterinary and Animal Sciences, University of Copenhagen, Frederiksberg, Denmark. [4]Department of Chemistry and Biotechnology, Tallinn University of Technology, Tallinn, Estonia. ✉e-mail: torben@food.ku.dk

microbiota transplantation (FMT) has shown high treatment efficacy in targeting recurrent *Clostridioides difficile* infections (rCDI)[5], as well as potential importance in colorectal cancer treatments[6], and the prevention of necrotizing enterocolitis[7,8]. FMT has also been investigated to alleviate symptoms of metabolic syndrome. Although some positive effects have been reported when using FMT to treat patients suffering from obesity and type-2 diabetes[9-11], these effects are usually limited and short-term. Despite the high success rates in treating rCDI[5] and the rigorous donor screening, FMT remains a last resort for patients who do not respond to antibiotics due to safety concerns[12]. The reasoning for this caution was exemplified by the death of a patient in 2019 due to a bacterial infection following FMT[13], as well as subsequent safety alerts from the authorities[14,15]. Consequently, while FMT has the potential to revolutionize treatments for many gut-related diseases, its widespread use seems implausible due to the inherent safety issues of transferring pathogenic microorganisms. The fecal matrix of FMT includes bacteria, archaea, eukaryotes, viruses, and gut metabolites, which may be responsible for both the beneficial and detrimental effects of FMT[16]. Therefore, reducing the complexity of the donor fecal matrix while maintaining its therapeutic efficacy would improve safety in clinical settings.

The gut virome is thought to play a vital role in shaping and maintaining the composition of the GM and host metabolism[17,18]. It mainly consists of bacteriophages (phages), which are viruses that infect bacteria in a host-specific manner[19], while archaeal and eukaryotic viruses constitute the remaining. Screening assays can be used to detect known pathogenic viruses before performing FMT. However, in recent years, it has become evident that the human gastrointestinal tract harbors hundreds of eukaryotic viruses with unknown functions[18,20,21]. Although most of these viruses are likely harmless, their potential role in later disease development should not be overlooked, as seen for the human papillomavirus that can induce cervical cancer years after infection[22].

Independent studies have successfully treated patients suffering from rCDI with sterile filtrated donor feces (containing mainly viruses and limited number of viable bacteria), which were shown to be as effective as FMT[5,23,24]. The successful treatments have been hypothesized to be driven by phage-based restoration of the GM[19,23]. This approach is often referred to as fecal virome transplantation (FVT). In preclinical settings, FVT has also been shown to alleviate symptoms of type-2 diabetes and obesity in male mice[25,26], prevent the onset of necrotizing enterocolitis in preterm piglets[7], restore the GM after antibiotic intervention[27], and improve the proliferation of commensal gut *Akkermansia muciniphila*[28]. These findings highlight the promising application of FVT as a GM restoring treatment targeting various diseases associated with GM dysbiosis. FVT has an advantage over FMT in that it transfers no or very few bacteria and has, compared with FMT, recently been shown to be less invasive for both the gut microbial structure and associated with a lower risk of damaging jejunum in broiler chickens[29]. However, the centrifugation and filtration steps used in preparing FVT cannot separate phages from eukaryotic viruses due to their similar sizes, that together with donor variability, makes widespread use of FVT to treat obesity and type-2 diabetes unlikely. In two recent studies, we aimed to enhance the safety of FVT by developing methodologies that selectively inactivate[30] (not yet peer-reviewed) or remove the eukaryotic viruses from the fecal matrix[31] (not yet peer-reviewed) while preserving an active enteric phage community. To achieve this, we utilized the differences in key characteristics between eukaryotic viruses and phages[30,31]; most eukaryotic viruses are enveloped RNA viruses[32,33] that only infect eukaryotic cells, and most phages are non-enveloped DNA viruses[33,34] infecting only bacteria. Solvent/detergent treatment (approved by the World Health Organization for treating blood plasma[35]) was used to inactivate enveloped viruses[36] (FVT-SDT), a compound (pyronin Y) that specifically binds to RNA[37,38] was applied to inactivate RNA viruses (FVT-PyT),

and an optimized chemostat fermentation of intestinal inoculum aimed at removing the eukaryotic viruses by dilution[31] (FVT-ChP). We have also demonstrated that the modified FVT-SDT and FVT-ChP showed promising results in treating *C. difficile* infections in a mouse model[30], which represents a simple disease etiology mainly caused by the toxin-producing *C. difficile*[39,40].

In this work, the more complex GM associated diet-induced obesity model[41] is included in this study to investigate whether the same modified FVTs[30] could be used to improve phenotypes from these two very different disease etiologies through phage-mediated restoration of a dysbiotic GM. Only male mice are included, since female mice are highly protected against diet-induced obesity[42]. Our main objective was therefore to screen whether the differently modified FVTs[30,31] had the potential to alleviate symptoms on co-morbidities of metabolic syndrome as a safer and/or more reproducible alternative to the unmodified FVT that we previously have used in a similar diet-induced obesity model[25].

## Results

We have previously shown that FVT alleviates symptoms associated with metabolic syndrome, including weight gain, impaired glucose tolerance, and MAFLD associated gene expressions in diet-induced obese male mice[25]. As a follow-up study, we hypothesize that different techniques that either deplete or inactivate the eukaryotic viral component in FVT, can improve safety while maintaining the alleviating effects of FVT on symptoms associated with metabolic syndrome[25]. The treatment efficacy of these different modified FVTs (FVT-ChP, FVT-SDT, and FVT-PyT) was compared with unmodified FVT and saline treatment of obese control mice. Lean control mice were also included to evaluate the validity of the conducted diet-induced obesity mouse model. All the transferred intestinal donor content used in the study originated from the same mixed donor material originating from lean male mice. Thus, all results presented only account for male mice and similar results can therefore not necessarily be expected to be observed for female mice.

### Improved clearance of blood glucose and alleviated symptoms of MAFLD following FVT

Oral glucose tolerance tests (OGTT) were conducted on mice after 13 weeks (Fig. S1A, B and Table S1) and 18 weeks (Fig. 1A, B) on their respective high-fat or low-fat diet to assess blood glucose regulation. At both study week 13 and week 18, mice on a high-fat diet had significantly ($p < 0.05$) elevated fasting blood glucose levels and impaired glucose regulation during the first 15 min after administration compared with the lean control (Fig. 1A and Table S2). Blood glucose levels of mice treated with the chemostat propagated virome (FVT-ChP) decreased sharply from 30 min to 60 min after glucose administration (Fig. 1A and Table S2), resulting in significantly ($p = 0.021$) improved blood sugar regulation of the FVT-ChP treated mice, compared with the obese control. No FVT treatments improved OGTT scores at the study week 13 (Fig. S1A, B and Table S1). A MAFLD activity score was determined by liver histopathology (Fig. 1C and S1C–E). Mice treated with unmodified FVT tended to have a reduced ($p = 0.245$) MALFD activity score compared to obese controls, but with a non-uniform response (Fig. S1C–E). Whereas FVT-ChP, FVT-SDT, and FVT-PyT treated mice scored similar to the obese control (Fig. 1C). Weight gain was measured to assess the impact of different treatments on body weight. The obese control mice showed a significant increase ($p < 0.05$) in body weight, MAFLD activity score, epididymal white adipose tissue (eWAT), and blood glucose levels compared to lean controls at the termination of the study (Fig. 1B–E), confirming the expected progression of the diet-induced obesity model. Body weight and eWAT size were not improved by any of the FVT treatments compared to obese controls at termination (Fig. 1D, E).

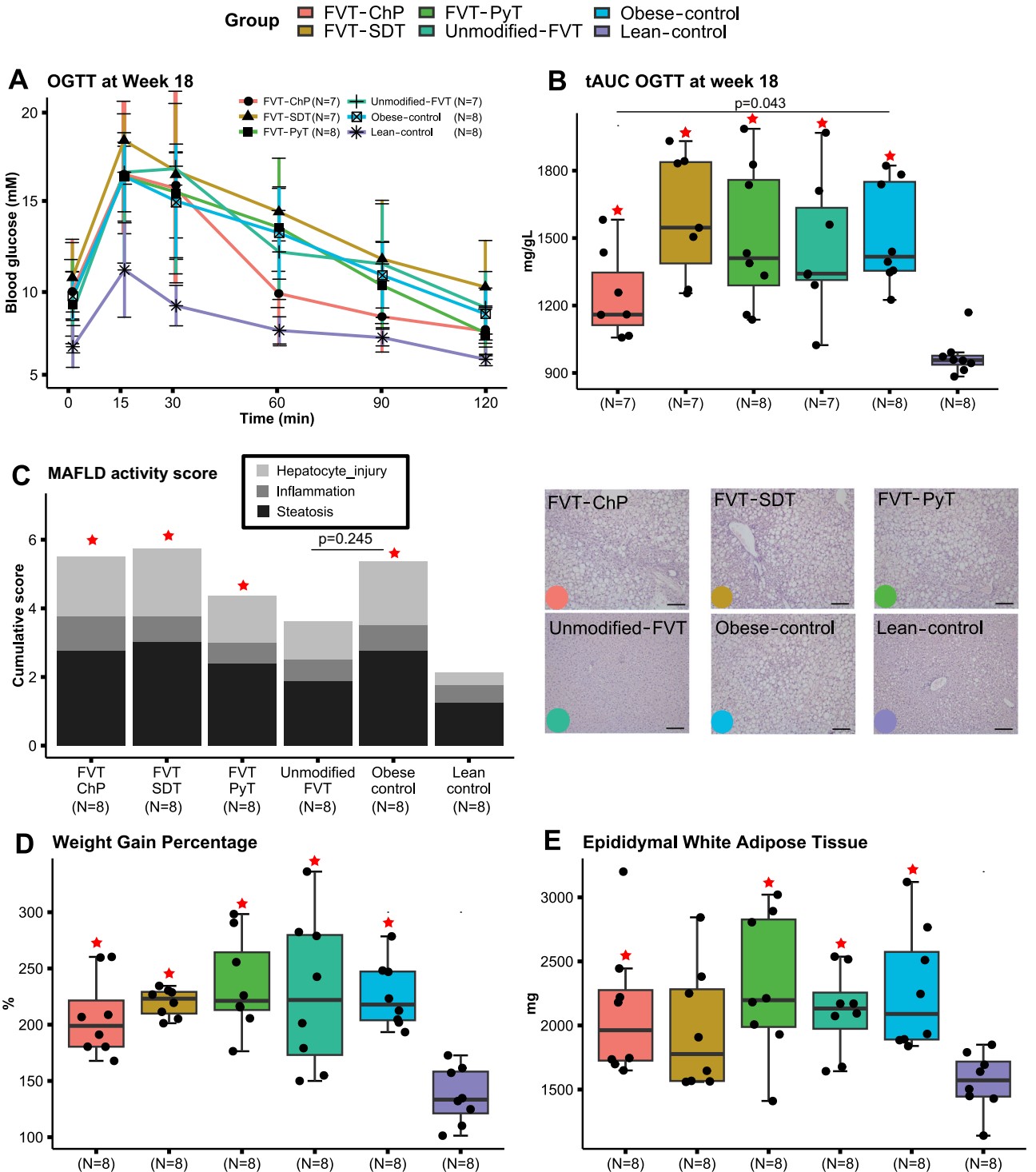

**Fig. 1 | Overview of mouse phenotypic characteristics at termination (23 weeks old) after being treated with modified and unmodified FVT treatments. A** Oral glucose tolerance tests (OGTT) measured at termination of the study (error bars describe mean ± SD). **B** Total area under the curve (tAUC), **C** metabolic dysfunction-associated fatty liver disease (MAFLD) activity score and representative histology images related to liver tissue pathology. **D** Weight gain percentage, and **E** measure of epididymal white tissue. The labels of *, **, *** between the FVT treatments, obese and lean control mice represent adjusted *p* < 0.05, 0.01, 0.001 (two-side Wilcoxon rank-sum test with FDR correction), the unadjusted *p* is presented in figure when it is less than 0.05. Combined boxplots distributions include median, min, max, 25 and 75 percentiles, and outliers (more than 1.5 IQR). Scale in histology images = 300 μm. The red star on top of the boxplot of the treatment represents a *p* < 0.05 between the treatment and the lean control mice (two-side Wilcoxon rank-sum test).

## Unmodified fecal virome affected the immune response in adipose tissue

Flow cytometric analysis was performed to evaluate the proportions of immune cells in eWAT (Fig. 2A–L) at termination (study week 18). Interestingly, mice treated with the unmodified FVT showed a decrease in the proportions of dendritic cells (%CD11c+ of CD45+) (*p* = 0.05), T helper cells (%CD4+ of TCRab+) (*p* = 0.015), activated macrophages (%CD11c+ of F4.80+) (*p* = 0.038), and central memory CD8 + T cells (%CD44 + CD62l+ of CD8+) (*p* = 0.05) compared to the obese control (Fig. 2A, C, E, G). Except for mice treated with

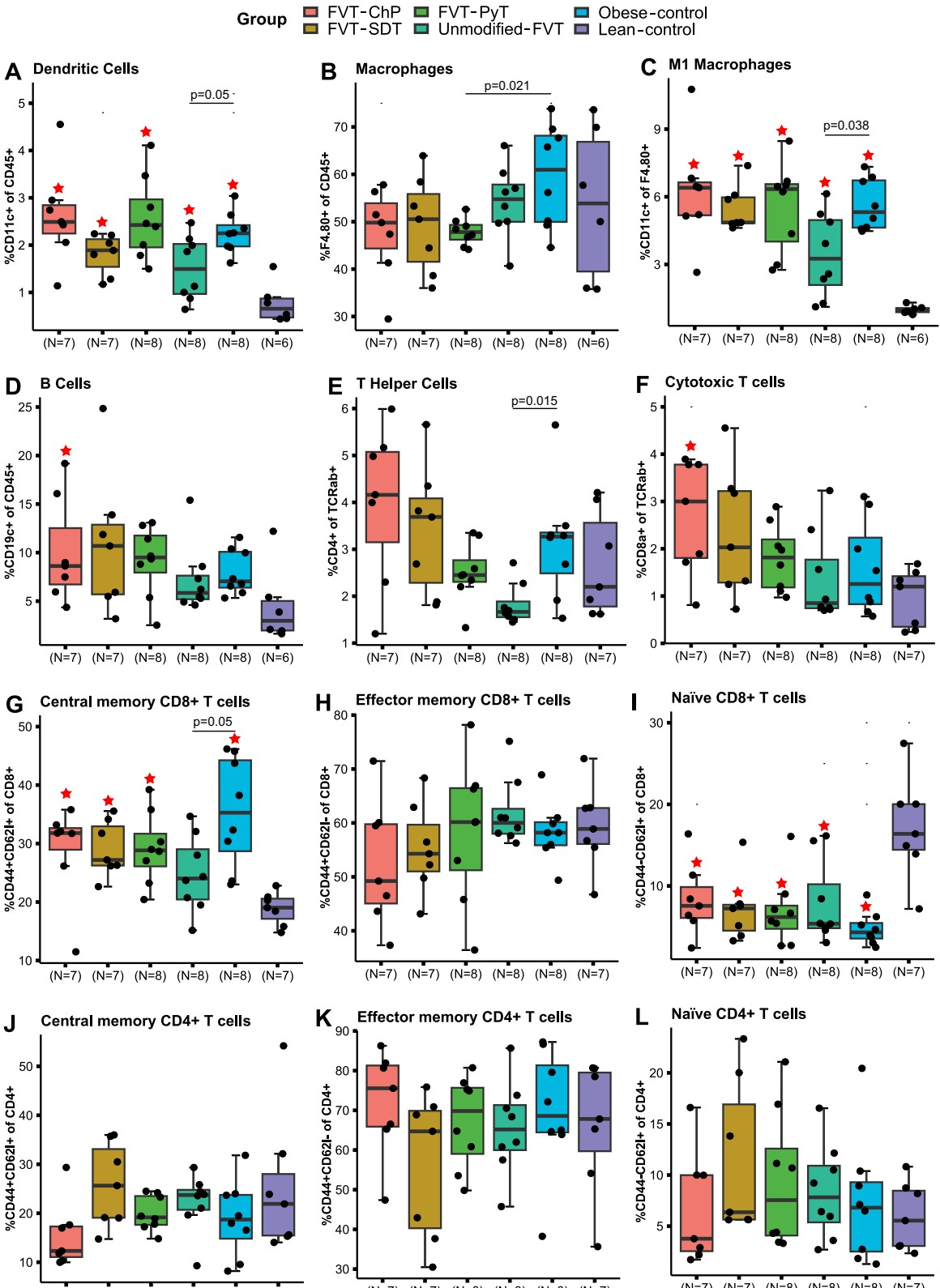

**Fig. 2 | Proportions of immune cells in adipose tissue at study termination (23 weeks old). A–L** are showing the overall flow cytometric profile in the mouse adipose tissue. The labels of *, **, *** between the FVT treatments, obese and lean control mice represent adjusted $p < 0.05$, 0.01, 0.001 (two-side Wilcoxon rank-sum test with FDR correction), the unadjusted $p$ is presented in figure when it is less than 0.05. Combined boxplots distributions include median, min, max, 25 and 75 percentiles, and outliers (more than 1.5 IQR). The red star on top of the boxplot of the treatment represents a $p < 0.05$ between the treatment and the lean control mice (two-side Wilcoxon rank-sum test).

unmodified FVT, mice fed a high-fat diet generally appeared with a significant ($p < 0.05$) increase in the proportions of dendritic cells (% CD11c+ of all CD45+ leukocytes), central memory CD8 + T cells (%CD44 + CD62l+ of CD8+), naïve CD8 + T cells (%CD44-CD62l+ of CD8+), and in the proportion of activated macrophages (%CD11c+ of F4.80+ macrophages), compared with the lean control group (Fig. 2A, G, I, C). The proportions of immune cells in the mesenteric lymph node (MLN) were also measured but showed no clear differences between treatment groups (Fig. S2).

## Inflammatory signal associated with the chemostat-propagated virome

Motivated by the observed effects on OGTT and histological measures by the unmodified FVT and FVT-ChP treated mice compared with the lean and obese control groups, we decided only to examine the cytokine profile of these four groups (representing a total of 32 mice). The serum cytokine profile of FVT-ChP treated mice showed significantly elevated ($p < 0.05$) expression of three pro-inflammatory cytokines, including IL-15, TNF-α, and MIP-2 compared to the obese control (Fig. S3). The overall cytokine profile of serum from mice treated with unmodified FVT was not different from that of obese control mice. Therefore, the cytokine profile does not explain the improved liver pathology (Fig. 1C). It should be noted that interpretation of the cytokine profiling is challenged by the lack of significant differences between the obese and lean control in 9 out of 10 investigated cytokines.

## Shifted gut bacteriome composition was linked to metabolic syndrome improvement

Upon arrival to our housing facility, the bacterial composition of the mice appeared comparable based on bacterial diversity (Shannon diversity index) and bacterial composition (Bray–Curtis dissimilarity) (Fig. 3A, B). The change in diet and environmental conditions in our housing facilities, compared with that of the vendor, clearly affected the bacterial diversity (Fig. 3A) of all groups, including the lean control. However, the bacterial diversity for the lean control was close to normalized at termination compared with the arrival. As expected, the *ad libitum* low-fat diet provided to the lean control mice, significantly ($p < 0.05$) affected the bacterial diversity and composition compared to the *ad libitum* high-fat diet fed groups (Fig. 3A, B). At the termination of the study, the FVT-ChP treated mice had a significantly lower ($p = 0.028$) bacterial diversity compared with obese control mice (Figs. 4A, 3A). Except for FVT-SDT ($p = 0.12$), all the different FVT-treated mice harbored a significantly ($p < 0.05$) altered gut bacterial composition compared to the obese control group at termination (Fig. 3B). The dominant bacterial phyla detected in the mice feces were *Firmicutes*, *Bacteroidetes*, *Proteobacteria*, *Verrucomicrobia*, *Deferribacteres*, and *Actinobacteria* (Fig. S4J). We adopted DESeq2 to identify differentially abundant bacterial taxa between the FVT treatment groups and obese control mice. The FVT-ChP treated mice were observed with a significant ($p < 0.05$) increase in the relative abundance of *Limosilactobacillus reuteri*, and a tendency ($p < 0.1$) of increase in bacterial taxa belonging to lactobacilli, *Allobaculum*, and *Bacteroidales* compared with the obese control mice (Fig. 3D). In contrast, unmodified FVT treated mice had a significant ($p < 0.05$) decrease in the relative abundance of bacteria belonging to the genus *Allobaculum* and a tendency ($p < 0.1$) of increase was observed for *Erysipelotrichaceae* and *Bacteroidales* (Fig. 3D).

## Altered bacterial compositions were associated with improvements of metabolic syndrome

Correlation analyses were performed to elucidate potential links between the bacterial GM component and the alleviating effects associated with the unmodified FVT and FVT-ChP. Using phenotypic characteristics and measured proportions of immune cells in adipose tissue as indicators of metabolic syndrome of the mice, we performed Spearman correlation analysis (FDR corrected $p < 0.05$, |Spearman coefficient | > 0.4) between these indicators and the most abundant bacterial taxa (relative abundance >1%). The improved glucose regulation was positively correlated with the relative abundance of *Allobaculum spp.* (coefficient = −0.47) and negatively correlated with the relative abundance of *Lactococcus spp.* (coefficient = 0.41) (Fig. 3E). The elevated relative abundance of *A. muciniphila (coefficient = −0.43)*, *Bacteroides spp.* (coefficient = −0.51), *Oscillospira spp.* (coefficient = −0.42) were correlated with lower MAFLD activity scores. In contrast, the relative abundance of unclassified lactobacilli was positively correlated (coefficient = 0.52) with the MAFLD activity score (Fig. 3E). When the size of eWAT increase, the relative abundance of taxa *Allobaculum spp.* (coefficient = −0.46) tended to be diminished, while the relative abundance of *Lactococcus spp.* (coefficient = 0.68) decrease (Fig. 3E). The bodyweight gain of the mice was negatively correlated (coefficient = −0.41) with the relative abundance of *A. muciniphila* (Fig. 3E).

The pairwise Spearman correlations were established between the proportions of immune cells from adipose tissue and bacterial taxa. The abundance of unclassified lactobacilli exhibited a positive correlation with the proportions of activated macrophages, central memory CD8 + T cells, dendritic cells, and cytotoxic T cells (respective coefficients = 0.54, 0.52, 0.63, and 0.41) (Fig. 3F). Conversely, the abundance of *Oscillospira spp.* demonstrated an inverse correlation with the proportions of four immune cells (*respective coefficients* = −0.65, −0.59, −0.56, and −0.42) (Fig. 3F). The negative correlation was also found between *Bacteroides spp.* and proportions of activated macrophages (coefficient = −0.44), as well as cytotoxic T cells (coefficient = −0.57) (Fig. 3F).

## Transplantation of modified fecal viromes shifted the GM composition

None of the FVT treatments affected gut viral diversity (Shannon diversity index), and neither any differences between the lean and obese control (Fig. 4A). The viral composition (Bray–Curtis dissimilarity) of mice treated with the different FVTs was different ($p < 0.1$) from the obese control at termination (Fig. 4B). This indicated that introducing a new viral community through FVT leads to changes in the recipient's gut viral composition. The taxonomic profiles of the recipient gut viromes were dominated by unclassified viruses belonging to the family *Microviridae*, order *Petitvirales* and order *Tubulavirales*-associated viruses (Fig. 4C), and the predicted (based on viral contigs) bacterial hosts were dominated by the genera of *Mucispirillum*, *Bacteroides*, and *Prevotella* (Fig. 4D). Differential viral relative abundance was analyzed at the level of viral contigs (vOTUs) (Fig. 4E and S5) to support the observed differences in the viral composition. When comparing both FVT-ChP and unmodified FVT treated mice to the obese control at the study termination, significant ($p < 0.05$) differences were observed in the relative abundance of viruses belonging to the family *Microviridae*, order *Petitvirales*, and class *Caudoviricetes* (Fig. 4E and S5). Pairwise Spearman's correlation analysis was conducted to investigate the influence of viral composition on the gut bacteriome. The relative abundance of different viral contigs of the family *Microviridae* both negatively (coefficient < −0.4) correlated with the genus *Allobaculum* and positively (coefficient >0.4) correlated with lactobacilli, *Bacteroides spp.*, and the family *Clostridiaceae* (Fig. 4F and S6). Furthermore, the viral contigs of order *Petitvirales* showed a strong positive (coefficient >0.4) correlation with the genus of *Lactococcus* (Fig. 4F and S6). Eukaryotic viruses appeared to be nearly depleted in the FVT-ChP in terms of relative abundance (0.1%) compared to the FVT-SDT (3.66%), unmodified FVT (1.07%), and FVT-PyT (7.41%) (h. It should be emphasized that the metavirome sequencing can solely be applied to evaluate the removal of viruses, while it cannot differentiate whether viral particles have

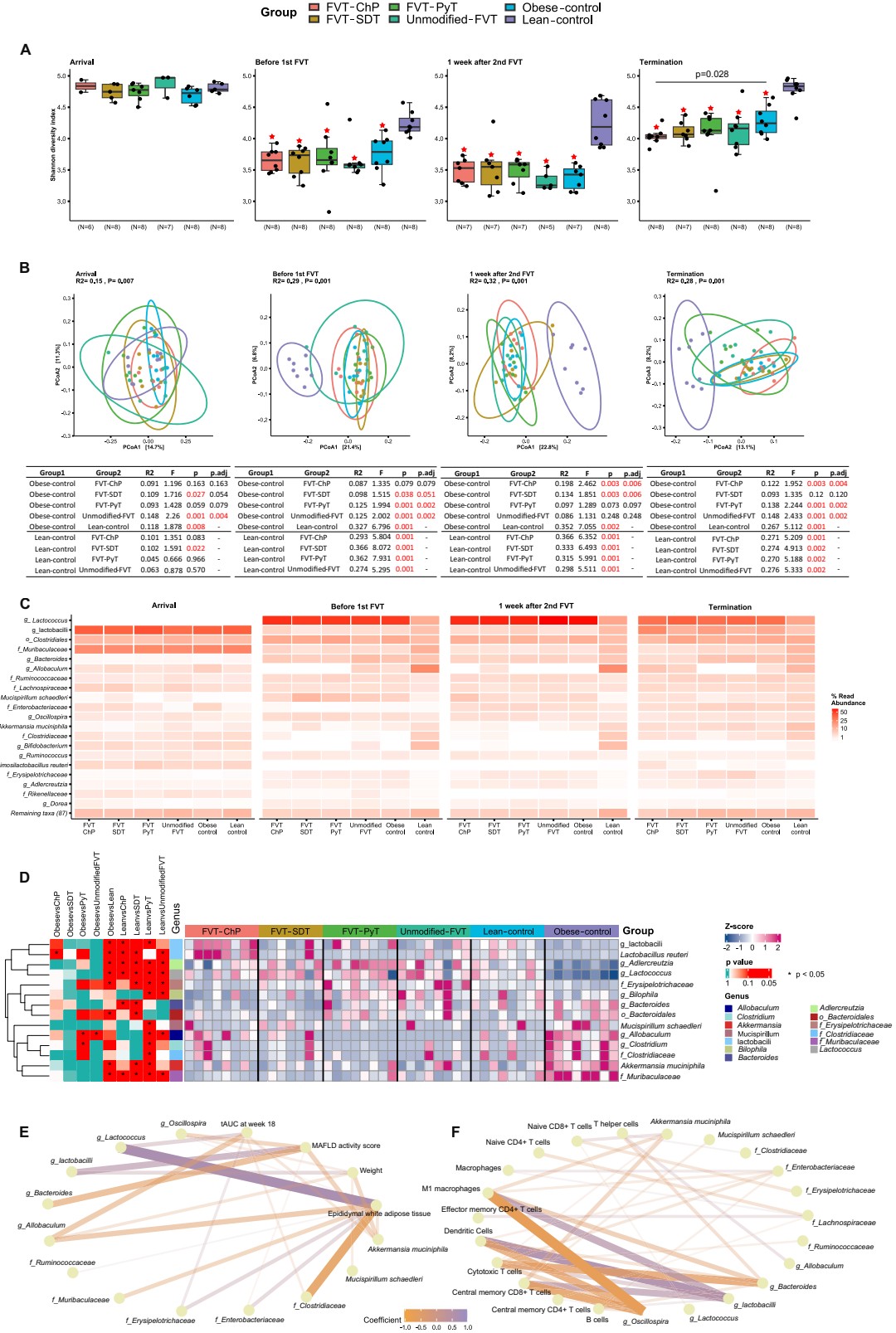

been inactivated or not from e.g., the solvent-detergent or pyronin Y treatment, and the relative abundance does not account for the quantity of viruses.

## Recipient-dependent response to unmodified FVT

In this study, mice treated with unmodified FVT revealed a tendency of a reduction in the MAFLD activity score compared with obese control

mice (Fig. 1C). However, this reduction was not uniformly observed across all mice within the unmodified FVT treated mice (Fig. S1C–E). Instead, it was predominantly driven by a subset of mice exhibiting a more pronounced cage independent response to the unmodified FVT treatment (Fig. S1C, D, E). To understand what caused the different responses to the same treatment, the study subjects were stratified into two distinct groups: those exhibiting a response to the

**Fig. 3 | Bacteriome analysis based on 16 S rRNA gene amplicon sequencing at four time points.** At arrival (before diet intervention (5 weeks old), before 1st FVT (11 weeks old), one week after 2nd FVT (14 weeks old), and study termination (23 weeks old). **A** The bacterial diversity (Shannon diversity index), combined boxplots distributions include median, min, max, 25 and 75 percentiles, and outliers (more than 1.5 IQR), **B** PCoA plot of the bacterial composition (Bray–Curtis dissimilarity), alongside a table showing pairwise PERMANOVA (one-sided) results ($p$ adjusted for FVT treatments and obese control mice comparisons by FDR correction), **C** heatmap representing the relative abundance of dominating bacterial taxa (two-sided Wald test), **D** heatmap highlighting significant ($p < 0.05$) difference in the differentially abundant bacterial taxa, and **E**, **F** Pairwise Spearman's correlations (two-sided) between relative abundance of bacterial taxa (relative abundance >1%) and phenotypic characteristics, and immune cell proportions in adipose tissue, after FDR correction for multiple comparison. The absolute values of coefficients are shown in the figure. The zOTUs were collapsed into the best possible taxonomical resolution. The labels of *, **, *** between the FVT treatments, obese and lean control mice represent adjusted $p < 0.05$, 0.01, 0.001 (two-side Wilcoxon rank-sum test with FDR correction), the unadjusted $p$ is presented in figure when it is less than 0.05. Combined boxplots distributions include median, min, max, 25 and 75 percentiles, and outliers (more than 1.5 IQR). The red star on top of the boxplot of the treatment represents a $p < 0.05$ between the treatment and the lean control mice (two-side Wilcoxon rank-sum test).

unmodified FVT treatment (unmodified-FVT response; characterized by hepatocyte injury score ≤ 1, inflammation score ≤ 1, and steatosis ≤ 2; number of mice = 4), and those showing no response (unmodified-FVT non-response; with hepatocyte injury score > 1, inflammation score > 1, and steatosis > 2; number of mice = 4).

Despite the low sample size, mice within the unmodified-FVT responder group displayed a significant ($p < 0.05$) downregulation in MAFLD severity (Fig. 5A) and a tendency of improved blood glucose regulation relative to the obese controls (Fig. 5D). The unmodified-FVT response group also demonstrated significantly ($p < 0.05$) reduced proportions of dendritic cells (%CD11c+ of CD45+ leukocytes), activated macrophages (%CD11c+ of F4.80+), T helper cells (%CD4+ of TCRab+), and central memory CD8 + T cells (%CD44 + CD62l+ of CD8 + ) in the adipose tissue, compared to the obese control mice (Fig. S7A, C, E, G). No clear differences were observed in the proportions of immune cells in the mesenteric lymph node (Fig. S8) or the blood serum cytokine profile (Fig. S9). Moreover, the bacterial profile of the mice responding to the unmodified FVT treatment was significantly separated from the non-responding ($p = 0.028$) and the obese control mice ($p = 0.005$) (Fig. 5F). This divergence was further underscored by the increased relative abundance of *Bacteroides* and *Dehalobacterium*, alongside a significant ($p < 0.05$) increase in *Oscillospira* and a decrease in lactobacilli in the unmodified-FVT responder group relative to the obese controls (Fig. 5H, K, L, N). These findings imply that responses to the same treatment regimen can vary due to individual differences, however, further studies need to be conducted to validify these observations, since the animal model was neither designed nor hypothesized to include the responder versus non-responder stratification.

## Discussion

To improve the safety of fecal virome transplantation (FVT), recently developed modification methodologies were applied to inactivate or remove the eukaryotic viral component of FVT while maintaining an active enteric phage community[30,31]. We examined the potential of three different modified FVTs to mitigate symptoms of metabolic syndrome compared with unmodified FVT and saline, using a diet-induced obesity male mouse model. Mice treated with a chemostat-propagated virome (FVT-ChP) that was nearly depleted from eukaryotic viruses by dilution, expressed a significantly ($p < 0.05$) improved blood glucose regulation compared with the obese control. Considering the reproducibility potential of chemostat-propagated enteric phageomes[31,43], refinement of this FVT modification strategy may open interesting perspectives for enhancing the reproducibility of FVT based studies and safety by significantly reducing the number of eukaryotic viruses transferred. We have previously shown that unmodified FVT could reduce weight gain and normalize blood glucose regulation in male mice[25], but this was only partly replicated in the present study with a non-uniformly cage-independent phenotypical response of mice treated with unmodified FVT. Mice that responded on the unmodified FVT treatment were characterized by improved oral glucose tolerance test (OGTT) measures, a low liver histopathology, and the bacterial composition was distinctively separated from the

non-responding mice and the obese control. This may be in line with our earlier observations where unmodified FVT showed capable to affect the expression of genes with important metabolic liver functions to be more like the lean control than the obese control mice[25]. The observed discrepancies between the previous[25] and the present study are likely attributable to variations in the gut microbiome (GM) of the donor[44]. Such variations can differ across mouse vendor facilities, posing challenges to reproducibility[45–48]. Additionally, while differences in the GM of the recipient contribute to these discrepancies, their impact might be less pronounced compared to the variability originating from donor materials[49,50].

The cytokine profile of the blood serum was examined since low-grade systemic inflammation is connected to metabolic syndrome[51,52]. Especially elevated levels of the cytokines MIP-2, IL-15, and TNF-α in the FVT-ChP treated mice indicated elevated inflammation. However, the interpretation was challenged by inconsistent differences in the cytokine levels between the lean and obese control groups (Fig. S3). The recruitment and activation of neutrophils via MIP-2 can prompt the release of diverse inflammatory mediators, which can hasten the onset of liver inflammation[53]. TNF-α is produced by adipose tissue and works as a pro-inflammatory cytokine that has been shown to play a role in the development of metabolic dysfunction-associated fatty liver disease (MAFLD)[54,55]. Overexpression of IL-15 in transgenic mice and IL-15 treatment of NOD mice have been reported to improve glucose tolerance[56]. As a double edge sword, the increased levels of pro-inflammatory cytokines (MIP-2, IL-15, and TNF-α) in the blood serum of FVT-ChP treated mice compared to the obese control may partly explain their improved blood glucose regulation, as well as the lack of improvement in their associated weight gain measures and histopathology liver score compared to the obese control mice.

Phage immunogenicity has garnered increased interest due to its potential role in regulating our immune system by sharing similar structures and proteins with eukaryotic viruses[57]. Phages may be both passively and actively distributed throughout various parts of our body[58]. The interactions between phages transferred along with the chemostat-propagated virome and the host immune system could help explain our observations of increased levels of three out of ten cytokines in the FVT-ChP treated mice, compared with the obese control. However, we lack evidence of specific viruses that could have triggered this immune response in the blood serum. Although low-grade systemic inflammation is connected to metabolic syndrom[51,52], it does not exclude the possibility of improved glucose regulation alongside increased levels of some cytokines in blood serum.

The inflammatory state of obesity is associated with an accumulation of macrophages in the adipose tissue[59], where the activated macrophages are recruited by CD8+ T cells[60]. Additionally, elevated levels of dendritic cells can play a role in the immune response to liver injury[61]. In this study, reductions in the proportions of CD8 + T cells, activated macrophages, T helper cells, and dendritic cells were observed in the mice treated with unmodified FVT compared to the obese control, which may be linked to the observed reduction in the liver pathology. Since we do not have the total cell count from adipose tissue and there were no significant differences in cytokine

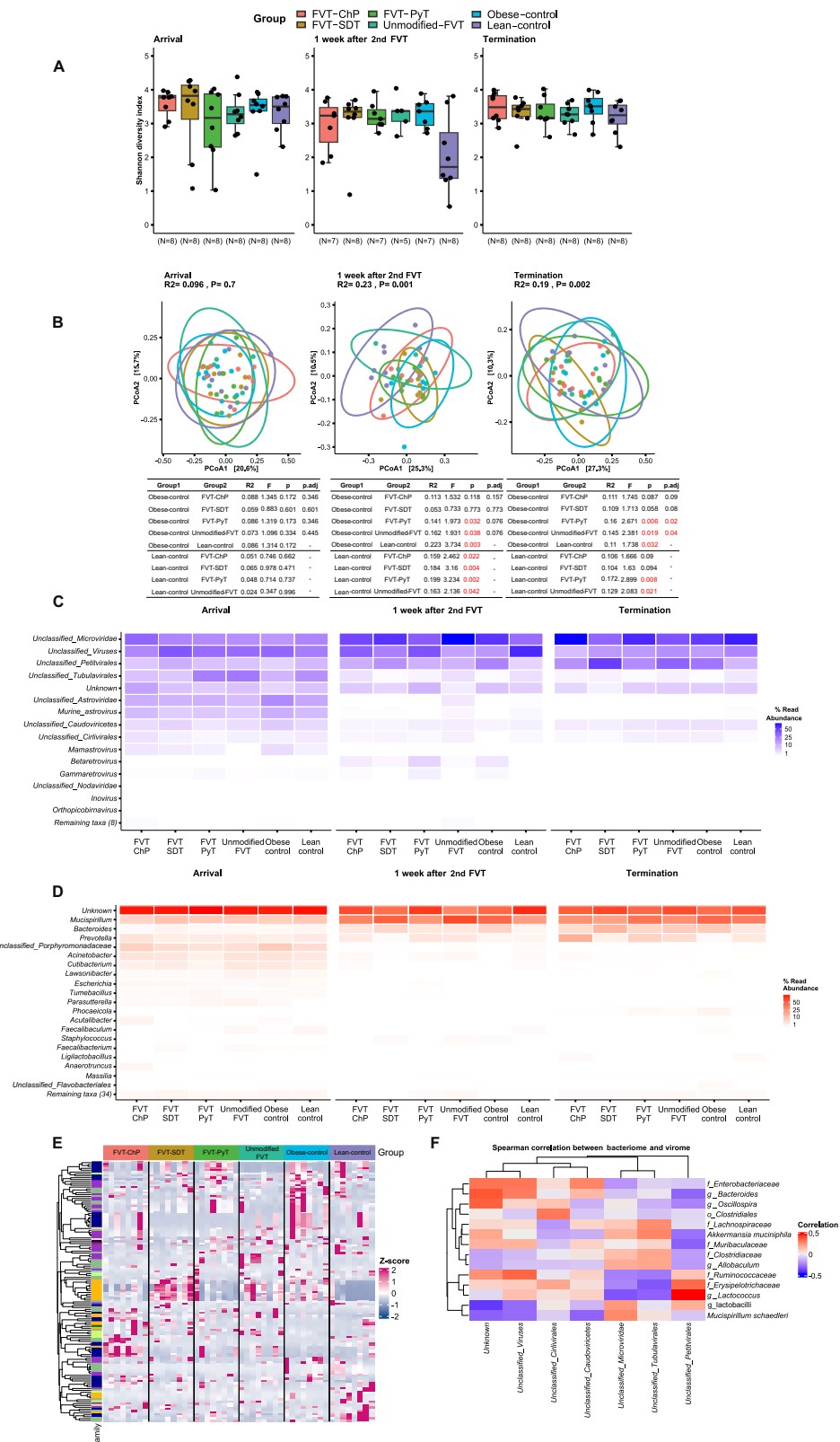

levels (Figs. S3 and S9), it is challenging to draw a general conclusion about the extent to which there was less inflammation in the unmodified FVT group. However, the phenotypical differences present in the liver tissue, adipose tissue, and blood cytokine profile all point in the same direction.

It is commonly accepted that high-fat diet[62] and housing conditions[45,47,63] substantially affects the GM of mice, which would explain the sudden overall decrease in the bacterial diversity in all groups both before and after the study intervention. The relative abundance of *Allobaculum spp.* was increased ($p = 0.083$) in feces samples from FVT-ChP treated mice and significantly ($p < 0.05$) decreased in unmodified FVT treated mice compared with the obese control (Fig. S4A). *Allobaculum* is a Gram-positive bacteria found in both murine and human hosts[64,65]. Prior studies have consistently

**Fig. 4 | Metavirome analysis based on whole-genome sequencing at three time points.** At arrival (before diet intervention (5 weeks old), one week after 2nd FVT (14 weeks old), and study termination (23 weeks old). **A** Boxplots showing the viral diversity (Shannon diversity index), **B** PCoA plot of the viral composition (Bray–Curtis dissimilarity), alongside a table showing one-sided pairwise PERMA-NOVA results (p adjusted for FVT treatments and obese control mice comparisons by FDR correction), **C** heatmap representing the relative abundance of dominating viral taxa, **D** heatmap representing the relative abundance of the predicted bacterial hosts (based on the viral contigs), **E** heatmap highlighting significant (p < 0.05, FDR correction) difference in the differentially abundant viral contigs based on vOTUs level (two-sided Wald test), **F** Pairwise Spearman's correlations (two-sided) between relative abundance of bacterial species (relative abundance >1%) and viral genus (relative abundance >0.1%), with the FDR correction for multi-comparison. The viral contigs were collapsed into the best possible taxonomical resolution. The labels of *, **, *** between the FVT treatments, obese and lean control mice represent adjusted p < 0.05, 0.01, 0.001 (two-side Wilcoxon rank-sum test with FDR correction), the unadjusted p is presented in figure when it is less than 0.05. Combined boxplots distributions include median, min, max, 25 and 75 per-centiles, and outliers (more than 1.5 IQR). The red star on top of the boxplot of the treatment represents a p < 0.05 between the treatment and the lean control mice (two-side Wilcoxon rank-sum test).

identified impaired intestinal integrity, thickness of mucus layer, and immunity (low-grade inflammation) as important factors contributing to the development of metabolic syndromes related diseases[66–69]. Mucin degrading bacteria like *Akkermansia muciniphila*[67,70,71] and *Allobaculum* spp[72–74]. have been reported to play a key role in these processes. *Allobaculum* has been reported as a particularly active glucose utilizer that produces lactate, butyrate[75], and impacts the metabolism of long chain fatty acids[76]. Through the production of short-chain fatty acids like butyrate, *Allobaculum* contributes to the protection of the intestinal barrier[72] stimulates the immune system, and putatively protects against metabolic syndrome[73]. This would be in accordance with supplementation of butyrate limiting hyperglycemia through the regulation of amongst glucagon-like peptide-1 (GLP-1) and insulin in serum[77]. This is further supported by *Allobaculum* being reported to be positive correlated to hypoglycemia and negative cor-related to HOMA-IR levels[78] (not yet peer-reviewed). Taken together, the current understanding of the role of *Allobaculum* on metabolic syndrome supports our observations of improved glucose regulation for the FVT-ChP treated mice that had increased relative abundance of *Allobaculum* compared with obese control (Fig. S4A).

Homologs of four enzymes involved in the butyrate pathway have been discovered in the genome of species belonging to genus of *Oscillospira* suggesting it as a potential contributor to butyrate production[79]. Thus, the significant increase (p < 0.05) in the relative abundance of *Oscillospira* (Fig. 5G) in the mice that responded on the unmodified FVT compared with the obese control, may have played an important role in the butyrate production in these animals. *Oscillospira* could be speculated to have a redundant metabolic role in the butyrate production along with *Allobaculum*, since *Allobaculum* was not detectable in the unmodified FVT treated mice (Fig. 5N). This might have contributed to the tendency of an enhanced blood glucose clearance in the mice that responded on the unmodified FVT (Fig. 5D, E). The level of *Oscillospira* spp. has also been negatively associated with hepatic fat accumulation in pediatric cases of MAFLD[80] and non-alcoholic steatohepatitis (NASH)[81], hence indicating a pro-tective role by *Oscillospira* of fat accumulation in the liver.

The determination of the differential viral abundance showed clear differences when comparing FVT-ChP treatment with unmodified FVT (Fig. S5). Thus, these differences in viral profiles may also have contributed to the development of the two different observed phe-notypes. The mechanisms behind the GM modulating effects of FVT are still poorly understood, but accumulating reports suggest that the phenotypic traits of FVT donors, to some extent, can be transferred to recipients, as the phages might catalyze a modulation of the recipient ecosystem to be similar to their origin. This is exemplified by phage donor profiles being transferred to the gut of *C. difficile* patients after successful FMT/FVT[23,82,83], FVT from lean mice could shift the GM composition in obese mice to resemble that of lean individuals[25], and FVT donor material originating from an ecosystem with a relatively high abundance of *A. muciniphila*, could significantly increase the abundance of the enteric endogenous (native) *A. muciniphila* in mice that received the FVT[28]. This may be driven by cascading events[19], as demonstrated in a gnotobiotic mouse model[84], where phage infections indirectly influence the bacterial balance. Thus, the far more complex viral composition of FVT could similarly alter the bacterial ecosystem and as observed in our previous study, affect the blood metabolome and GM, leading to systemic effects[25]. This concept might seem counterintuitive given the general belief in the strain-specific nature of phages, however, a recent study proposed that phages could interact with distantly related microbial hosts[85]. Phage satellites have, amongst other, been suggested to contribute to broader host ranges[86,87]. Also, the transfer of potentially beneficial metabolic genes from temperate phages to their bacterial hosts[88–91], may enhance host competitiveness and contribute to overall GM changes. These findings align with recent research demonstrating how metabolic functions and bacterial inter-action networks were context-dependent of variables like nutrition, host environment, and bacterial compositions[92], supporting the hypothesis that cascading events initiated by FVT could catalyze GM modulating effects. In addition to the bacteria-phage relations, the impact of the immune system on gut health should not be neglected. Recent evidence suggests that bacteriophages interact with our immune system[57,58] through mechanisms like TLR3 and TLR9[93,94]. Sti-mulation of the immune system may thus be another mechanism behind the efficacy of FVT.

Both the modified FVT-SDT (3.66%) and FVT-PyT (7.41%) exhibited a notably higher relative abundance of eukaryotic viruses when com-pared to the unmodified FVT (1.07%). The inactivation process invol-ving solvent detergent and pyronin Y might also lead to the destruction of a certain fraction of phages. Consequently, this could impact the relative abundance distribution between phages and eukaryotic viruses. While further experimental data is required to substantiate this hypothesis, it aligns with our other study that delves into the efficacy of these methods[30]. In this parallel investigation, pyronin Y reduced the phage activity of the majority of the examined phages, whereas solvent/detergent only affected the activity of one of the tested phages. The chemostat fermentation setup aimed to mimic mouse gut conditions[31] and utilized the same intestinal content as other FVT treatments. However, the chemostat propagation will inevitably alter phage composition and diversity compared to the original inoculum, as also illustrated by the differences in phage composition in the chemostat-propagated virome compared to other treatments (Fig. S10). This is also offering a possible explanation of the decreased Shannon diversity index observed in the FVT-ChP group, compared to the obese control. The chosen sample size (n = 8) group housed in two cages, is previously validated as sufficient for diet-induced obesity[95], thereby accommodating the 3R principle of redu-cing the number of animals in preclinical studies[96]. The combination of group housing and the coprophagic behavior of the mice may cause in cage-associated effects[97], which constitutes a limitation of the study. However, statistical analysis of the phenotypical measures (Table S4) showed no pronounced cage-associated effects concerning the main findings. A recent report suggests to decrease animal density in the cages to increase the statistical power[97], which could address the cage-associated variance issue while maintaining accommodation of the 3Rs[96] in future studies. The viral diversity across all groups showed no cage effects for the Shannon diversity index. Similarly, the viral

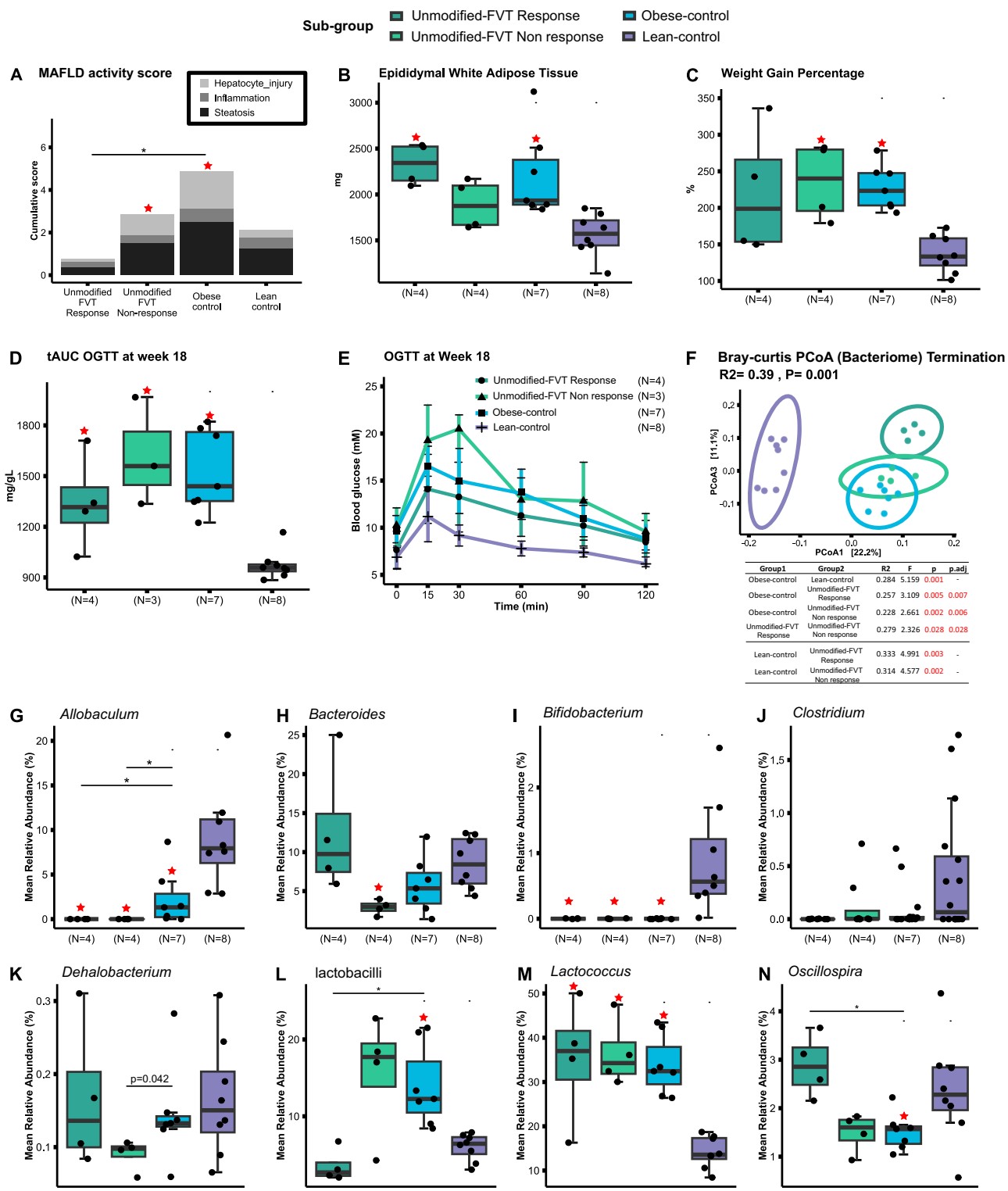

**Fig. 5 | Overview of the unmodified FVT treatment stratification analysis based on response and non-response groups at termination (23 weeks old).**
**A** metabolic dysfunction-associated fatty liver disease (MAFLD) activity score, **B** measure of epididymal white adipose tissue, **C** weight gain percentage, **D** Total area under the curve (tAUC) of the (**E**) oral glucose tolerance tests (OGTT) measured at termination of the study study (error bars describe mean ± SD), (**F**) PCoA plot of the bacterial composition (Bray–Curtis dissimilarity), alongside a table showing one sided pairwise PERMANOVA results (p adjusted for FVT treatments and obese control mice comparisons by FDR correction), and (**G–N**) boxplots showing the relative abundance of selected bacterial taxa. The labels of *, **, *** between the FVT treatments, obese and lean control mice represent adjusted p < 0.05, 0.01, 0.001 (two-side Wilcoxon rank-sum test with FDR correction), the unadjusted p is presented in figure when it is less than 0.05. Combined boxplots distributions include median, min, max, 25 and 75 percentiles, and outliers (more than 1.5 IQR). The red star on top of the boxplot of the treatment represents a p < 0.05 between the treatment and the lean control mice (two-side Wilcoxon rank-sum test).

composition in all FVT groups remained unaffected by cage effects, while the lean and obese control groups displayed cage-associated effects. Regarding bacteria, only the Shannon diversity index of the lean control group showed cage-associated effects. However, bacterial composition in most groups exhibited cage-associated effects, likely influenced by coprophagic behavior, a common issue in mouse studies[97,98]. Thus, it cannot be ruled out that cage-associated differences in the bacterial composition may have influenced the treatment efficacy of the different FVTs.

Most drug-based treatments for chronic diseases are long-term or periodic[99], but this study only treated mice with FVT twice. Compared with the phenotypical changes induced by a long-term high-fat diet, two FVT treatments may be insufficient, raising the question of whether the number and frequency of FVT treatments should be adjusted in future intervention studies.

To summarize, it was possible to partly maintain the alleviating effects of FVT[25] on metabolic syndrome-associated symptoms with a reproducible chemostat-propagated enteric virome that was nearly depleted from eukaryotic viruses. While the unmodified FVT showed the most substantial phenotypic change by reducing diet-induced liver tissue damage and lowered the proportions of immune cells in the adipose tissue. The mice that received the unmodified FVT either responded to the treatment or did not. In contrast, the two other FVT modifications (FVT-SDT and FVT-PyT) did not appear as promising modification strategies of FVT targeting diet-induced obesity, since no clear phenotypical improvements were observed.

The concept of chemostat-propagated phageomes (FVT-ChP) could address the main limitations that are associated with unmodified FVT: procuring and reproducing sufficient enteric phage solutions, in vitro donor-recipient compatibility screening of the individual GM, and minimizing the transfer of eukaryotic viruses from donors to recipients. Considering that the concept of FVT-ChP as a modification of FVT already in this premature stage has demonstrated phenotypical effects in treating two distinct disease etiologies of diet-induced obesity and *C. difficile* infection[30], it urges for refinements as a therapeutic tool targeting other diseases associated with GM dysbiosis.

## Methods

All procedures for handling animals used for collecting donor material and in the diet-induced obesity model were carried out as previously conducted[25] and in accordance with the Directive 2010/63/EU and the Danish Animal Experimentation Act with the license ID: 2012-15-2934-00256 and ID: 2017-15-0201-01262 C1, respectively. The licenses were approved by the Animal Experiments Inspectorate (Ministry of Food, Agriculture, and Fisheries of Denmark). All mice were housed in AAALAC accredited animal facilities at ambient temperature ($22 \pm 2\,°C$), 12 h light/dark cycle (06.00 a.m.–06.00 p.m.), with humidity of $55\% \pm 10\%$, air changes 15–20 times per hour, shielded from ultrasounds >20 kHz. Cages (Scanbur) were enriched with bedding, cardboard housing, tunnel, nesting material, felt pad, and biting stem (Brogaarden).

### The animal origin and preparation of donor viromes

A total of 54 male C57BL/6 N mice were purchased for the purpose of harvesting intestinal content for downstream FVT applications. Upon arrival, the mice were 5 weeks old and obtained from three vendors: 18 C57BL/6NTac mice from Taconic (Denmark), 18 C57BL/6NRj mice from Janvier (France), and 18 C57BL/6NCrl mice from Charles River (Germany). The mice were earmarked upon arrival, randomly (simple randomization) assigned according to vendor to 3 cages with 6 mice each, and housed at the Section of Preclinical Disease Biology, University of Copenhagen, Denmark, following previously described conditions[48]. They were provided with *ad libitum* access to a low-fat diet (LF, Research Diets D12450J) for a period of 13 weeks until they reached 18 weeks of age, which was the planned termination point.

Unfortunately, malocclusions resulted in malnutrition for two C57BL/6NRj mice, and they were euthanized before the intended termination date. All mice were euthanized by cervical dislocation, and samples of intestinal content from the cecum and colon were collected and suspended in 500 μL of autoclaved anoxic PBS buffer (NaCl 137 mM, KCl 2.7 mM, $Na_2HPO_4$ 10 mM, $KH_2PO_4$ 1.8 mM, pH 7.4). Subsequently, all samples were stored at $-80\,°C$. In order to preserve the viability of strict anaerobic bacteria, 6 mice from each vendor (a total of 18 mice) were sacrificed and immediately transferred to an anaerobic chamber (Coy Laboratory) containing an atmosphere of approximately 93% $N_2$, 2% $H_2$, and 5% $CO_2$, maintained at room temperature. The samples collected from these mice within the anaerobic chamber were used for anaerobic chemostat cultivation to produce the chemostat propagated virome (FVT-ChP). The intestinal content from the remaining 34 mice was sampled under aerobic conditions and used to generate the fecal virome for downstream processing of the unmodified FVT, FVT-SDT, and FVT-PyT treatments. A flow diagram illustrating the aforementioned processes is provided (Fig. S11).

### Unmodified fecal virome

As previously conducted[25], thawed intestinal content from the cecum and colon was suspended in 29 mL autoclaved SM buffer (100 mM NaCl, 8 mM $MgSO_4 \cdot 7H_2O$, 50 mM Tris-HCl with pH 7.5), followed by homogenization in BagPage+ 100 mL filter bags (Interscience) with a laboratory blender (Seward) at maximum speed for 120 s. The filtered and homogenized suspension was subsequently centrifuged using a centrifuge 5920 R (Eppendorf) at $4500 \times g$ for 30 min at $4\,°C$. The fecal supernatant was sampled for further processing of FVT solutions, while the pellet was resuspended in PBS buffer for bacterial DNA extraction. The fecal supernatant was filtered through a 0.45 μm Minisart High Flow PES syringe filter (Sartorius) to remove bacteria and other larger particles. Ultrafiltration was performed to both concentrate the fecal filtrate using Centriprep Ultracel YM-30K units (Millipore) and remove metabolites below the size of 30 kDa, that by its design constitute of an inner and outer tube. The permeate in the inner tube was discarded several times during centrifugation at $1500 \times g$ at $20\,°C$ until approximately 0.5 mL was left in the outer tube, which at this point was considered as a fecal virome. The 30 kDa filter from the Centriprep Ultracel YM-30K units was removed with a sterile scalpel and added to the fecal virome to allow viral particles to diffuse overnight at $4\,°C$. The fecal viromes from single animals were mixed according to cages (due to their coprophagic behavior) to ensure the possibility to trace back the origin of specific bacterial or viral taxa. One third of fecal virome of mice from all cages were mixed which represented the unmodified FVT and was immediately stored at $-80\,°C$. The remaining fecal virome aliquots were further processed for inactivation of eukaryotic viruses by either dissolving the lipid membrane of enveloped viruses with solvent/detergent treatment (FVT-SDT) or inhibiting replication of RNA viruses with pyronin Y treatment (FVT-PyT).

### Solvent/detergent-treated fecal virome (FVT-SDT)

The solvent/detergent treatment is commonly used for inactivating enveloped viruses (most eukaryotic viruses are enveloped) in blood plasma, while non-enveloped viruses (most phages are non-enveloped) are not inactivated[36,100]. The fecal viromes were treated according to the recommendations from the WHO for clinical use of solvent/detergent treated plasma; incubation in 1% (w/v) tri(n-butyl) phosphate (TnBP) and 1% (w/v) Triton X-100 at $30\,°C$ for 4 h[35]. However, most of the inactivation usually occur within the first 30-60 minutes of the solvent/detergent treatment[101]. Column chromatography with Amberlite XAD-7 resin (Thermo Scientific) was used for removal of solvent TnBP and detergent Triton X-100 after the solvent/detergent inactivation procedure[102]. The binding capacity per mL Amberlite XAD-7 is 2.8 mg for TnBP (shows no optical absorbance at

 

280 nm) and 9 mg for Triton X-100 (shows optical absorbance at 280 nm)[102]. The applied volume of Amberlite XAD-7 in the column was set to 150% of the theoretical binding capacity to ensure a sufficient removal of TnBP and Triton X-100. The resin column was equilibrated with 0.01 M phosphate buffer ($Na_2HPO_4$ and $NaH_2PO_4$) pH 7.1 containing 0.5 M NaCl until optical density at 280 nm ($OD_{280nm}$) was <0.02. Each solvent/detergent treated fecal virome (mixed by cage) was added separately to the column and the $OD_{280nm}$ was measured to follow the concentration of proteins (expected viral particles and other metabolites >30 kDa) and until $OD_{280nm}$ was <0.02. A 0.01 M phosphate buffer containing 1 M NaCl was used to release potential residual particles from the resin[102]. The removal of the solvent/detergent agents from the fecal virome yielded approximately 100 mL viral flow-through from the column, which was concentrated to 0.5 mL using Centriprep Ultracel YM-30K units. The final product constituted the FVT-SDT treatment and was stored at −80 °C. The TnBP and Triton X-100 were eluted from the resin with 96% analytical grade ethanol (Thermo Scientific) until $OD_{280nm}$ was <0.005, washed with 0.01 M phosphate buffer containing 0.5 M NaCl, cleaned with 0.5 M NaOH, and finally re-equilibrated with 0.01 M phosphate buffer containing 0.5 M NaCl before processing additional samples using the same Amberlite XAD-7 resin beads (maximum 3 times).

### Pyronin Y treated fecal virome (FVT-PyT)

Pyronin Y (Merck) is a strong, red-colored fluorescent compound. It has been reported to exhibit efficient binding to single-stranded and double-stranded RNA (ss/dsRNA), while its binding to single-stranded and double-stranded DNA (ss/dsDNA) is less effective[37,38]. The fecal filtrate was treated with 100 μM pyronin Y and incubated at 40 °C overnight to inactivate viral particles containing RNA genomes[30]. To remove the pyronin Y molecules that were not bound, the pyronin Y treated fecal filtrate suspensions were diluted in 50 mL SM buffer and subsequently concentrated to 0.5 mL using Centriprep Ultracel YM-30K units. This process was repeated three times, resulting in a transparent appearance of the pyronin Y treated fecal filtrate, which constituted the FVT-PyT treatment and was stored at −80 °C.

### Chemostat propagated fecal virome (FVT-ChP)

The preparation of the chemostat propagated virome was performed as described in another study[31]. Briefly, anaerobic-handled mouse cecum content was utilized for chemostat propagation. The culture medium was formulated to resemble the low-fat (LF) diet (Research Diets D12450J) provided to the donor mice as their feed, and growth conditions such as temperature (37 °C) and pH (6.4) were set to simulate the environmental conditions present in the mouse cecum. The end cultures, which underwent fermentation with a slow dilution rate (0.05 volumes per hour), exhibited a microbial composition that resembled the initial microbial composition profile of the donor material[31]. These batches were combined to form the FVT-ChP treatment, and were stored at −80 °C.

### Epifluorescence microscopy

Virus-like particle (VLP) counts were evaluated of all applied fecal viromes (unmodified FVT, FVT-SDT, FVT-ChP, and FVT-PyT, Fig. S10) by epifluorescence microscopy. 10 μL from each FVT sample was suspended in 5 mL of 0.02 μm filtered and autoclaved SM buffer. Sterile filter holders (Swinnex) were mounted with a ceramic ($Al_2O_3$) 0.02 μm Anodisc filter (Whatman) and the sample was subsequently added to a syringe and filtered. This allowed the VLPs to be retained on the surface of the Anodisc filter, which gently was removed from the filter holder to a Kimwipe for drying at room temperature for 20 min. SYBR Gold dye (Thermo Scientific) was suspended in 0.02 μm filtered and autoclaved MilliQ water to gain a 20x working solution of SYBR Gold, of which 100 μL was deposited on a Petri dish for each sample. The dried Anodisc filters were added on top of the SYBR Gold droplet and

incubated in 30 min in a dark room for staining. To remove redundant non-bound SYBR Gold dye, the filters were remounted on the filter holders and washed with 5 mL 0.02 μm filtered and autoclaved SM buffer, followed by drying on Kimwipes for 30 min. Sample mounting solution (25% glycerol and PBS buffer, autoclaved and 0.02 μm filtered) was added to a microscope glass slide (Thermo Scientific) to facilities the Anodisc filter positioning, and additional mounting solution was added on top of the Anodisc filter to facilitate attachment of the glass coverslip. Immersion oil was added to the top of the coverslip. Images were taken using an Axioskop 50 microscope (Zeiss) connected with Micoscope Illumnator VHW 50f-2b (Zeiss) with Neofluar 100x magnification objective (Zeiss) with a 490 nm filter, and CoolSnap camera (RS Photometrics) with the RSImage software. The VLP count was performed using the analyze function of the Fiji ImageJ software v. 1.53 u. Based on these measures, the viral concentration was normalized using SM buffer to $2 \times 10^9$ VLP/mL per treatment.

### Animal model design of diet-induced obesity

Forty-eight male C57BL/6NTac mice at 5 weeks old (Taconic Biosciences, Denmark) were divided randomly (simple randomization) into 6 groups divided in 12 cages with 4 mice per cage: lean control, obese control, unmodified FVT (as the control for modified FVTs), FVT-ChP, FVT-SDT, and FVT-PyT (Fig. 6). Each group was represented by 8 mice where the single animals were interpreted as the experimental unit. Mice were housed with 2 cages per group to account for potential cage effect bias. Decision on sample size ($n = 8$) was based on a previous publication where we validated the suitable sample size in measuring prediabetic symptoms by either HbA1c levels or oral glucose tolerance test (OGTT)[95], which was further supported by a subsequent study[25]. Only male C57BL/6NTac mice were included since female mice are highly protected against diet-induced obesity[42]. For 18 weeks, mice were fed *ad libitum* high-fat diet (Research Diets D12492), except lean controls that were fed a low-fat diet (Research Diets D12450J). After 6 weeks on their respective diets, mice were administered 0.15 mL of the unmodified FVT, FVT-ChP, FVT-SDT, or FVT-PyT by oral gavage twice with one week of interval (study weeks 7 and 8). Obese and lean controls received 0.15 mL SM buffer as sham. The titer of the applied FVT virome was approximately $2 \times 10^9$ VLP/mL (Fig. S10A–D). Treatments, handling, and sampling of cages (cage 1–12) were performed in the order as lean control, obese control, unmodified FVT, FVT-ChP, FVT-SDT, and FVT-PyT (cage 1–6) and repeated with cage 7-12 in same group order. The location of the cages had similar expose to light, noise, distance to ceiling, floor, and entrance. Blood serum, epididymal white adipose tissue (eWAT), the mesenteric lymph node (MLN), liver tissue, intestinal content from the cecum and colon, and tissue from the colon and ileum was sampled at termination at study week 18 (23 weeks old) and stored at −80 °C until downstream analysis. Liver tissue was fixated in 10% neutral-buffered formalin (Sarstedt) for histological analysis and stored at room temperature until further processed.

### Cytokine analysis

Serum samples were diluted 1:2 and analyzed for IFN-γ, GM-CSF, IL-15, IL-6, IL-10, KC/GRO, MIP-2 TNF-α, IL-17A/F, and IL-22 in a customized metabolic group 1 U-PLEX (Meso Scale Discovery, Rockville, ML) according to manufacturer's instructions. Samples were read using the MESO QuickPlex SQ 120 instrument (Meso Scale Discovery) and concentrations were extrapolated from a standard curve using their software Discovery Workbench v.4.0 (Meso Scale Discovery). Measurements out of detection range were assigned the value of lower or upper detection limit.

### Histology

Formalin-fixed liver biopsies were embedded in paraffin, sectioned, and stained with hematoxylin & eosin for histopathological

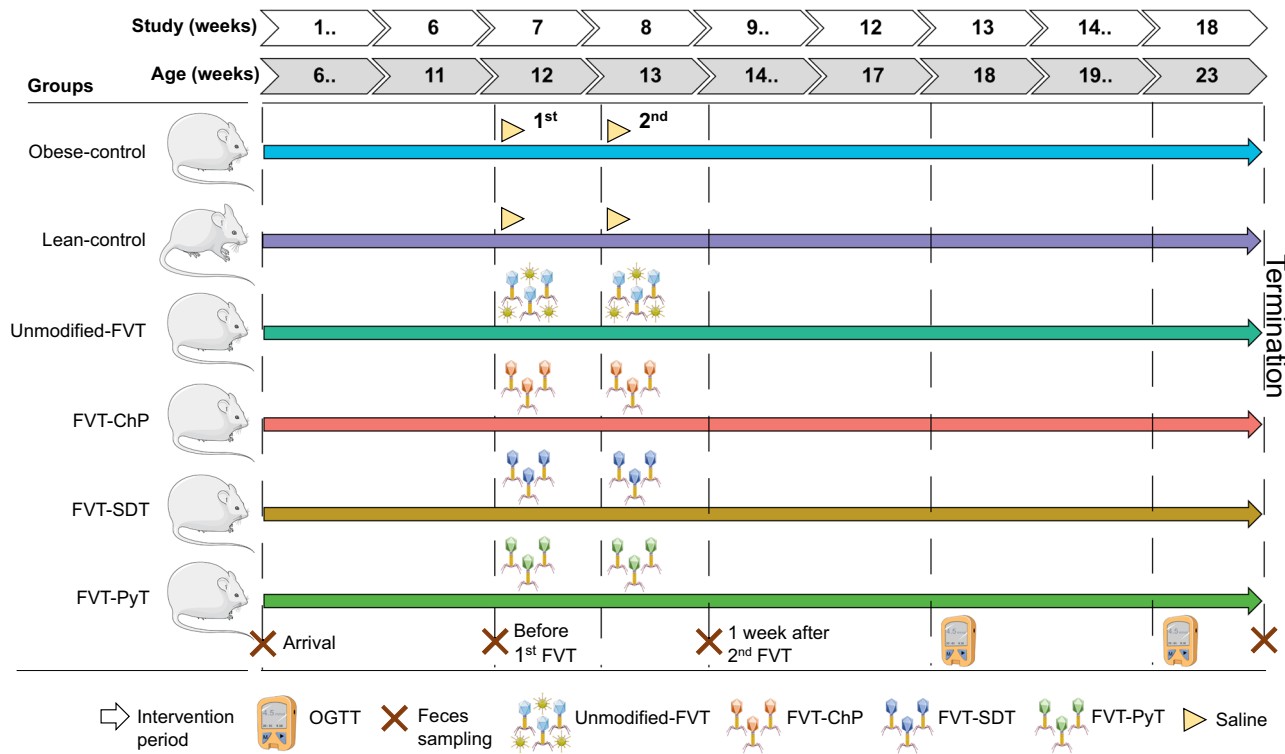

**Fig. 6 | Overview of the animal study design. Forty-eight male C57BL/6NTac mice (5 weeks old) were divided into six groups.** The mice were fed with either a high-fat or low-fat diet. The mice were treated with different FVTs or saline, after 6 and 7 weeks on their respective diets. The FVT treatments represented an unmodified FVT (containing all viruses but limited number of viable bacteria), chemostat propagated fecal virome nearly depleted from eukaryotic viruses by dilution (FVT-ChP), solvent-detergent treated FVT to inactivate enveloped viruses (FVT-SDT), and pyronin Y treated FVT to inactive RNA viruses, and a saline solution for the obese and lean control. Oral glucose tolerance tests (OGTT) were measured twice (18 and 23 weeks old). The crosses mark the timepoint of feces samples that were used for gut microbiome analysis. Selected artwork (bacteriophage, lean mice, obese mice, and glucometer) shown in the figure was provided by Servier Medical Art, licensed under a Creative Commons Attribution 4.0 Unported Licens (https://creativecommons.org/licenses/by/4.0/).

evaluation[103] using the standard tissue processing schedule. We used a cumulative semi-quantitative MAFLD activity score (MAS) comprising the following three histological features: Steatosis (0-3), immune cell margination and infiltration (0-2), and hepatocellular ballooning (0-2) with higher values corresponding to increased dissemination. 8-bit RGB color images were captured with a K5C camera attached to a DM2500 microscope system using a HC FL PLAN 10x objective with numeric aperture 0.25 and acquired with LAS EZ software (all Leica Microsystems). No modifications were made to the captured 8-bit RGB color images, and no further processing was made to the image files post capture.

## Flow cytometric analysis

Directly after euthanizing the mice, the MLN and eWAT were placed in ice cold PBS buffer. Single cell suspensions were prepared by disrupting the lymph node between two microscope glasses and passing it through a 70 μm nylon mesh. eWAT cells were prepared by mincing with scalpels and then digested in a collagenase solution for 20 min at 37 °C with horizontal shaking in 4 mg/ml collagenase II (Sigma-Aldrich) in 5% FBS in Hank's Balanced Salt Solution (HBSS) medium (Applichem). The suspension was filtered through 70 μm filters and washed through centrifugation (8 min at $800 \times g$) in additional HBSS medium[104]. After washing in PBS/HBSS and resuspension, $1 \times 10^6$ cells were surface stained for 30 min with anti-mouse antibodies for PerCP-Cy5.5 conjugated CD11c, PE-conjugated CD19, APC-conjugated F4.80, and FITC-conjugated CD45 for the detection of the different antigen-presenting cells. All antibodies were purchased from eBiosciences: APC F4.80 clone: BM8 (cat#: 17-4801-02, lot#:2123749), FITC-CD62L clone: MEL-14 (cat#: 11-0621-82, lot#: 2608902), PE-CD44 BD clone:

IM7 (cat#:553134, lot#: 3267602), APC-CD8a clone: 53-6.7 (cat#: 17-0081-81, lot#2087760), PerCP-Cy5.5-CD4 clone: RM4-5 (cat#: 45-0042-82, lot#: 2115770), PE-CD19 clone: 6D5 (cat#:115507, lot#:B120254), PerCP-Cy5.5-CD11c clone: N418 (cat# 45-0114-82), FITC-CD45 clone: 30-F11 (cat#: 11-0451-82), FITC-CD8a clone: 53-6.7 (cat#: 11-0081-82), PE-CD4 clone: GK 1.5 (cat#: 12-0041-82), APC-TCRb BD clone: H57-597 (cat#: 553174). For the detection of T cell subsets, $1 \times 10^6$ cells were initially surface stained for 30 min with FITC-conjugated CD8α, PE-conjugated CD4, and APC-conjugated TCRαβ (ebiosciences). Specifically for the differentiation of memory versus naïve T cells, the cells were stained with FITC-conjugated CD62l, PE-conjugated CD44, PerCP-Cy5.5-conjugated CD4, and APC-conjugated CD8α (ebiosciences). All antibodies were diluted 1:100 in Flow Cytometry Staining Buffer (eBiosciences). Analysis was performed using an Accuri C6 flow cytometer (Accuri Cytometers, now BD Biosciences) and accompanying software BD Accuri C6 software version 264.21.

## Pre-processing of fecal samples for separation of viruses and bacteria

The gut bacterial and viral composition of the mice were investigated at respectively four and three time points: upon arrival at our housing facilities (5 weeks old), before 1st FVT (11 weeks old, but only the bacteriome), one week after the 2nd FVT (13 weeks old), and at termination (23 weeks old), which represented in total 192 fecal samples. Separation of viruses and bacteria from fecal samples generated a fecal bacterial pellet and fecal supernatant by centrifugation and 0.45 μm filtering as described in the previous section of "Unmodified fecal viromes", except the volume of fecal homogenate was adjusted to 5 mL using SM buffer.

## Bacterial DNA extraction, sequencing, and pre-processing of raw data

The DNeasy PowerSoil Pro Kit (Qiagen) was used to extract bacterial DNA from the fecal bacterial pellet by following the instructions of the manufacturer and was performed by author XM who at this stage was blinded. The final purified DNA was stored at −80 °C, and the DNA concentration was determined using Qubit HS Assay Kit (Invitrogen) on the Qubit 4 Fluorometric Quantification device (Invitrogen). The primers (NXt_388_F: 5′-TCGTCGGCAG CGTCAGATGT GTATAAGAGA CAGACWCCTA CGGGWGGCAG CAG-3′ and NXt_518_R: 5′-GTCTCGTGGGC TCGGAGATGTG TATAAGAGAC AGATTACCGC GGCTGCTGG-3, Integrated DNA Technologies; Leuven, Belgium) compatible with Nextera Index Kit is used to amplify the V3 region of the 16 S rRNA gene. The first PCR reaction: mixing 12 µL AccuPrime™ SuperMix II (Life Technologies), 0.5 µL of primer mix (10 µM), 5 µL of genomic DNA (~20 ng/µL), 2 µL of either spermine, DSS, or both (in respective concentrations) to a total volume of 20 µL. Cycling was run on SureCycler 8800 with conditions as follow: 95 °C for 2 min; 33 cycles of 95 °C for 15 s, 55 °C for 15 s and 68 °C for 30 s; followed by final step at 68 °C for 5 min. 12 µL Phusion High-Fidelity PCR Master Mix (Thermo Fisher Scientific), 2 µL corresponding P5 and P7 primer (Nextera Index Kit), 2 µL PCR product and nuclease-free water for a total volume of 25 µL were mixed. The 2nd cycling conditions were 98 °C for 1 min; 12 cycles of 98 °C for 10 s, 55 °C for 20 s and 72 °C for 20 s; and 72 °C for 5 min. AMPure XP beads (Beckman Coulter Genomic) were used to purify the PCR products. AMPure XP beads were mixed with the PCR product and incubated at room temperature for 5 minutes and mounted on a magnetic rack for 2 min. The supernatant was discarded, and the beads were washed with 150 µL of 80% ethanol twice without disturbing the beads. The samples were removed from the magnetic rack and the beads were mixed with 25 µL of PCR grade water and incubated at room temperature for 2 min. The PCR tube was then mounted to the magnetic rack again for 2 minutes before the sampling of clean DNA products. The bacterial community composition was determined by Illumina NextSeq based high-throughput sequencing of the 16 S rRNA gene V3 region. Quality control of reads, dereplicating, purging from chimeric reads and constructing zOTU was conducted with the UNOISE pipeline[105] (not yet peer-reviewed) and taxonomically assigned with Sintax[106] (not yet peer-reviewed). Taxonomical assignments were obtained using the EZtaxon for 16 S rRNA gene database[107]. Code describing this pipeline can be accessed at github.com/jcame/Fastq_2_zOTUtable. The average sequencing depth after quality control (Accession: PRJEB58786, available at ENA) for the fecal 16 S rRNA gene amplicons was 67,454 reads (min. 12,790 reads and max. 295,746 reads).

## Viral RNA/DNA extraction, sequencing and pre-processing of raw data

The sterile filtered fecal supernatant was concentrated using centrifugal filters Centrisart with a filter cut-off at 100 kDA (Sartorius) by centrifugation at 1500 × $g$ at 4 °C (dx.doi.org/10.17504/protocols.io.b2qaqdse). Fecal supernatant (140 µL) was treated with 5 units of Pierce Universal Nuclease (ThermoFisher Scientific) for 10 min at room temperature prior to viral DNA extraction to remove free DNA/RNA molecules. The viral DNA/RNA was extracted from the fecal supernatants with the Viral RNA mini kit (Qiagen), and was performed by author XM who at this stage was blinded. Reverse transcription was performed with SuperScript VILO Master mix by following the instructions of the manufacturer and subsequently cleaned with DNeasy blood and tissue kit (Qiagen) by only following step 3-8 in the manual from the manufacturer. In brief, the DNA/cDNA samples were mixed with ethanol, bound to the silica filter, washed two times, and eluted with 40 µL elution buffer. Multiple displacement amplification (MDA, to include ssDNA viruses) was performed using the GenomiPhi V3 DNA amplification kit (Cytiva) by following the instructions of the

manufacturer, except for the genome amplification time was decreased from 90 min to 30 min[21]. Sequencing library preparation was conducted using the Nextera XT kit (Illumina) with a modified protocol compared with the manufacturer. 5 µL Tagment DNA Buffer, 2.5 µL genomic DNA (in total 0.5 ng DNA), and 2.5 µL Amplicon Tagment Mix are mixed, incubated at 55 °C for 5 min, then held on 10 °C. 2.5 µL Neutralize Tagment Buffer was added into the mix and incubated at room temperature for 5 min. 7.5 µL Nextera PCR Mix and 2.5 µL of different Nextera Index primers i5 and i7 of each sample were added to the mix, following by PCR (72 °C for 3 min, 95 °C for 30 s, 16 cycles of 95 °C for 30 s, 55 °C for 30 s, and 72 °C for 30 s, followed by final step at 72 °C for 5 min) on SureCycler 8800. The PCR products were then purified using AMPure XP beads (Beckman Coulter Genomic) as described in the previous section of "Bacterial DNA extraction, sequencing, and pre-processing of raw data". The pooled purified PCR products were sent to the company Novogene for sequencing using the NovaSeq platform (Illumina). The average sequencing depth of raw reads (Accession: PRJEB58786, available at ENA) for the fecal viral metagenome was 13,145,283 reads (min. 693,882 reads and max. 142,821,858 reads). Using Trimmomatic v0.35, raw reads were trimmed for adapters and low-quality sequences (<95 % quality, <50nt) were removed. High-quality reads were dereplicated and checked for the presence of PhiX control using BBMap (bbduk.sh) (https://www.osti.gov/servlets/purl/1241166). Virus-like particle-derived DNA sequences were subjected to within-sample de-novo assembly-only using Spades v3.13.1 and the contigs with a minimum length of 2,200nt, were retained. Contigs from all samples were pooled and dereplicated by chimera-free species-level clustering at ~95% identity using the script described in ref. 21, and available at https://github.com/shiraz-shah/VFCs. Contigs were classified as viral by VirSorter2[108] (full categories | dsDNAphage, ssDNA, RNA, *Lavidaviridae*, NCLDV | viral quality = 1), VIBRANT[109] (High-quality | Medium-quality | Complete), CheckV[110] (High-quality | Medium-quality | Complete), and VirBot[111]. Any contigs not classified as viral by any of the four software's were discarded. The taxonomical categories of other/remaining taxa, unclassified virus, and unknown that are used in the different figures are different entities. Other/remaining taxa encompasses all remaining low abundance taxa not depicted in the plot. Unknown refers to contigs that may be viruses but lack specific data records confirming their viral origin, and unclassified virus represents viruses that have been identified as having viral origin but could not be further classified. Taxonomy was inferred by blasting viral ORFs against a database of viral proteins created from the following: VOGDB v217 (vogdb.org), NCBI (downloaded 14/10/2023), COPSAC[21] and an RNA phage database[112], selecting the best hits with a minimum e-value of $10e^{-6}$. Phage-host predictions were done with IPhoP[113], which utilizes a combination of other host predictors. Following assembly, quality control, and annotations, reads from all samples were mapped against the viral (high-quality) contigs (vOTUs) using bowtie2[114] and a contingency table of contig length and sequencing depth normalized reads, here defined as vOTU table (viral contigs). Code describing this pipeline can be accessed in https://github.com/frejlarsen/vapline3. Mock phage communities (phage C2, T4, phiX174, MS2, and Phi6, Table S3) were used as positive controls (normalized to ~10⁶ PFU/mL for each phage) for virome sequencing to validate the sequencing protocol's ability to include the different genome types of ssDNA, dsDNA, ssRNA, and dsRNA.

## Statistics and reproducibility

Initially, the dataset was purged for zOTU's/viral contigs, which were detected in less than 5% of the samples, but the resulting dataset still maintained 99.8% of the total reads. R version 4.3.0 was used for subsequent analysis and presentation of data. A minimum threshold of sequencing reads for the bacteriome and virome analysis was set to 2000 reads and 15,000 reads, respectively. The main packages used were phyloseq[115], vegan[116], DESeq2[117], ampvis2[118](not yet peer-

reviewed), ggpubr, psych, igraph, ggraph, pheatmap, Complex-Heatmap, and ggplot2. The contamination of viral contig was removed by read count detected in negative controls through R package microDecon[119] (runs = 1, regressions = 1), and 41.5% of entries were removed. Cumulative sum scaling (CSS) normalization was performed using the R software using the metagenomeSeq package. α-diversity analysis was based on raw read counts and statistics were based on ANOVA. β-diversity was represented by Bray−Curtis dissimilarity and statistics were based on one-sided pairwise PERMANOVA corrected with FDR (false discovery rate). DESeq2 was used to identify differential microorganisms on the summarized bacterial species level and viral contigs (vOTUs) level based on two-sided Wald test. The correlation network of bacterial association with phenotypic and immunologic variables, and the correlation heatmap between bacterial zOTUs and viral contigs (vOTUs) were calculated using pairwise Spearman's correlations (two-sided) and corrected with FDR. The non-parametric two-side Wilcoxon rank-sum tests were adopted for analysis of the phenotypic variables, cytokine levels, immune cell levels, Shannon diversity index (α-diversity), PERMANOVA of β-diversity, and abundance of single bacterial genus, the comparison was conducted between FVT treatments and obese control mice, with FDR correction adopted, the comparison between obese and lean control mice was also conducted. The code used for achieving all statistical analysis and figure output is available in the Zenodo repository https://doi.org/10.5281/zenodo.11262999.

In order to comply with the 3R principle of reducing experimental animals[96], and based on previous experiments proving sufficient for diet-induced obesity[95], the sample size was set to $n = 8$, and each group of treatments was evenly place in two cages. The mice were subjected to an OGTT at week 13 and 18 of the study (18 and 23 weeks old) by blinded personnel, and food intake and mouse weight were monitored frequently to evaluate sudden diabetes associated weight loss and behavioral changes (only the ear-tagged mouse ID was available). Author TSR was aware at all stages of the group allocation of the cages, but not the ear-tagged mouse ID, while the animal caretakers were blinded at all stages; thus, TSR handed the FVT treatments or saline to the animal caretakers. OGTT outliers were removed if 2 of 3 parameters were applicable: 1) tAUC > 2000 mg/dL, fasting glucose >11.5 mM, or 3) loss of 5–10% in body weight. The cytokine analysis, histological assessment, and the flow cytometric measures were performed by a blinded investigator. The bacterial/viral DNA extraction was performed by author XM, who was blinded at this stage. No mouse was excluded when evaluating body weight, eWAT size, and liver histopathology. Based on exclusion criteria described in the methods, two mice (FVT-ChP: 1, FVT-SDT: 1) were excluded from the week 13 OGTT measure, and three mice (FVT-ChP: 1, FVT-SDT: 1, unmodified FVT: 1) from the week 18 OGTT measure. flow cytometric analysis failure resulted in the exclusion of three mice (FVT-ChP: 1, FVT-SDT: 1, lean control: 1) from the flow cytometric analysis of adipose tissue and six mice from MLN tissue (lean control: 2, obese control: 1 FVT-PyT: 1, unmodified FVT: 2). Fecal samples of mice were excluded from bacteriome (11 samples) and virome (6 samples) analysis due to lack of sample material or low sequencing reads (Fig. S12).

### Reporting summary
Further information on research design is available in the Nature Portfolio Reporting Summary linked to this article.

## Data availability
All source data is provided in the Supplementary Materials, at https://doi.org/10.5281/zenodo.11262999, and all 529 sequencing datasets are available in the ENA database under accession number PRJEB58786. Source data are provided with this paper.

## Code availability
The codes used for our analyses are available at https://doi.org/10.5281/zenodo.11262999, https://github.com/frejlarsen/vapline3, and https://github.com/jcame/Fastq_2_zOTUtable.

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

## Acknowledgements

This work was supported by the Lundbeck Foundation under Grant R324-2019-1880 and the Novo Nordisk Foundation under Grant NNF-20OC0063874 both received by DSN. We thank the animal care-takers Helene Farlov and Mette Nelander at Section of Preclinical Disease Biology (University of Copenhagen, Denmark) for taking care of the animals during the study and assisting with the animal handling. A thanks to PhD fellow Kaare Dyekær Tranæs (Copenhagen Prospective Studies on Asthma in Childhood, Copenhagen University Hospital) for contributing with internal review and proof-reading of the manuscript. Selected artwork (bacteriophage, lean mice, obese mice, and glucometer) shown in Fig. 6 were used from figures provided by Servier Medical Art (Servier; https://smart.servier.com/), licensed under a Creative Commons Attribution 4.0 Unported License.

## Author contributions

T.S.R. and D.S.N. conceived the research idea and designed the study. T.S.R., S.B.L., L.S.F.Z., C.H.F.H., and A.K.H. conducted, monitored, and supervised the animal experiments. T.S.R., S.B.L., D.S.N., S.A., and K.A. prepared and designed the methodology of the chemostat-propagated virome (FVT-ChP). T.S.R. and S.B.L. prepared and designed the methodology of the unmodified FVT, solvent-detergent (FVT-SDT) and pyronin Y treated FVT (FVT-PyT) and evaluated viral-like-particle count with fluorescence microscopy. X.M. performed nucleotide extractions and library preparation for sequencing. L.S.F.Z. and C.H.F.H. conducted immune cell count and cytokine profile analysis. A.B. prepared and scored the liver histology. T.S.R., S.B.L., X.M., F.L., J.L.C.M., and D.S.N. contributed to the bioinformatic analysis and interpretation. T.S.R., S.B.L., X.M., C.H.F.H., L.S.F.Z., D.S.N., A.K.H., and A.B. analyzed and interpreted the phenotypical measures of the mice. T.S.R. and D.S.N. supervised the study. D.S.N. was responsible for the funding. X.M. and T.S.R. wrote the first draft of the manuscript. All authors critically revised and approved the final version of the manuscript.

## Competing interests

The authors declare no competing interests.

## Ethics declarations

Animal welfare. All procedures involving handling of animals included in the diet-induced obesity model (license ID: 2017-15-0201-01262 C1) and donor animals (license ID: 2012-15-2934-00256) were approved by the Animal Experiments Inspectorate (Ministry of Food, Agriculture, and Fisheries of Denmark) and conducted in accordance with Directive 2010/63/EU and the Danish Animal Experimentation Act. Inclusion. All collaborators in this study have met the authorship criteria mandated by Nature Portfolio journals and have been included as authors because their participation was crucial for designing and conducting the study. Roles and responsibilities were agreed upon among collaborators before the research commenced. This work encompasses findings that are locally and internationally relevant. The research was not subject to severe restrictions or prohibitions in the researchers' setting.
