## [Peer Review File · Nature Communications]

Transfer of modified gut viromes improves symptoms associated with metabolic syndrome in obese male miceREVIEWER COMMENTS

Reviewer #1 (Remarks to the Author):

The study conducted by Mao et al focused on an interesting topic on the effect of viral/phage transfer on non-alcoholic liver disease. The effects of FVT (fecal viral transfer) on metabolic disorders (obesity/diabetes) have been demonstrated by many reports. What positions this study from other studies is that they focused on phage transplants while eliminating the eukaryotic viruses from the donor fecal virome. However, the gut eukaryotic virome consists of both RNA viruses and DNA viruses. While the authors tried to remove the eukaryotic RNA viruses, the eukaryotic DNA viruses were not removed within the viral transplant; it makes the claimed "eukaryotic-virus-free FVT" not valid. In addition, the results did not differ significantly between different "eukaryotic-virus-free" FVT groups and unmodified FVT, in obesity phenotype, immune profile, bacterial profile, and viral profile. Moreover, there is a lack of mechanistic evidence/insights for furthering our understanding in FVT. In addition, I have several other concerns:

1. How many mice were used in each group and what statistical test should be stated.
2. Line 136-137, the authors reasoned that FVT-ChP induced undesirable increase in systemic inflammation. However, it did not investigate why this happened? Is it because of specific viral transfer into the blood? In contrast, unmodified FVT might be result in undesired eukaryotic viral transfer. In addition, how to reconcile the increased systemic inflammation and the relatively better OGTT test phenotype?
3. The bacteria analysis is lack of depth or resolution. ASV, OTU, or metagenomic analysis might be more conducive. So is the virome analysis.

Reviewer #2 (Remarks to the Author):

Reviewer #3 (Remarks to the Author):

Reviewer #4 (Remarks to the Author):

In this manuscript, Mao et al. investigate the effect of fecal virome transplantation (FVT) in a mouse model of diet-induced obesity. They argue that unmodified fecal filtrates may contain harmful levels of eukaryotic viruses, and thus prepared 3 modified FVT treatments for testing: FVT-ChP (chemostat-propagated virome to dilute out eukaryotic viruses), FVT-SDT (solvent/detergent treated to inactivate enveloped viruses), and FVT-PyT (pyronin Y treated to inactivate RNA viruses).

They reported that:

- None of the FVT treatments affected body weight and eWAT size with respect to obese controls (Fig. 1A-B).
- Interestingly, FVT-ChP (but none of the other treatments) led to a significant decline in blood glucose levels 60 (but not 30) minutes after glucose administration, as compared to obese controls (Fig. 1D).
- Unmodified FVT (but none of the modified FVT treatments) led to a significant decrease in NALFD

score (Fig. 1E) and levels of central memory CD8+ T cells, dendritic cells, and M1 macrophages (Fig. 2A,C,G) as compared to obese controls.

- FVT-ChP led to a significant increase in cytokine levels with respect to the other FVT treatments and obese controls (Fig. 3).

- Bacteriome and virome diversity did not significantly differ between the treatments and obese controls (Fig. 4-5).

Major points:

1. First and foremost, it appears that 2 of their 3 FVT treatments (FVT-SDT and FVT-PyT) did not achieve their initial aim, since they measured eukaryotic virus relative abundances of 1.82% and 1.75% for FVT-SDT and FVT-PyT, respectively, as compared to 0.61% for unmodified FVT and 0.015% for FVT-ChP. This should be given more prominence in a paper about modified FVT, instead of being relegated to the last sentence of their Results. Is there any reason to argue for either of these treatments as promising modifications to FVT? If not, then would it be more pertinent to refocus the paper around FVT-ChP?

2. Because most of their results rely on statistical significance testing, the authors need to be a lot more explicit with reporting these results:

a. I could not find the meaning of "a", "b", or "ab" anywhere in the text or captions.

b. What exactly is being tested - difference between treatment and obese control? What statistical test did they use? How significant are their p-values?

c. The number of datapoints in each plot should be stated - it isn't clear that it's always 8 per condition.

d. In Fig. 1D, the authors should state significance with respect to both obese and lean controls, and also at later timepoints. Is it true that at 90 minutes, blood glucose levels are significantly elevated compared to lean controls except for FVT-ChP?

3. The authors should make clear in the Introduction how this study expands upon or differs from their previous work on FVT in a diet-induced obesity model (ref. 24). Is it merely a repeat of previous experiments including modified FVT?

4. The authors should more deeply investigate the reasons underlying their FVT-ChP observations:

a. What is mediating the delayed effect in blood glucose response? They mentioned in the Discussion that this could be caused by elevated levels of Allobaculum. Can the authors demonstrate more directly that this is the cause of the improved blood glucose response?

b. How do the positive and negative aspects of FVT-ChP motivate improvements to FVT modification?

5. Why did gut bacterial Shannon diversity of the controls decrease after the 2nd FVT (Fig. 4A), despite only being given SM buffer?

6. The authors should state how were the mice housed. Individually? Co-housed within each treatment condition?

Minor points:

i. In the Discussion, they stated that they "address the safety considerations associated with the risk of infections by eukaryotic viruses". It does not appear that safety considerations formed a major part of their results.

Reviewer #5 (Remarks to the Author):

The manuscript by Mao et al describes the results of experiments wherein mice exposed to a high-fat diet (or control low-fat diet) were treated with various preparations from mouse feces containing different fractions of the fecal virome. This is very novel work, but there are several concerns regarding the data as presented.

Major concerns

One major concern is the lack of scientific premise or rationale. It is not clear why efficacy in a

model of *C. difficile* infection would lead to the hypothesis that fecal virome transfer (FVT) would be beneficial in diet-induced obesity. Specifically, it is not clear from the introduction which viral preparation method is hypothesized to provide benefits, or why any certain sample treatment is going to preferentially induce changes in the microbiome.

A major methodological concern is the lack of correction for multiple tests, to control the false discovery rate. This should be applied to comparisons of the virome and bacterial microbiome. The sample size is very small and it is unclear whether mice were housed individually or group-housed. If mice were group-housed, cage effects must be considered, and the sample size (per treatment group) becomes questionable. The variability in many of the outcome measures suggests the need for additional samples.

Figures and figure panels are cited in haphazard order (Figure 1C and D cited before 1A and B), or not specifically at all (e.g., only three panels of Figure 2 specifically cited; Figure 3 cited as single figure with no mention of individual panels. If data merits inclusion in a primary figure, it should be described and cited (in alphanumeric order).

It is difficult to interpret the findings shown in Figure 2 without absolute numbers. Percentages of each immune cell subset must be accompanied by absolute cell numbers isolated/counted in each group. Related to this, the authors describe a 'signal of alleviated adipose inflammation' (line 121) in mice that received the unmodified FVT, but considering the lack of difference (from obese control) in macrophages (Fig 2B), B cells (Fig 2D), T helper cells (Fig 2E), CTLs (Fig 2F), Eff memory CD8+ and CD4+ T cells (Fig 2H and 2L), and central memory CD4+ T cells, there would need to be substantial reduction in the absolute number (and not just percentage) of the other cell types listed in the manuscript (DCs, M1 macrophages, and central memory CD8+ T cells) to support that interpretation.

The lack of difference between obese and lean controls in nine of the ten analytes shown in Figure 3 makes it difficult to interpret the stated differences.

Changes in beta-diversity are very minor. PERMANOVA p values should be accompanied by F values. Based on the minimal separation of treatment groups on PCoA at the later timepoints, it is highly speculative to make associations between the microbiome and the metabolic outcomes. If the metabolic outcomes were repeatable in larger cohorts of mice, a mechanistic role for FVT-induced changes in the bacterial microbiome would require parallel studies using germ-free of antibiotic-treated mice to demonstrate a requirement of the gut microbiome for the beneficial effects of FVT.

Minor concerns

There is an apparent disconnect in the Introduction with focused discussion on methods of reducing exposure to potential pathogenic eukaryotic viruses, and the real focus of the study which is the ability of FVT to alter metabolic outcomes or the composition of the gut microbiome. If the ability of the various treatments to eradicate potential pathogens is tangential to the study, perhaps refocus the Introduction to provide mechanistic rationale for why FVT prepared in any specific manner would selectively alter the gut microbiome or host metabolic phenotypes.

Ref 25 is not peer-reviewed. Pre-prints should not be included as peer-reviewed references.

The legend for Figure 5 does not describe panel D.

Please find all responses in the rebuttal letter written in red and the text added to the manuscript is indicated in *italics* within the rebuttal letter, with corresponding line references to the tracked changes in the Word document. Also a clean version of the manuscript is uploaded.

Reviewer 1+3

Comments from Reviewer #1:

We thank the reviewer for helping to improve our manuscript with the constructive comments provided below, all of which we have duly addressed.

The study conducted by Mao et al focused on an interesting topic on the effect of viral/phage transfer on non-alcoholic liver disease. The effects of FVT (fecal viral transfer) on metabolic disorders (obesity/diabetes) have been demonstrated by many reports.

We acknowledge that other studies have investigated the effect of alleviating symptoms of metabolic disorders using FVT. However, to the best of our knowledge there are no more than these three [2–4] as original articles using FVT-like approaches targeting metabolic disorders. One of them is our own, and reference [3] is from a collaborator, Hilde Herrema, reporting a feasibility study in humans. We therefore disagree that similar FVT-like approaches targeting metabolic disorders have been used in many others reports.

What positions this study from other studies is that they focused on phage transplants while eliminating the eukaryotic viruses from the donor fecal virome. However, the gut eukaryotic virome consists of both RNA viruses and DNA viruses. While the authors tried to remove the eukaryotic RNA viruses, the eukaryotic DNA viruses were not removed within the viral transplant; it makes the claimed “eukaryotic-virus-free FVT” not valid.

Thanks for pointing out this important concern, and we regret that we haven’t clearly described the preparation of the fecal viromes that was performed in two other studies [5,6]. We acknowledge that the wording “eukaryotic-virus-free” used in the abstract about the chemostat-propagated virome may be misleading. We have therefore changed the wording to avoid miscommunication.

See line 35-37.

“In contrast to the obese control group, mice administered a modified FVT, nearly depleted from eukaryotic viruses (0.1%), exhibited enhanced blood glucose clearance although without a concurrent reduction in weight gain.”

Despite the ambiguous wording used in the abstract, we would like to emphasize that we do not claim that the eukaryotic viruses are eliminated from the FVT treatments in the manuscript. In line 90-101 we instead describe how we in two other studies[5,6] aimed to develop and evaluate methodologies that selectively either inactivate or removes eukaryotic viruses.

“In two recent studies, we aimed to enhance the safety of FVT by developing methodologies that selectively inactivate[6] (not yet peer-reviewed) or remove the eukaryotic viruses from the fecal matrix[5] (not yet peer-reviewed) while preserving an active enteric phage community. To achieve this, we utilized the differences in key characteristics between eukaryotic viruses and phages[5,6]; most eukaryotic viruses are enveloped RNA viruses[7,8] that only infect eukaryotic cells, and most phages are non-enveloped DNA viruses[8,9] infecting only bacteria. Solvent/detergent treatment (approved by the World Health

Organization for treating blood plasma[10]) was used to inactivate enveloped viruses[11] (FVT-SDT), a compound (pyronin Y) that specifically binds to RNA[12,13] was applied to inactivate RNA viruses (FVT-PyT), and an optimized chemostat fermentation of intestinal inoculum aimed at removing the eukaryotic viruses by dilution[5] (FVT-ChP)."

In[6] we take the advantage of the majority of eukaryotic viruses being enveloped RNA viruses [7,14], here we evaluate the application of solvent-detergent [11,15] treatment to inactivate enveloped viruses, and pyronin Y [12,13] to inactivate RNA viruses. While inactivated viruses may still be physically and chemically interact within the environment (and thereby still detectable by metavirome sequencing), the inactivation ensures that the viruses are no longer infectious, and thereby increasing the safety of the treatments. Whereas in [5] we show that a chemostat-fermentation of enteric donor material can, by dilution, nearly deplete eukaryotic viruses from the fermentation culture (although not completely with the number of dilution volumes applied).

It is important to emphasize that metavirome sequencing does not differentiate between active or inactive viruses and the relative abundance does not account for the quantity of viruses, which is why eukaryotic viral taxa appear in Fig. S10. We have elaborated on this in line 244-247:

"It should be emphasized that the metavirome sequencing can solely be applied to evaluate the removal of viruses, while it cannot differentiate whether viral particles have been inactivated or not from e.g. the solvent-detergent or pyronin Y treatment and the relative abundance does not account for the quantity of viruses."

In addition, the results did not differ significantly between different "eukaryotic-virus-free" FVT groups and unmodified FVT, in obesity phenotype, immune profile, bacterial profile, and viral profile.

We do not agree in the overall conclusion of this comment since we did indeed observe significant differences between treatment groups and controls, but we acknowledge that the presentation of the data should be clearer, and we have therefore updated all figures. The different modified FVTs had effects on several of the following parameters compared to the obese control group.

1. **Diet-induced obesity phenotype:** its correct, that no effect was observed for weight gain and eWAT, but the OGTT was significantly improved by the FVT-ChP treatment. See Fig. 1, Table S2, and Line 126-134.
 - a. *"Oral glucose tolerance tests (OGTT) were conducted on mice after 13 weeks (Fig. S1A-B & Table S1) and 18 weeks (Fig. 1A & 1B) on their respective high-fat or low-fat diet to assess blood glucose regulation. At both study week 13 and week 18, mice on a high-fat diet had significantly ($p < 0.05$) elevated fasting blood glucose levels and impaired glucose regulation during the first 15 minutes after administration compared with the lean control (Fig. 1A & Table S2). Blood glucose levels of mice treated with the chemostat propagated virome (FVT-ChP) decreased sharply from 30 to 60 minutes after glucose administration (Fig. 1A & Table S2), resulting in significantly ($p = 0.021$) improved blood sugar regulation of the FVT-ChP treated mice, compared with the obese control."*

Furthermore, we discovered a pattern of responders/non-responders where the unmodified FVT had significantly improved liver histopathology compared with the obese control. We have extended the manuscript by a section describing this observation of responders/non-responders, and Fig. 5, Fig. S7, S8, S9, and see line 251-279

- b. *"In this study, mice treated with unmodified FVT revealed a tendency of a reduction in the MAFLD activity score compared with obese control mice (Fig. 1C). However, this reduction was not uniformly observed across all mice within the unmodified FVT treated mice (Fig.*

S1C–E). Instead, it was predominantly driven by a subset of mice exhibiting a more pronounced cage independent response to the unmodified FVT treatment (Fig. S1C, S1D & S1E). To understand what caused the different responses to the same treatment, the study subjects were stratified into two distinct groups: those exhibiting a response to the unmodified FVT treatment (Unmodified-FVT Response; characterized by hepatocyte injury score ≤ 1 , inflammation score ≤ 1 , and steatosis ≤ 2 ; number of mice = 4), and those showing no response (Unmodified-FVT Non-Response; with hepatocyte injury score > 1 , inflammation score > 1 , and steatosis > 2 ; number of mice = 4). Despite the low sample size, mice within the unmodified-FVT responder group displayed a significant ($p < 0.05$) downregulation in MAFLD severity (Fig. 5A) and a tendency of improved blood glucose regulation relative to the obese controls (Fig. 5D). The unmodified-FVT response group also demonstrated significantly ($p < 0.05$) reduced proportions of dendritic cells (%CD11c+ of CD45+ leukocytes), activated macrophages (%CD11c+ of F4.80+), T helper cells (%CD4+ of TCRab+), and central memory CD8+ T cells (%CD44+CD62l+ of CD8+) in the adipose tissue, compared to the obese control mice (Fig. S7A, S7C, S7E, & S7G). No clear differences were observed in the proportions of immune cells in the mesenteric lymph node (Fig. S8) or the blood serum cytokine profile (Fig. S9). Moreover, the bacterial profile of the mice responding to the unmodified FVT treatment was significantly separated from the non-responding ($p = 0.028$) and the obese control mice ($p = 0.005$) (Fig. 5F). This divergence was further underscored by the increased relative abundance of *Bacteroides* and *Dehalobacterium*, alongside a significant ($p < 0.05$) increase in *Oscillospira* and a decrease in *Lactobacilli* in the unmodified-FVT responder group relative to the obese controls (Fig. 5H, 5K, 5L, and 5N). These findings imply that responses to the same treatment regimen can vary due to individual differences, however, further studies need to be conducted to validate these observations, since the animal model was neither designed nor hypothesized to include the responder versus non-responder stratification.

- c. **Immune profile:** The FACS analysis of the adipose tissue showed a significant decrease in the cell proportions of dendritic cells, M1 Macrophages, T helper cells and Central memory CD8+ T cells when comparing the unmodified FVT with the obese control. Although not expected, significantly increased levels of the 3 of 10 cytokines (IL-15, TNF- α , MIP-2) were observed for the FVT-ChP group when compared with the obese control. However, interpretation of the blood serum cytokine profile was challenged since the cytokine levels of the lean and obese control were not consistently different, the former Fig. 3 is moved to supplementary Fig. S3. Also see line 166-177.
“Motivated by the observed effects on OGTT and histological measures by the unmodified FVT and FVT-ChP treated mice compared with the lean and obese control groups, we decided only to examine the cytokine profile of these four groups (representing a total of 32 mice). The overall cytokine profile of serum from mice treated with unmodified FVT was not different from that of obese control mice. Therefore, the cytokine profile does not explain the improved liver pathology (Fig. 1C). In contrast, the serum cytokine profile of FVT-ChP treated mice showed significantly elevated ($p < 0.05$) expression of three pro-inflammatory cytokines, including IL-15, TNF- α , and MIP-2 compared to the obese control (Fig. S3). It should be noted that interpretation of the cytokine profiling is challenged by the lack of significant differences between the obese and lean control in 9 out of 10 investigated cytokines.
2. **Bacterial profile:** Indeed no differences were observed at the alpha-diversity level. However, the microbial community profiles (beta-diversity) of the FVT-ChP and unmodified FVT treated groups were significantly altered compared to the obese control, see line 185-187 and Fig. 3B.

- a. *“Except for FVT-SDT ($p = 0.12$), all the different FVT-treated mice harbored a significantly ($p < 0.05$) altered gut bacterial composition compared to the obese control group at termination (Fig. 3B).”*

Furthermore, in Fig. 3D and Fig. S4 and line 190-194 we describe how specific bacterial taxa, where amongst *Allobaculum* and lactobacilli differed in the FVT-ChP group compared with the obese control as determined by differential abundance analysis.

- b. *“The FVT-ChP treated mice were observed with a significant ($p < 0.05$) increase in the relative abundance of *Limosilactobacillus reuteri*, and a tendency ($p < 0.1$) of increase in bacterial taxa belonging to other lactobacilli, *Allobaculum*, and *Bacteroidales* compared with the obese control mice (Fig. 3D).”*

We have in the revision added a heatmap showing a more complete overview of taxa significantly differentiating between the obese control and the different FVT treatments to extend the analysis further (Fig. 3D). Furthermore, three correlation analyses between a) the bacteriome and the phenotypical data (Fig. 3E), b) the bacteriome and immune cell count in adipose tissue (Fig. 3F), and c) the bacteriome and virome are included (Fig. S6), see line 198-222.

- a. *“Correlation analyses were performed to elucidate potential links between the bacterial GM component and the alleviating effects associated with the unmodified FVT and FVT-ChP. Using phenotypic characteristics and measured proportions of immune cells in adipose tissue as indicators of metabolic syndrome of the mice, we performed Spearman correlation analysis (FDR corrected $p < 0.05$, $|\text{Spearman coefficient}| > 0.4$) between these indicators and the most abundant bacterial taxa (relative abundance $> 1\%$). The improved glucose regulation was positively correlated with the relative abundance of *Allobaculum* spp. (coefficient = -0.47) and negatively correlated with the relative abundance of *Lactococcus* spp. (coefficient = 0.41) (Fig. 3E). The elevated relative abundance of *A. muciniphila* (coefficient = -0.43), *Bacteroides* spp. (coefficient = -0.51), *Oscillospira* spp. (coefficient = -0.42) were correlated with lower MAFLD activity scores. In contrast, the relative abundance of unclassified lactobacilli was positively correlated (coefficient = 0.52) with the MAFLD activity score (Fig. 3E). When the size of eWAT increase, the relative abundance of taxa *Allobaculum* spp. (coefficient = -0.46) tended to be diminished, while the relative abundance of *Lactococcus* spp. (coefficient = 0.68) decrease (Fig. 3E). The bodyweight gain of the mice was negatively correlated (coefficient = -0.41) with the relative abundance of *A. muciniphila* (Fig. 3E).*

*Furthermore, the population of the immune cell proportions in the eWAT significantly correlated with gut bacteriome. The pairwise Spearman correlations were established between the proportions of immune cells from adipose tissue and bacterial taxa. The abundance of unclassified lactobacilli exhibited a positive correlation (respective coefficients = 0.54 , 0.52 , 0.63 , and 0.41) with the proportions of activated macrophages, central memory CD8+ T cells, dendritic cells, and cytotoxic T cells (Fig. 3F). Conversely, the abundance of *Oscillospira* spp. demonstrated an inverse correlation with the proportions of four immune cells (respective coefficients = -0.65 , -0.59 , -0.56 , and -0.42) (Fig. 3F). The negative correlation was also found between *Bacteroides* spp. and proportions of activated macrophages (coefficient = -0.44), as well as cytotoxic T cells (coefficient = -0.57) (Fig. 3F).”*

- b. We have also included an additional 16S rRNA gene data at the time just before the 1st FVT treatments to improve the resolution of the longitudinal development of the bacterial profiles (Fig. 3).

3. Virome profile: The same “picture” applies for the virome, where no differences were observed at the alpha-diversity level, while at the beta-diversity level the gut viromes likely resulting from

transfer of unmodified FVT and FVT-PyT were significantly (adj. $p < 0.05$) different compared to the obese control, and FVT-ChP and FVT-SDT showed tendencies (adj. $p < 0.1$) (Fig. 4). Here we have made a heatmap of the overall differential abundant taxa of both predicted bacterial hosts as well as the viral taxonomy (Fig. S5). However, during the time of revision we finalized an improved pipeline for the metavirome assembly and analysis which have consequently resulted in updated figures in Fig. 4, Fig. S5, Fig. S6, Fig. S10, as well as added text to line 224-249:

“None of the FVT treatments affected gut viral diversity (Shannon diversity index), but the viral composition (Bray-Curtis dissimilarity) of mice treated with FVT was all significantly ($p < 0.05$) different from the obese control at termination (Fig. 4B). This suggests that introducing a new viral community through FVT leads to changes in the recipient’s gut viral composition. The taxonomic profiles of the recipient gut viromes were dominated by Microviridae, Petitvirales and Tubulavirales-associated viruses (Fig. 4C), and the predicted (based on viral contigs) bacterial hosts were dominated by the genera of Mucispirillum, Bacteroides, and Prevotella (Fig. 4D). Differential viral relative abundance was analyzed at the level of viral contigs (vOTUs) (Fig. 4E & S5) to support the observed differences in viral composition. When comparing both FVT-ChP and unmodified FVT treated mice to the obese control at the study termination, significant ($p < 0.05$) differences were observed in the relative abundance of viruses belonging to the family Microviridae, order Petitvirales, and class Caudoviricetes (Fig. 4E & S5). Pairwise Spearman’s correlation analysis was conducted to investigate the influence of viral composition on the gut bacteriome. The relative abundance of different viral contigs of the family Microviridae both negatively (coefficient < -0.4) correlated with the genus Allobaculum and positively (coefficient > 0.4) correlated with unclassified lactobacilli, Bacteroides spp., and the family Clostridiaceae (Fig. 4F & S6). Furthermore, the viral contigs of order Petitvirales showed a strong positive (coefficient > 0.4) correlation with the genus of Lactococcus (Fig. 4F & S6). Eukaryotic viruses appeared to be nearly depleted in the FVT-ChP in terms of relative abundance (0.1%) compared to the FVT-SDT (3.66%), unmodified FVT (1.07%), and FVT-PyT (7.41%) (Fig. S10E). It should be emphasized that the metavirome sequencing can solely be applied to evaluate the removal of viruses, while it cannot differentiate whether viral particles have been inactivated or not from e.g. the solvent-detergent or pyronin Y treatment and the relative abundance does not account for the quantity of viruses. Fecal samples of mice were excluded from bacteriome (11 samples) and virome (6 samples) analysis due to lack of sample or low sequencing reads (Fig S12).”

Moreover, there is a lack of mechanistic evidence/insights for furthering our understanding in FVT.

Thanks, we agree that we need to address the mechanistic of the FVT-driven changes in host phenotypes and gut microbiome (GM) profiles. Since the research field of FVT (and even FMT) is yet in its infancy it is not possible to provide any mechanistic evidence, instead we have proposed a hypothesis based on the current literature.

Our hypothesis can be divided into two arms.

- 1) bacteriophage-driven restoration/modulation of the GM,
- 2) stimulation and activation of the immune system.

1. Several studies have shown how the bacteriophage donor profile seems to be transferred to the gut of *C. difficile* patients after FMT/FVT [16–18], and we have previously shown that FVT from lean mice could push the GM composition in obese mice to look more as a “lean” profile [4]. Furthermore, we recently showed how a FVT using donors with a relative high fecal abundance of *Akkermansia muciniphila*, could significantly increase the abundance of the enteric endogenous (native) *A. muciniphila* in the recipients of the FVT [19]. With these observations in mind, we speculate that FVT donor phenotype at least partially can be transferred to the recipient. Upon transfer, the phages will influence bacterial abundance establishing an ecosystem with similarities to its origin. It is admittedly counterintuitive how this should fit to the general notion of bacteriophages being highly strain specific. However, a recent study suggest that phages may interact with hosts that span distantly related microbial domains [20], where amongst other phage satellites [21,22] could contribute to broader host ranges. In a previous review [23], we suggested that it could be cascading events where some phages infect their hosts which indirectly affect the ecosystem by changing the balance of competition between other bacteria. This was very elegantly shown in a gnotobiotic mouse model by Hsu et al. [24] where the decrease in one bacterium due to phage infections increased the abundance of one or more of the other bacteria in a simplified GM consortium. Hence, similar effects may apply to FVT that are far more complex and undefined than the few phages used by Hsu et al.. FVT has the potential to alter the balance of the bacterial ecosystem, consequently influencing the overall metabolome, as evidenced in this study [4]. This alteration in the metabolome may give rise to systemic effects, manifesting as observable changes in phenotype. Another likely factor is the transfer of potential useful metabolic genes from temperate phages to their bacterial hosts [25–28] (although only sparsely studied in the gut) which again could improve the competitiveness of some of the hosts, and thereby contributing to the overall GM changes. This fits well to another recent study showing how the nutritional and host environment determines community ecology [29], so if cascading events initiated by the FVT have caused in conditional changes in the ecosystem, these changes may then further catalyze the effect of FVT.
2. The effect of the immune system on gut health should not be neglected. During the last couple of years, it seems more and more likely that bacteriophages to some extent react with our immune system through e.g. TLR3 and TLR9 [30,31], and a recent review summarize the current knowledge on the immunogenicity of bacteriophages and e.g. highlight similarities between eukaryotic viruses and bacteriophages (see figure below) [32]. Thus, a stimulation of the immune system may also contribute to some of the mechanisms behind FVT.

Figure copied from [32].

We have therefore added the text below to the discussion (see lines 377-401):

*“The mechanisms behind the GM modulating effects of FVT are still poorly understood, but accumulating reports suggest that the phenotypic traits of FVT donors, to some extent, can be transferred to recipients, as the phages might catalyze a modulation of the recipient ecosystem to be similar to their origin. This is exemplified by phage donor profiles being transferred to the gut of *C. difficile* patients after successful FMT/FVT[16–18], FVT from lean mice could shift the GM composition in obese mice to resemble that of lean individuals[4], and FVT donor material originating from an ecosystem with a relatively high abundance of *A. muciniphila*, could significantly increase the abundance of the enteric endogenous (native) *A. muciniphila* in mice that received the FVT[19]. This may be driven by cascading events[23], as demonstrated in a gnotobiotic mouse model[24], where phage infections indirectly influence the bacterial balance. Thus, the far more complex viral composition of FVT could similarly alter the bacterial ecosystem and as observed in our previous study, affect the blood metabolome and GM, leading to systemic effects[4]. This concept might seem counterintuitive given the general belief in the strain-specific nature of phages, however, a recent study proposed that phages could interact with distantly related microbial hosts[20]. Phage satellites have, amongst other, been suggested to contribute to broader host ranges[21,22]. Also the transfer of potentially beneficial metabolic genes from temperate phages to their bacterial hosts[25–28], may enhance host competitiveness and contribute to overall GM changes. These findings align with recent research demonstrating how metabolic functions and bacterial interaction networks were context-dependent of variables like nutrition, host environment, and bacterial compositions[29], supporting the hypothesis that cascading events initiated by FVT could catalyze GM modulating effects. In addition to the bacteria-phage relations, the impact of the immune system on gut health should not be neglected. Recent evidence suggests that bacteriophages interact with our immune system through mechanisms like TLR3 and TLR9[30,31]. A recent review has summarized the current understanding of phage immunogenicity highlighting parallels with eukaryotic viruses[32]. Stimulation of the immune system may thus be another mechanism behind the efficacy of FVT.”*

In addition, I have several other concerns:

1. How many mice were used in each group and what statistical test should be stated.

Thank you for bringing to our attention these missing pieces of information. It is essential that these details are clearly stated. In our study, each group consisted of 8 mice, which were divided into two cages, 4 in each cage. Considering the experimental data did not follow the normal distribution, we used the Wilcoxon rank sum test, a non-parametric statistical hypothesis test, with FDR correction to evaluate changes between FVT treatments mice and obese control mice in both phenotype and GM composition. The obese and lean control mice were also compared with Wilcoxon rank sum test, to show that high-fat diet induced obesity model of mice had worked. The number of mice included in each analysis is both present in the x-axis of figures or described in the result. Single animals were considered as the experimental units. However, we acknowledge the findings from a new paper published by Aaron Ericsson's group, which suggests that decreasing housing density can improve statistical power [33]. This is a valuable insight, and we will certainly take it into consideration in future studies. Please refer to Line 440-442 for the rationale behind our chosen sample size.

We have incorporated these details into the methods section at Line 546-549. Furthermore, we have ensured that the manuscript adheres to the ARRIVE E10 guidelines. As a result, sample sizes are now explicitly highlighted for each figure whenever possible, and the relevant statistical information is provided figure legend or described in the result (and of course the methods).

2. Line 136-137, the authors reasoned that FVT-ChP induced undesirable increase in systemic inflammation. However, it did not investigate why this happened? Is it because of specific viral transfer into the blood? In contrast, unmodified FVT might be result in undesired eukaryotic viral transfer.

Thank you for highlighting these ambiguous observations, which have puzzled us as well. From a scientific standpoint, we believe it is crucial to report results that may not necessarily be easily explained in detail, as they could hold biological significance or of course just represent artifacts. Therefore, we are reporting these observations while refraining from making strong claims and have toned down the impact of these results, due to the lack of significant differences between the lean and obese control groups in most of the analysis.

As mentioned earlier, the growing interest in phage immunogenicity is due to its potential role in maintaining and modulating our immune system [32], and since phages may be passively and actively present in various parts of our body [34]. Interactions between phages transferred with FVT-ChP and the host immune system may partly explain our observation of significantly increased levels of 3 out of 10 tested cytokines in the FVT-ChP group compared to the obese controls (Fig. S3). It is important to emphasize that the highly selective setup of the chemostat-propagated viromes sets the FVT-ChP apart from other viromes (see Fig. S10), which could result in different interactions with the immune system compared to the other FVT treatments. Nevertheless, we currently lack evidence to pinpoint specific viruses that might have triggered such an immune response in the blood.

The reviewer is absolutely correct that one of the challenges regarding the safety of unmodified FVT, as well as the more widely used FMT, is the risk of transferring uncharacterized eukaryotic viruses that may go undetected in screening assays. This concern forms the basis of our main project hypothesis: the idea that by removing or inactivating the majority of eukaryotic viruses from FVT while preserving the GM-modulating effects reported by us and others [4,6,18,19,35,36], we can mitigate this risk of infection. However, based on our cytokine profiling and the analysis of immune cell counts in the adipose and MLN tissue, we cannot claim that unmodified FVT leads to an increased inflammatory response in the host. On the contrary, we observed a decrease in the proportions of activated M1 macrophages, dendritic cells, T

helper cells, and central memory CD8+ T cells (see Fig. 2) in the adipose tissue of mice treated with unmodified FVT compared to the obese control.

We acknowledge the importance of these considerations and have incorporated the following text at Line 319-328 in the manuscript to address these points:

“Phage immunogenicity has garnered increased interest due to its potential role in regulating our immune system by sharing similar structures and proteins with eukaryotic viruses [32]. Phages may be both passively and actively distributed throughout various parts of our body [34]. The interactions between phages transferred along with the chemostat-propagated virome and the host immune system could help explain our observations of increased levels of three out of ten cytokines in the FVT-ChP treated mice, compared with the obese control. However, we lack evidence of specific viruses that could have triggered this immune response in the blood serum. Although low-grade systemic inflammation is connected to metabolic syndrome [37,38], it does not exclude the possibility of improved glucose regulation alongside increased levels of some cytokines in blood serum.”

In addition, how to reconcile the increased systemic inflammation and the relatively better OGTT test phenotype?

In light of the observations mentioned above, we are equally puzzled by this counterintuitive finding. Unfortunately, we lack data that can fully explain this phenomenon and our measures are challenged by inconsistent differences of the cytokine levels between the lean and obese control groups, so we decided to tone down the impact of these measures. However, in our discussion at Line 339-348, we speculate that the increased expression of IL-15 may have contributed to the improved glucose tolerance [39], while the elevated cytokine levels of MIP-2 and TNF- α could potentially be linked to or indicative of the MAFDL status of the liver tissue in the FVT-ChP treated mice. It is important to note that while low-grade systemic inflammation is associated with metabolic syndrome [37,38], it does not necessarily rule out the possibility of improved glucose regulation in conjunction with increased levels of certain cytokines. To underscore the limitations of these results, we have included the following text at Line 326-328:

“Although low-grade systemic inflammation is connected to metabolic syndrom[37,38], it does not exclude the possibility of improved glucose regulation alongside increased levels of some cytokines in blood serum.”

3. The bacteria analysis is lack of depth or resolution. ASV, OTU, or metagenomic analysis might be more conducive. So is the virome analysis.

We regret any lack of clarity in our previous communication. As described earlier, we did include a more in-depth analysis of the virome and bacteriome, which can be seen in Figure S4, S5, and S6. However, we felt the need to further enhance the bioinformatic analysis of GM compositions by reanalyzing the complete dataset using an improved metavirome characterization pipeline which have a deeper classification on viral contigs, focusing on both viral taxonomy and host predictions. We have also added a heatmap of the bacteriome (see Fig 3D) and the correlation networks between phenotype and bacteriome (see Fig 3E-F). For virome, we added two heatmaps with different resolution in the manuscript and supplementary material (see Fig 4E & S5, S6). A correlation heatmap between bacteriome and virome were also added in Fig 4F and S6. We hope that the additional analysis addresses this concern. We have expanded bioinformatic analysis which also can be found at:

line 183-196:

“At the termination of the study, the FVT-ChP treated mice had a significantly lower ($p = 0.028$) bacterial diversity compared with obese control mice (Fig. 4A & 3A). Except for FVT-SDT ($p = 0.12$), all the different FVT-treated mice harbored a significantly ($p < 0.05$) altered gut bacterial composition compared to the obese control group at termination (Fig. 3B). The dominant bacterial phyla detected in the mice feces were Firmicutes, Bacteroidetes, Proteobacteria, Verrucomicrobia, Deferribacteres, and Actinobacteria (Fig. S4J). We adopted DESeq2 to identify differentially abundant bacterial taxa between the FVT treatment groups and obese control mice. The FVT-ChP treated mice were observed with a significant ($p < 0.05$) increase in the relative abundance of *Limosilactobacillus reuteri*, and a tendency ($p < 0.1$) of increase in bacterial taxa belonging to other lactobacilli, *Allobaculum*, and *Bacteroidales* compared with the obese control mice (Fig. 3D). On contrary unmodified FVT treated mice had a significant ($p < 0.05$) decrease in the relative abundance of bacteria belonging to the genus *Allobaculum* and a tendency ($p < 0.1$) of increase was observed for *Erysipelotrichaceae* and *Bacteroidales* (Fig. 3D).”

line 198-222:

“Correlation analyses were performed to elucidate potential links between the bacterial GM component and the alleviating effects associated with the unmodified FVT and FVT-ChP. Using phenotypic characteristics and measured proportions of immune cells in adipose tissue as indicators of metabolic syndrome of the mice, we performed Spearman correlation analysis (FDR corrected $p < 0.05$, $|\text{Spearman coefficient}| > 0.4$) between these indicators and the most abundant bacterial taxa (relative abundance $> 1\%$). The improved glucose regulation was positively correlated with the relative abundance of *Allobaculum* spp. (coefficient = -0.47) and negatively correlated with the relative abundance of *Lactococcus* spp. (coefficient = 0.41) (Fig. 3E). The elevated relative abundance of *A. muciniphila* (coefficient = -0.43), *Bacteroides* spp. (coefficient = -0.51), *Oscillospira* spp. (coefficient = -0.42) were correlated with lower MAFLD activity scores. In contrast, the relative abundance of unclassified lactobacilli was positively correlated (coefficient = 0.52) with the MAFLD activity score (Fig. 3E). When the size of eWAT increase, the relative abundance of taxa *Allobaculum* spp. (coefficient = -0.46) tended to be diminished, while the relative abundance of *Lactococcus* spp. (coefficient = 0.68) decrease (Fig. 3E). The bodyweight gain of the mice was negatively correlated (coefficient = -0.41) with the relative abundance of *A. muciniphila* (Fig. 3E).

The pairwise Spearman correlations were established between the proportions of immune cells from adipose tissue and bacterial taxa. The abundance of unclassified lactobacilli exhibited a positive correlation (respective coefficients = 0.54 , 0.52 , 0.63 , and 0.41) with the proportions of activated macrophages, central memory CD8⁺ T cells, dendritic cells, and cytotoxic T cells (Fig. 3F). Conversely, the abundance of *Oscillospira* spp. demonstrated an inverse correlation with the proportions of four immune cells (respective coefficients = -0.65 , -0.59 , -0.56 , and -0.42) (Fig. 3F). The negative correlation was also found between *Bacteroides* spp. and proportions of activated macrophages (coefficient = -0.44), as well as cytotoxic T cells (coefficient = -0.57) (Fig. 3F).

line 224-242:

“None of the FVT treatments affected gut viral diversity (Shannon diversity index), but the viral composition (Bray-Curtis dissimilarity) of mice treated with FVT was all significantly ($p < 0.05$) different from the obese control at termination (Fig. 4B). This suggests that introducing a new viral community through FVT leads to changes in the recipient’s gut viral composition. The taxonomic profiles of the recipient gut viromes were dominated by Microviridae, *Petitvirales* and *Tubulavirales*-associated viruses (Fig. 4C), and the predicted (based on viral contigs) bacterial hosts were dominated by the genera of *Mucispirillum*, *Bacteroides*, and *Prevotella* (Fig. 4D). Differential viral relative abundance was analyzed at the level of viral contigs (vOTUs) (Fig. 4E & S5) to support the observed differences in viral composition. When comparing both FVT-ChP and unmodified FVT treated mice to the obese control at the study termination, significant ($p < 0.05$) differences

were observed in the relative abundance of viruses belonging to the family Microviridae, order Petitvirales, and class Caudoviricetes (Fig. 4E & S5). Pairwise Spearman's correlation analysis was conducted to investigate the influence of viral composition on the gut bacteriome. The relative abundance of different viral contigs of the family Microviridae both negatively (coefficient < -0.4) correlated with the genus Allobaculum and positively (coefficient > 0.4) correlated with unclassified lactobacilli, Bacteroides spp., and the family Clostridiaceae (Fig. 4F & S6). Furthermore, the viral contigs of order Petitvirales showed a strong positive (coefficient > 0.4) correlation with the genus of Lactococcus (Fig. 4F & S6)."

Reviewer 2+4

Comments from Reviewer #2:

Thanks a lot for your time and constructive comments which we have addressed accordingly.

In this manuscript, Mao et al. investigate the effect of fecal virome transplantation (FVT) in a mouse model of diet-induced obesity. They argue that unmodified fecal filtrates may contain harmful levels of eukaryotic viruses, and thus prepared 3 modified FVT treatments for testing: FVT-ChP (chemostat-propagated virome to dilute out eukaryotic viruses), FVT-SDT (solvent/detergent treated to inactivate enveloped viruses), and FVT-PyT (pyronin Y treated to inactivate RNA viruses).

They reported that:

- None of the FVT treatments affected body weight and eWAT size with respect to obese controls (Fig. 1A-B).
- Interestingly, FVT-ChP (but none of the other treatments) led to a significant decline in blood glucose levels 60 (but not 30) minutes after glucose administration, as compared to obese controls (Fig. 1D).
- Unmodified FVT (but none of the modified FVT treatments) led to a significant decrease in NALFD score (Fig. 1E) and levels of central memory CD8⁺ T cells, dendritic cells, and M1 macrophages (Fig. 2A,C,G) as compared to obese controls.
- FVT-ChP led to a significant increase in cytokine levels with respect to the other FVT treatments and obese controls (Fig. 3).
- Bacteriome and virome diversity did not significantly differ between the treatments and obese controls (Fig. 4-5).

Major points:

1. First and foremost, it appears that 2 of their 3 FVT treatments (FVT-SDT and FVT-PyT) did not achieve their initial aim, since they measured eukaryotic virus relative abundances of 1.82% and 1.75% for FVT-SDT and FVT-PyT, respectively, as compared to 0.61% for unmodified FVT and 0.015% for FVT-ChP. This should be given more prominence in a paper about modified FVT, instead of being relegated to the last sentence of their Results.

Thank you for bringing up this important question, which supports the need for clearer communication from our side.

First, the percentages we mention in the results are meant to highlight that the majority of eukaryotic viruses are removed from the FVT-ChP treatment through dilution during the chemostat fermentation, a

point also presented in another manuscript currently under review [5]. The use of solvent-detergent in our study is a conventional method for inactivating enveloped viruses [10,11,15] while pyronin Y represents a more exploratory approach for inactivating RNA viruses, owing to its potential cytostatic/cytotoxic properties in its interaction with RNA [12,13]. In a separate study (also currently undergoing revision [6]), we evaluated the efficacy of solvent-detergent and pyronin Y in inactivating enveloped and/or RNA viruses. Although these viruses are inactivated, their viral particles remain intact and thus detectable for sequencing, as reflected in the mentioned relative abundance. Our claim of inactivation depends on the results from that study [6], **not** the sequencing results shown in this current study. Sequencing serves only as a suitable method to assess the "removal" of viruses, not viral inactivation.

To make this clearer, we have incorporated the following text at Line 244-247:

"It should be emphasized that the metavirome sequencing can solely be applied to evaluate the removal of viruses, while it cannot differentiate whether viral particles have been inactivated or not from e.g. the solvent-detergent or pyronin Y treatment and the relative abundance does not account for the quantity of viruses."

With the updated viral bioinformatic pipeline the relative abundance of eukaryotic viruses changed due to its improved taxonomic resolution. However, the relative differences between the FVT inoculums remains similar, hence the chemostat propagated virome had a clearly lower relative abundance of eukaryotic viruses compared with the other FVTs.

This is updated in manuscript at Line 242-244:

"Eukaryotic viruses appeared to be nearly depleted in the FVT-ChP in terms of relative abundance (0.1%) compared to the FVT-SDT (3.66%), unmodified FVT (1.07%), and FVT-PyT (7.41%) (Fig. S10E)."

The attentive reader would also notice that the modified FVT-SDT and FVT-PyT remain to appear with a higher relative abundance of eukaryotic viruses compared with the unmodified FVT. It could be speculated that the inactivation with solvent detergent and pyronin Y may also destroy a certain fraction of phages, which then would affect the distribution of relative abundance between phages and eukaryotic viruses. Although additional experimental data would be needed, it fits with our other study that also investigates the efficacy of these two methods [6]. Here the pyronin Y treatment clearly decreased the phage activity of most investigated phages, while solvent detergent only decreased phage activity for one of the investigated phages (see figure below).

To address this observation, we have added the following at line 402-410:

"Both the modified FVT-SDT (3.66%) and FVT-PyT (7.41%) exhibited a notably higher relative abundance of eukaryotic viruses when compared to the unmodified FVT (1.07%). It could be speculated that the inactivation process involving solvent detergent and pyronin Y might also lead to the destruction of a certain fraction of phages. Consequently, this could impact the relative abundance distribution between phages and eukaryotic viruses. While further experimental data is required to substantiate this hypothesis, it aligns with our other study that delves into the efficacy of these methods³⁰. In this parallel investigation, pyronin Y reduced the phage activity of the majority of the examined phages, whereas solvent/detergent only affected the activity of one of the tested phages."

Figure another study that are under revision[6]: Evaluation of inactivation of phage activity (plaque-forming units, PFU/mL) with solvent/detergent (S/D) or pyronin-Y treatment was evaluated on their respective bacterial hosts. A) Three non-enveloped phages (phiX174, C2, and T4) and one enveloped phage (phi6) were treated with solvent/detergent. The solvent/detergent treatment completely inactivated the enveloped phage phi6, as indicated by PFU/mL dropping to below detection limit, while the non-enveloped phages phiX174 and T4 remained unaffected. Phage C2, on the other hand, experienced a minor decrease of 1 log₁₀ in PFU/mL. B) Phages representing ssDNA (phiX174), dsDNA (C2 and T4), ssRNA (MS2), and dsRNA (phi6) were treated with pyronin-Y to inhibit RNA phage activity. After exploring various combinations of pyronin-Y concentrations, temperatures, and incubation times, an overnight incubation at 40°C with 100 μM pyronin-Y was chosen. This treatment resulted in a 5 log₁₀ reduction in PFU/mL for the ssRNA phage MS2 and complete inactivation of the dsRNA phage phi6. Phi6 demonstrated temperature sensitivity as incubation at 40°C alone was sufficient to inactivate this enveloped phage, while incubation temperature at 20°C along with 100 μM pyronin-Y led to a reduction of 4 log₁₀ PFU/mL of phage phi6. Unfortunately, the pyronin-Y treatment at 40°C also affected the plaque-forming ability of phages C2 (dsDNA), T4 (dsDNA), and phiX174 (ssDNA), resulting in reductions of 1, 2.5, and 5 log₁₀ PFU/mL, respectively. Dashed lines mark the detection limit of the applied assay.

Is there any reason to argue for either of these treatments as promising modifications to FVT? If not, then would it be more pertinent to refocus the paper around FVT-ChP?

Our project was built on the hypothesis that the removal or inactivation of eukaryotic viruses from FVT could both improve safety while maintain treatment efficacy similar to what we had previously demonstrated with unmodified FVT in the context of targeting diet-induced obesity [4], and as observed by other researchers in the case of *C. difficile* infections in humans [18]. The rationale behind this approach was to employ the diet-induced obesity model to represent a complex gut-associated disease, while the *C. difficile* model allowed us to assess a comparatively "simpler" gut-associated disease primarily caused by the overgrowth of one specific pathogen.

Regrettably, our findings showed that FVT-PyT had no discernible effect in any of our models, whereas FVT-SDT demonstrated the ability to protect all treated mice from reaching the humane endpoint in the *C. difficile* model [6], but no clear effect in the diet-induced obesity model. Given the fundamentally different disease etiologies underlying these models, it is not surprising that different modified FVTs yielded different results. Consequently, it is relevant to emphasize the significance of FVT-SDT, rather than just FVT-ChP. We believe it is important to transparently report both the outcomes that worked and those that did not, which is why we included the results for FVT-PyT and FVT-SDT.

We have already included similar rationale in the introduction of the initial submission at line 78-95, but have also added the following at line 115-122:

“As a follow-up study, we hypothesize that different techniques that either deplete or inactivate the eukaryotic viral component in FVT, can improve safety while maintaining the alleviating effects of FVT on symptoms associated with metabolic syndrome [4]. The treatment efficacy of these different modified FVTs (FVT-ChP, FVT-SDT, and FVT-PyT) was compared with unmodified FVT and saline treatment of obese control mice. Lean control mice were also included to evaluate the validity of the conducted diet-induced obesity mouse model. All the transferred intestinal donor content used in the study originated from the same mixed donor material originating from lean male mice.”

+ line 428-430:

“In contrast, the two other FVT modifications (FVT-SDT and FVT-PyT) did not appear as promising modification strategies of FVT targeting diet-induced obesity, since no clear phenotypical improvements were observed.”

2. Because most of their results rely on statistical significance testing, the authors need to be a lot more explicit with reporting these results:

a. I could not find the meaning of "a", "b", or "ab" anywhere in the text or captions.

b. What exactly is being tested - difference between treatment and obese control? What statistical test did they use? How significant are their p-values?

Thank you for your comments regarding points (a) and (b). We recognize that using letters in this context may not be the clearest way to present our data. In response, we have adopted the more commonly accepted asterisk (*) notation in all figures and have provided detailed descriptions of the methods used in the figure legends. This update has been applied to all included figures, including all supplemental figures.

Furthermore, to enhance clarity and transparency, we have added tables containing all relevant statistics of oral glucose tolerance at each time point at study week 13 and 18, both with and without corrections, along with a full description of the tests that we have employed. You can find this in Table S1 and S2. We

have also included all the relevant statistics in the GitHub page of this study (<https://github.com/MaoAria15/DIO>).

c. The number of datapoints in each plot should be stated - it isn't clear that it's always 8 per condition.

We fully agree that including the number of data points in our figures enhances transparency and aligns with the ARRIVE E10 guidelines. To address your comments and improve the clarity of our presentation, we have updated all figures to include the number of data points included in each plot. The reasons for excluding certain data points were also described in the result.

d. In Fig. 1D, the authors should state significance with respect to both obese and lean controls, and also at later timepoints.

Thank you for your comment. We chose to include both tAUC and continuous blood glucose measurements to enhance transparency regarding which time points contributed to the improved glucose regulation in the FVT-ChP treated mice. However, we acknowledge the importance of highlighting the effectiveness of the model. We conducted statistical tests between obese and lean mice at all time points in the figure, and the remaining comparisons can be found in the Table S1 and S2. The Fig. 1D has been changed to Fig. 1A.

The data structure of our study did not align with the prerequisites for employing the linear regression model we used previously. Instead of adhering to the linear regression model, we changed our statistical approach and opted for the Wilcoxon rank sum test, a non-parametric statistical hypothesis test. To enhance clarity in our statistical analysis, we formulated two key questions for exploration:

1. Can modified and unmodified viral transplantation induce a discernible difference in mice phenotype and microbiome compared to obese control mice?
2. Does the high-fat diet-induced obesity mice model effectively work in comparison to lean control mice fed a low-fat diet?

Consequently, our focus was solely on comparing each FVT treatment against the obese control and assessing the obese mice against the lean control. Given that the primary inquiry of this study revolves around the first question, we excluded the comparison between obese and lean control mice from the False Discovery Rate (FDR) correction to mitigate the risk of false negatives. In the updated results, the blood glucose level exhibited a significant decrease compared to the obese control (Table S2), indicating that treating mice with chemostat-propagated FVT leads to improved blood glucose regulation. We also put the detailed statistic result in our GitHub page: <https://github.com/MaoAria15/DIO>.

Is it true that at 90 minutes, blood glucose levels are significantly elevated compared to lean controls except for FVT-ChP?

Yes, this is correct. Please see Table S2.

3. The authors should make clear in the Introduction how this study expands upon or differs from their previous work on FVT in a diet-induced obesity model (ref. 24). Is it merely a repeat of previous experiments including modified FVT?

Thank you for your feedback, and we agree that this point should be clarified. The DIO model in this study is replicated as closely as possible to our previous publication in which we exclusively used **unmodified** FVT [4]. We recognize that widespread use of FVT is unlikely unless the challenges posed by donor

variability and the safety concern of the potential transfer of eukaryotic viruses are effectively addressed. As a result, our objective in this study is to identify the most suitable FVT modification that can effectively address these challenges while preserving its treatment efficacy. The development of safer and more reproducible FVTs would make it easier to apply the treatment in clinical settings targeting lifestyle-associated diseases like obesity and type-2 diabetes.

To provide further clarity regarding the study's objective and how it distinguishes itself from our previous work, we have expanded on this point at Line 108-111.

“Our main objective was therefore to screen whether or not the differently modified FVTs[5,6] had the potential to alleviate symptoms on co-morbidities of metabolic syndrome as a safer and/or more reproducible alternative to the unmodified FVT that we previously have used in a similar diet-induced obesity model[4].”

4. The authors should more deeply investigate the reasons underlying their FVT-ChP observations:

a. What is mediating the delayed effect in blood glucose response? They mentioned in the Discussion that this could be caused by elevated levels of *Allobaculum*. Can the authors demonstrate more directly that this is the cause of the improved blood glucose response?

Thanks, we really appreciate this comment. Mapping the direct molecular mechanisms and effects of *Allobaculum* would necessitate extensive experiments, which are beyond the scope of this study. However, existing literature may provide insights into how *Allobaculum* could be implicated in the improved glucose response. It's important to note that our 16S rRNA gene sequencing can only assess the genus *Allobaculum* and not its specific species or strain.

The below text is added at line 346-361:

*“Prior studies have consistently identified impaired intestinal integrity, thickness of mucus layer, and immunity (low-grade inflammation) as important factors contributing to the development of metabolic syndromes related diseases[40–43]. Mucin degrading bacteria like *Akkermansia muciniphila*[41,44,45] and *Allobaculum spp.*[46–48] have been reported to play a key role in these processes. *Allobaculum* has been reported as a particularly active glucose utilizer that produces lactate, butyrate[49], and impacts the metabolism of long chain fatty acids[50]. Through the production of short-chain fatty acids like butyrate, *Allobaculum* contributes to the protection of the intestinal barrier[46] stimulates the immune system, and putatively protects against metabolic syndrome[47]. This would be in accordance with supplementation of butyrate limiting hyperglycemia through the regulation of amongst glucagon-like peptide-1 (GLP-1) and insulin in serum[51]. This is further supported by *Allobaculum* being reported to be positive correlated to hypoglycemia and negative correlated to HOMA-IR levels[52] (not yet peer-reviewed). Taken together, the current understanding of the role of *Allobaculum* on metabolic syndrome supports our observations of improved glucose regulation for the FVT-ChP treated mice that had increased relative abundance of *Allobaculum* compared with obese control (Fig. S4A).”*

b. How do the positive and negative aspects of FVT-ChP motivate improvements to FVT modification?

Thank you for bringing up this important perspective. To summarize the positive aspects of using a chemostat-propagated phageome (FVT-ChP), even in this early phase of development, it has demonstrated phenotypical effects in two distinct disease models: diet-induced obesity and *C. difficile* infection [6]. FVT-ChP addresses several significant challenges that currently hinder broader clinical applications of unmodified FVT or FMT.

- 1) **Procurement and Reproduction:** It allows for the procurement and reproduction of a sufficient quantity of suitable enteric phage solutions.
- 2) **In-Vitro Screening:** With an ample and reproducible supply of phage solutions, it becomes feasible to perform *in vitro* screening of various chemostat-propagated phageomes to ensure compatibility with the highly individual GMs of recipients [53]. Such screening could, for instance, assess if obesity-associated taxa decrease in abundance and whether the overall GM profile shifts toward resembling a lean and healthy profile/fingerprint.
- 3) **Virus Transfer:** Given the viral size considerations, sterile filtration methods employed in FVT do not eliminate the risk of transferring eukaryotic viruses that may infect the recipient. Although it is possible to screen donor feces for known pathogenic viruses, the human gastrointestinal tract is known to harbor hundreds of eukaryotic viruses with unknown functions [54–56]. Many of these viruses are likely harmless to the human host, but it cannot be ruled out that they might play a role in later disease development, as seen in the case of human papillomavirus (HPV), which is a risk factor for cervical cancer years after infection [57]. Here, the dilution feature of a chemostat can help ensure that very few or no eukaryotic viruses are transferred to the recipient.

However, there are certain negative and limiting factors that needs to be addressed for FVT-ChP as a treatment:

- 1) **Recipient compatibility:** A limiting factor of FVT-ChP used in this study, was that it only improved the OGTT measure and not the other main endpoint of body weight gain. Therefore, it is important to screen for more suitable recipient compatibility of both FVT-ChP from single donors and mixtures.
- 2) **Selectivity:** As a double edge sword, the chemostat fermentation allows for increase control of the phage propagation and increases reproducibility, but the applied conditions such as media composition, temperature and pH will act highly selective on which bacteria and phages that can persist and propagate in these conditions.
- 3) **Inflammatory Response:** Although the cytokine measures were challenged by the lack of difference between the lean and obese control, it remains relevant to further investigate the mechanisms and importance of the increased cytokine levels observed in blood serum for the FVT-ChP treated mice. It should be clarified whether it was a prolonged response with potential adverse effects for the recipient or if it is a more transient reaction on the presence of foreign phages/viruses.

Considering that the same chemostat-propagated phageome (FVT-ChP) in this early and preliminary phase has demonstrated phenotypical effects in two very distinct disease etiologies of diet-induced obesity and *C. difficile* infection [6], it emphasizes the need for refinements to enhance treatment efficacy.

We have added the following text at line 431-438 to include these perspectives.

“The concept of chemostat-propagated phageomes (FVT-ChP) could address the main limitations that are associated with unmodified FVT: procuring and reproducing sufficient enteric phage solutions, in vitro donor-recipient compatibility screening of the individual GM, and minimizing the transfer of eukaryotic viruses from donors to recipients. Considering that the concept of FVT-ChP as a modification of FVT already in this premature stage has demonstrated phenotypical effects in treating two distinct disease etiologies of diet-induced obesity and C. difficile infection[6], it urges for refinements as a therapeutic tool targeting other diseases associated with GM dysbiosis.”

5. Why did gut bacterial Shannon diversity of the controls decrease after the 2nd FVT (Fig. 4A), despite only being given SM buffer?

Thank you for this question. The change in the Shannon diversity index observed in all groups is likely driven by the dietary change (from a low-fat diet to a high-fat diet) and the altered environmental conditions in our housing facility compared to those in the vendor's facility where the mice originated. Therefore, this observation appears to be independent of the SM buffer. When we examine the lean control group (on a low-fat diet), we can see that the Shannon diversity index approaches normalization at the end of the study compared to arrival, which supports the explanation mentioned above. As an additional control measurement, we also sequenced the 16S rRNA gene profile just before the first FVT treatment. Unfortunately, we did not sequence the metavirome at this timepoint due to financial constraints. It is noteworthy that at this timepoint, the mice exhibited similar bacterial diversity as after the second FVT treatment, suggesting that the observed decrease in alpha-diversity is unlikely to be related to either the SM buffer or the FVTs.

We have added one more timepoint of bacteriome analysis to Figure 4 and the following text to address this at line 178-182 and 340-342:

“Upon arrival to our housing facility, the bacterial composition of the mice appeared comparable based on bacterial diversity (Shannon diversity index) and bacterial composition (Bray-Curtis dissimilarity) (Fig. 3A & 3B). The change in diet and environmental conditions in our housing facilities, compared with that of the vendor, clearly affected the bacterial diversity (Fig. 3A) of all groups, including the lean control.”

“It is commonly accepted that high-fat diet[58] and housing conditions[59–61] substantially affects the GM of mice, which would explain the sudden overall decrease in the bacterial diversity in all groups both before and after the study intervention.”

6. The authors should state how were the mice housed. Individually? Co-housed within each treatment condition?

Thank you for making us aware of this missing information. This should obviously be clearly stated. Each group represented 8 mice divided in two cages (4 mice per cage). In terms of animal welfare, it cannot be argued to house the animals individually in the used animal model. However, a new paper from Aaron Ericssons label argue that decreasing housing density improves statistical power [33], which is something we surely will consider in future studies. In addition, we have made sure that the manuscript follows the ARRIVE E10 guidelines.

Minor points:

i. In the Discussion, they stated that they "address the safety considerations associated with the risk of infections by eukaryotic viruses". It does not appear that safety considerations formed a major part of their results.

Thank you for your comment. You are correct in pointing out that our study does not provide a comprehensive evaluation of the overall safety. Instead, we aim to assess that inactivation or removal of eukaryotic viruses would enhance safety by reducing the risk of eukaryotic virus infections, thereby improving safety while still maintaining the previously observed treatment efficacy. We have throughout the manuscript made it clearer that improvement of safety in regard to removing potential infectious eukaryotes viruses is a major part of developing modified FVTs.

Reviewer 5

Reviewer #5 (Remarks to the Author):

The manuscript by Mao et al describes the results of experiments wherein mice exposed to a high-fat diet (or control low-fat diet) were treated with various preparations from mouse feces containing different fractions of the fecal virome. This is very novel work, but there are several concerns regarding the data as presented.

Thanks to the reviewer for the useful comments and recognition of the novelty of our research. We acknowledge the highlighted limitations of our study, which we to our best have addressed accordingly.

Major concerns
One major concern is the lack of scientific premise or rationale. It is not clear why efficacy in a model of *C. difficile* infection would lead to the hypothesis that fecal virome transfer (FVT) would be beneficial in diet-induced obesity.

We acknowledge your feedback and agree that in retrospect, the introduction did not effectively communicate the scientific rationale and hypothesis, a concern also raised by other reviewers. Our hypothesis is by no means meant to suggest that effects observed in a *C. difficile* model can be directly translated to a diet-induced obesity model. Instead, we pose these two main questions:

1. Can modified FVT treatments (to improve safety) alleviate symptoms in two very different animal disease etiologies through phage-mediated restoration of a dysbiotic GM? This is irrespective of whether the dysbiosis is driven by one pathogenic bacterium (like *C. difficile*) or an overall GM dysbiosis/imbalance (like metabolic syndrome). Addressing this question would provide insights into a broader spectrum of FVT applications.
2. If modified FVTs can maintain treatment efficacy while improving safety (in term of eukaryotic viral transfer) and/or donor variability.

These scientific rationales have now been clarified, and you can find the revisions at Line 101 -107:

*“We have also demonstrated that the modified FVT-SDT and FVT-ChP showed promising results in treating *C. difficile* infections in a mouse model[6], which represents a simple disease etiology mainly caused by the toxin-producing *C. difficile*[62,63]. The more complex GM associated diet-induced obesity model[64] was included in this study to investigate whether the same modified FVTs [6] could be used to improve phenotypes from these two very different disease etiologies through phage-mediated restoration of a dysbiotic GM. Only male mice were included, since female mice are highly protected against diet-induced obesity[1].*

Our main objective was therefore to screen whether or not the differently modified FVTs[5,6] had the potential to alleviate symptoms on co-morbidities of metabolic syndrome as a safer and/or more reproducible alternative to the unmodified FVT that we previously have used in a similar diet-induced obesity model[4].”

Specifically, it is not clear from the introduction which viral preparation method is hypothesized to provide benefits, or why any certain sample treatment is going to preferentially induce changes in the microbiome.

Both our research and that of other investigators [2,4,6,19,35,36,65] have demonstrated how FVT-like approaches can be utilized to alter the phenotype of the recipient, likely through phage-driven GM modulations. Our objective is not to speculate which treatment that will provide benefits but, rather, to screen for the most promising modification methods of FVT that can both maintain treatment efficacy and address some of the key challenges associated with unmodified FVT, such as donor variability and the potential transfer of eukaryotic viruses.

This aspect has been addressed in the previous comments as well, and we have expanded the references to include other studies demonstrating GM modulations using FVT to emphasize our belief that GM restoration potentially can be used to target various diseases. You can find these revisions at Line 76-85.

“Independent studies have successfully treated patients suffering from rCDI with sterile filtrated donor feces (containing mainly viruses and limited number of viable bacteria), which were shown to be as effective as FMT[18,66,67]. The successful treatments have been hypothesized to be driven by phage-based restoration of the GM[18,23]. This approach is often referred to as fecal virome transplantation (FVT). In preclinical settings, FVT has also been shown to alleviate symptoms of type-2 diabetes and obesity in male mice[2,4], prevent the onset of necrotizing enterocolitis in preterm piglets[36], restore the GM after antibiotic intervention[35], and improve the proliferation of commensal gut Akkermansia muciniphila[19], all likely through GM modulations. These findings highlight the promising application of FVT as a GM-restoring treatment targeting various diseases associated with GM dysbiosis.”

A major methodological concern is the lack of correction for multiple tests, to control the false discovery rate. This should be applied to comparisons of the virome and bacterial microbiome.

Thanks for this important comment. We acknowledge that the consideration of false discovery rate of microbiome analysis was missing. This is now implemented throughout data analysis using Wilcoxon FDR corrections. To improve the clarity, we detailed describes the statistical methods that we use at line 682-686. Our multiple corrections are based on asking the following two questions:

1. Does the high-fat diet induce obesity-associated symptoms compared to a low-fat diet? Essentially, this question aims to determine whether the model is effective or not. As a result, when comparing Obese and Lean Control, we do not apply corrections to the remaining groups.
2. Does any of the FVTs (both modified and unmodified) alter the phenotype of the mice compared with the obese control? This results in four comparisons that should be corrected for, as our research questions do not involve comparing the different FVTs with each other or with the lean control.

Also the methods have been extended accordingly in lines 651 - 672

“Initially, the dataset was purged for zOTU’s/viral contigs, which were detected in less than 5% of the samples, but the resulting dataset still maintained 99.8% of the total reads. R version 4.3.0 was used for subsequent analysis and presentation of data. A minimum threshold of sequencing reads for the bacteriome and virome analysis was set to 2,000 reads and 15,000 reads, respectively. The main packages used were phyloseq[68], vegan[69], DESeq2[70], ampvis2[71](not yet peer-reviewed), ggpubr, psych, igraph, ggraph, pheatmap, ComplexHeatmap, and ggplot2. The contamination of viral contig was removed by read count detected in negative controls through R package microDecon[72] (runs = 1, regressions = 1), and 41.5% of entries were removed. Cumulative sum scaling (CSS) normalization was performed using the R software using the metagenomeSeq package. α -diversity analysis was based on raw read counts and statistics were based on ANOVA. β -diversity was represented by Bray-Curtis dissimilarity and statistics were

based on pairwise PERMANOVA corrected with FDR (false discovery rate). DESeq2 was used to identify differential microorganisms on the summarized bacterial species level and viral contigs (vOTUs) level. The correlation network of bacterial association with phenotypic and immunologic variables, and the correlation heatmap between bacterial zOTUs and viral contigs (vOTUs) were calculated using pairwise Spearman's correlations and corrected with FDR. The two-side Wilcoxon rank-sum tests were adopted for analysis of the phenotypic variables, cytokine levels, immune cell levels, Shannon diversity index (α -diversity), PERMANOVA of β -diversity, and abundance of single bacterial genus, the comparison was conducted between FVT treatments and obese control mice, with FDR correction adopted, the comparison between obese and lean control mice was also conducted."

The sample size is very small and it is unclear whether mice were housed individually or group-housed. If mice were group-housed, cage effects must be considered, and the sample size (per treatment group) becomes questionable. The variability in many of the outcome measures suggests the need for additional samples.

Thank you for bringing the missing information in our animal study to our attention. We have taken the necessary steps to ensure that the manuscript aligns with the ARRIVE E10 checklist [73]. In our study, each group was represented by 8 mice, which were divided into two cages.

We agree that the sample sizes do present a limitation to our results, and we fully acknowledge the consideration of the coprophagic-driven cage effect [33]. Our decision regarding the sample size ($n = 8$) was primarily based on a prior publication where we validated the appropriate sample size for measuring prediabetic symptoms using HbA1c levels or the oral glucose tolerance test (OGTT) [74], and this was further supported by a subsequent study [4]. In line with animal welfare principles and to adhere to the "reduction" aspect of the 3Rs [75], we could not justify including additional animals if the $n = 8$ was deemed sufficient to detect an effect [74]. However, we have recently come across a study from Aaron Ericsson's lab arguing that reducing housing density improves statistical power [33]. This is certainly something we plan to consider in future studies, as well as increasing the number of animals to account for the variance observed in many of the outcome measures, as the reviewer correctly pointed out. We have added the following text to address this important limitation in the manuscript, see Line 411-418.

"The chosen sample size ($n = 8$) group housed in two cages, is previously validated as sufficient for diet-induced obesity[74], thereby accommodating the 3R principle of reduction[75]. The combination of group housing and the coprophagic behaviour of the mice may cause in cage-associated effects[33], which constitutes a limitation of the study. However, statistical analysis of the phenotypical measures (Table S4) showed no pronounced cage-associated effects concerning the main findings of our study. A recent report suggests decreasing animal density in the cages to increase the statistical power[33], which could address the cage-associated variance issue while maintaining accommodation of the 3Rs[75] in future studies."

Figures and figure panels are cited in haphazard order (Figure 1C and D cited before 1A and B), or not specifically at all (e.g., only three panels of Figure 2 specifically cited; Figure 3 cited as single figure with no mention of individual panels. If data merits inclusion in a primary figure, it should be described and cited (in alphanumeric order).

Thanks for noting this mistake. We have made sure that the figures are cited in numeric and alphabetic order.

It is difficult to interpret the findings shown in Figure 2 without absolute numbers. Percentages of each immune cell subset must be accompanied by absolute cell numbers isolated/counted in each group.

We acknowledge that it would increase the transparency to include absolute cell numbers for the FACS derived immune cell count in Fig. 2, however, based on our experience we don't believe it's possible to achieve trustworthy absolute values:

1. Since we also need adipose tissue for other analyses, not all adipose tissue is utilized for flow cytometry. Therefore, the total cell count is not always meaningful. Furthermore, there are numerous processing steps, especially including leukocyte isolation by gradient centrifugation, where there is considerable variation in the number of cells obtained from sample to sample and the number lost along the way. Consequently, the cell count might be unreliable.
2. We also lack a reliable method to count cells in adipose tissue since there are always numerous remaining fat cells, making it difficult to count only the leukocytes in the cell counter.

To address this important comment, we have instead used the wording of "proportions" instead of "number" of cells throughout the manuscript.

Related to this, the authors describe a 'signal of alleviated adipose inflammation' (line 121) in mice that received the unmodified FVT, but considering the lack of difference (from obese control) in macrophages (Fig 2B), B cells (Fig 2D), T helper cells (Fig 2E), CTLs (Fig 2F), Eff memory CD8+ and CD4+ T cells (Fig 2H and 2L), and central memory CD4+ T cells, there would need to be substantial reduction in the absolute number (and not just percentage) of the other cell types listed in the manuscript (DCs, M1 macrophages, and central memory CD8+ T cells) to support that interpretation.

We agree with the reviewer, that this statement should be toned down. We have therefore removed that sentence as an overall observation of alleviated adipose inflammation, but instead maintain the statement that the lower proportions of the mentioned immune cells support the alleviated symptoms of MAFDL in the same unmodified FVT group.

In addition, we have added to the discussion at line 334-339:

"Since we do not have the total cell count from adipose tissue and there were no significant differences in cytokine levels (Fig. S3 & S9), it is challenging to draw a general conclusion about the extent to which there was less inflammation in the unmodified FVT group. However, the phenotypical differences present in the liver tissue, adipose tissue, and blood cytokine profile all point in the same direction."

The lack of difference between obese and lean controls in nine of the ten analytes shown in Figure 3 makes it difficult to interpret the stated differences.

This is a valid point, and we appreciate your input. To clarify, it is important to note that 9 out of the 10 cytokines showed no difference between the lean and obese control groups. While we acknowledge that this limitation impacts the interpretation of the cytokine profile, the results are as they are. In response to this, we have highlighted this limitation in the manuscript to ensure that readers are aware of it. Additionally, we have removed the word "undesirable" as we do not have enough information to determine whether it is undesirable or not. You can find these revisions in the manuscript at Line 307-309:

"However, the interpretation was challenged by inconsistent differences in the cytokine levels between the lean and obese control groups (Fig. S3)."

Changes in beta-diversity are very minor. PERMANOVA p values should be accompanied by F values. Based on the minimal separation of treatment groups on PCoA at the later timepoints, it is highly speculative to make associations between the microbiome and the metabolic outcomes. If the metabolic outcomes were repeatable in larger cohorts of mice, a mechanistic role for FVT-induced changes in the bacterial

microbiome would require parallel studies using germ-free or antibiotic-treated mice to demonstrate a requirement of the gut microbiome for the beneficial effects of FVT.

We acknowledge your observation that the differences in beta-diversity are not clearly visible in Figure 3 and 4. Therefore, we have recalculated the F-values and corrected p-values and have integrated these into the figures. Furthermore, we have visually highlighted the significant group differences in red color to enhance the presentation of the results.

Thank you for your valuable suggestions for future studies. We recognize the necessity for additional and larger studies aimed at investigating the mechanisms underlying FVT and providing solid evidence of the link between the microbiome and metabolic outcomes. As such, we are not claiming to have established definitive evidence for this link. However, we are suggesting possible associations, drawing from other studies that have indicated links between the GM (and its metabolites) and symptoms of metabolic syndrome [76,77]. In addition, we have previously demonstrated clear metabolic changes in conjunction with alterations in the GM and gut metabolome following FVT treatment [4].

We have also provided a more detailed hypothesis outlining how changes in the GM and metabolome may be influenced by the phages in the FVT, subsequently affecting the host. This has been integrated into the manuscript to enhance the clarity of our hypotheses.

See line 377-401

“The mechanisms behind the GM modulating effects of FVT are still poorly understood, but accumulating reports suggest that the phenotypic traits of FVT donors, to some extent, can be transferred to recipients, as the phages might catalyze a modulation of the recipient ecosystem to be similar to their origin. This is exemplified by phage donor profiles being transferred to the gut of C. difficile patients after successful FMT/FVT[16–18], FVT from lean mice could shift the GM composition in obese mice to resemble that of lean individuals[4], and FVT donor material originating from an ecosystem with a relatively high abundance of A. muciniphila, could significantly increase the abundance of the enteric endogenous (native) A. muciniphila in mice that received the FVT[19]. This may be driven by cascading events[23], as demonstrated in a gnotobiotic mouse model[24], where phage infections indirectly influence the bacterial balance. Thus, the far more complex viral composition of FVT could similarly alter the bacterial ecosystem and as observed in our previous study, affect the blood metabolome and GM, leading to systemic effects[4]. This concept might seem counterintuitive given the general belief in the strain-specific nature of phages, however, a recent study proposed that phages could interact with distantly related microbial hosts[20]. Phage satellites have, amongst other, been suggested to contribute to broader host ranges[21,22]. Also the transfer of potentially beneficial metabolic genes from temperate phages to their bacterial hosts[25–28], may enhance host competitiveness and contribute to overall GM changes. These findings align with recent research demonstrating how metabolic functions and bacterial interaction networks were context-dependent of variables like nutrition, host environment, and bacterial compositions[29], supporting the hypothesis that cascading events initiated by FVT could catalyze GM modulating effects. In addition to the bacteria-phage relations, the impact of the immune system on gut health should not be neglected. Recent evidence suggests that bacteriophages interact with our immune system through mechanisms like TLR3 and TLR9[30,31]. A recent review has summarized the current understanding of phage immunogenicity highlighting parallels with eukaryotic viruses[32]. Stimulation of the immune system may thus be another mechanism behind the efficacy of FVT.”

Minor

concerns

There is an apparent disconnect in the Introduction with focused discussion on methods of reducing

exposure to potential pathogenic eukaryotic viruses, and the real focus of the study which is the ability of FVT to alter metabolic outcomes or the composition of the gut microbiome. If the ability of the various treatments to eradicate potential pathogens is tangential to the study, perhaps refocus the Introduction to provide mechanistic rationale for why FVT prepared in any specific manner would selectively alter the gut microbiome or host metabolic phenotypes.

Thanks, we have made the rationale of the study clearer, see the reply for your initial major concern.

Ref 25 is not peer-reviewed. Pre-prints should not be included as peer-reviewed references.'

We appreciate your valuable input regarding the citation of preprints. We agree that preprints should not be cited as peer-reviewed references. In response to this, we have included a parenthetical note after each preprint citation, specifically in its first mention, to make it clear that these sources have not yet undergone full peer review. While it will ultimately be at the discretion of the editor to decide whether preprints are allowed, we agree with the importance of maintaining transparency in our references.

The legend for Figure 5 does not describe panel D.

We are not completely sure what is not described, but we have tried to improve the description and made it clearer in Figure 4 (former Fig. 5).

References

- 1 Pettersson US, Waldén TB, Carlsson P-O, *et al.* Female mice are protected against high-fat diet induced metabolic syndrome and increase the regulatory T cell population in adipose tissue. *PLoS One*. 2012;7:e46057.
- 2 Borin JM, Liu R, Wang Y, *et al.* Fecal virome transplantation is sufficient to alter fecal microbiota and drive lean and obese body phenotypes in mice. *Gut Microbes*. 2023;15. doi: 10.1080/19490976.2023.2236750
- 3 Wortelboer K, de Jonge PA, Scheithauer TPM, *et al.* Phage-microbe dynamics after sterile faecal filtrate transplantation in individuals with metabolic syndrome: a double-blind, randomised, placebo-controlled clinical trial assessing efficacy and safety. *Nat Commun*. 2023;14:5600.
- 4 Rasmussen TS, Mentzel CMJ, Kot W, *et al.* Faecal virome transplantation decreases symptoms of type 2 diabetes and obesity in a murine model. *Gut*. 2020;69:2122–30.
- 5 Adamberg S, Rasmussen TS, Larsen SB, *et al.* Reproducible chemostat cultures to eliminate eukaryotic viruses from fecal transplant material. *bioRxiv*. Published Online First: 2023. doi: 10.1101/2023.03.15.529189
- 6 Rasmussen TS, Forster S, Larsen SB, *et al.* Overcoming donor variability and risks associated with fecal microbiota transplants through bacteriophage-mediated treatments. *bioRxiv*. Published Online First: 2023. doi: 10.1101/2023.03.17.532897
- 7 Rey FA, Lok S-M. Common features of enveloped viruses and implications for immunogen design for next-generation vaccines. *Cell*. 2018;172:1319–34.
- 8 Koonin E V., Dolja V V., Krupovic M. Origins and evolution of viruses of eukaryotes: The ultimate modularity. *Virology*. 2015;479–480:2–25. <https://doi.org/10.1016/j.virol.2015.02.039>
- 9 Sausset R, Petit MA, Gaboriau-Routhiau V, *et al.* New insights into intestinal phages. *Mucosal Immunol*. 2020;13:205–15.
- 10 WHO Health Product Policy and Standards Team. Guidelines on viral inactivation and removal procedures intended to assure the viral safety of human blood plasma products. 2004;WHO-TRS924-Annex4-1-82. <https://www.who.int/publications/m/item/WHO-TRS924-Annex4> (accessed 2 January 2023)
- 11 Horowitz B, Bonomo R, Prince AM, *et al.* Solvent/detergent-treated plasma: a virus-inactivated substitute for fresh frozen plasma. *Blood*. 1992;79:826–31.
- 12 Kapuscinski J, Darzynkiewicz Z. Interactions of pyronin Y(G) with nucleic acids. *Cytometry*. 1987;8:129–37.
- 13 Darzynkiewicz Z, Kapuscinski J, Carter SP, *et al.* Cytostatic and cytotoxic properties of pyronin Y: relation to mitochondrial localization of the dye and its interaction with RNA. *Cancer Res*. 1986;46:5760–6.

- 14 Koonin E V., Dolja V V., Krupovic M. Origins and evolution of viruses of eukaryotes: The ultimate modularity. *Virology*. 2015;479–480:2–25.
- 15 Prince AM, Horowitz B, Brotman B. Sterilisation of hepatitis and HTLV-III viruses by exposure to tri(n-butyl)phosphate and sodium cholate. *Lancet*. 1986;1:706–10.
- 16 Zuo T, Wong SH, Lam K, *et al*. Bacteriophage transfer during faecal microbiota transplantation in *Clostridium difficile* infection is associated with treatment outcome. *Gut*. 2017;67:gutjnl-2017-313952.
- 17 Fujimoto K, Kimura Y, Allegretti JR, *et al*. Functional restoration of bacteriomes and viromes by fecal microbiota transplantation. *Gastroenterology*. 2021;160:2089-2102.e12.
- 18 Ott SJ, Waetzig GH, Rehman A, *et al*. Efficacy of sterile fecal filtrate transfer for treating patients with *Clostridium difficile* infection. *Gastroenterology*. 2017;152:799-811.e7.
- 19 Rasmussen TS, Mentzel CMJ, Danielsen MR, *et al*. Fecal virome transfer improves proliferation of commensal gut *Akkermansia muciniphila* and unexpectedly enhances the fertility rate in laboratory mice. *Gut Microbes*. 2023;15. doi: 10.1080/19490976.2023.2208504
- 20 Hwang Y, Roux S, Coclet C, *et al*. Viruses interact with hosts that span distantly related microbial domains in dense hydrothermal mats. *Nat Microbiol*. 2023;8:946–57.
- 21 Barcia-Cruz R, Goudenège D, Moura de Sousa JA, *et al*. Phage inducible chromosomal minimalist island (PICMI), a family of satellites of marine virulent phages. *bioRxiv*. Published Online First: 2023. doi: 10.1101/2023.07.18.549517
- 22 Eppley JM, Biller SJ, Luo E, *et al*. Marine viral particles reveal an expansive repertoire of phage-parasitizing mobile elements. *Proceedings of the National Academy of Sciences*. 2022;119. doi: 10.1073/pnas.2212722119
- 23 Rasmussen TS, Koefoed AK, Jakobsen RR, *et al*. Bacteriophage-mediated manipulation of the gut microbiome - promises and presents limitations. *FEMS Microbiol Rev*. 2020;44:507–21.
- 24 Hsu BB, Gibson TE, Yeliseyev V, *et al*. Dynamic modulation of the gut microbiota and metabolome by bacteriophages in a mouse model. *Cell Host Microbe*. 2019;25:803-814.e5.
- 25 Huang X, Jiao N, Zhang R. The genomic content and context of auxiliary metabolic genes in roseophages. *Environ Microbiol*. 2021;23:3743–57.
- 26 Heyerhoff B, Engelen B, Bunse C. Auxiliary metabolic gene functions in pelagic and benthic viruses of the Baltic sea. *Front Microbiol*. 2022;13. doi: 10.3389/fmicb.2022.863620
- 27 Wittmers F, Needham DM, Hehenberger E, *et al*. Genomes from uncultivated pelagiphages reveal multiple phylogenetic clades exhibiting extensive auxiliary metabolic genes and cross-family multigene transfers. *mSystems*. 2022;7. doi: 10.1128/msystems.01522-21
- 28 Moura de Sousa JA, Pfeifer E, Touchon M, *et al*. Causes and consequences of bacteriophage diversification via genetic exchanges across lifestyles and bacterial taxa. *Mol Biol Evol*. 2021;38:2497–512.

- 29 Weiss AS, Niedermeier LS, von Stempel A, *et al.* Nutritional and host environments determine community ecology and keystone species in a synthetic gut bacterial community. *Nat Commun.* 2023;14:4780.
- 30 Sweere JM, Van Belleghem JD, Ishak H, *et al.* Bacteriophage trigger antiviral immunity and prevent clearance of bacterial infection. *Science (1979).* 2019;363:eaat9691.
- 31 Gogokhia L, Buhrke K, Bell R, *et al.* Expansion of bacteriophages is linked to aggravated intestinal inflammation and colitis. *Cell Host Microbe.* 2019;25:285-299.e8.
- 32 Champagne-Jorgensen K, Luong T, Darby T, *et al.* Immunogenicity of bacteriophages. *Trends Microbiol.* 2023;31:1058–71.
- 33 Russell A, Copio JN, Shi Y, *et al.* Reduced housing density improves statistical power of murine gut microbiota studies. *Cell Rep.* 2022;39. doi: 10.1016/j.celrep.2022.110783
- 34 Kan L, Barr JJ. A mammalian cell's guide on how to process a bacteriophage. *Annu Rev Virol.* 2023;10:183–98.
- 35 Draper LA, Ryan FJ, Dalmasso M, *et al.* Autochthonous faecal viral transfer (FVT) impacts the murine microbiome after antibiotic perturbation. *BMC Biol.* 2020;18:173.
- 36 Brunse A, Deng L, Pan X, *et al.* Fecal filtrate transplantation protects against necrotizing enterocolitis. *ISME J.* 2022;16:686–94.
- 37 Guarner V, Rubio-Ruiz ME. Low-grade systemic inflammation connects aging, metabolic syndrome and cardiovascular disease. *Interdiscip Top Gerontol.* 2015;40:99–106.
- 38 Mansyur MA, Bakri S, Patellongi IJ, *et al.* The association between metabolic syndrome components, low-grade systemic inflammation and insulin resistance in non-diabetic Indonesian adolescent male. *Clin Nutr ESPEN.* 2020;35:69–74.
- 39 Quinn le BS, Anderson BG, Conner JD, *et al.* Overexpression of interleukin-15 in mice promotes resistance to diet-induced obesity, increased insulin sensitivity, and markers of oxidative skeletal muscle metabolism. *Int J Interferon Cytokine Mediat Res.* 2011;3:29–42.
- 40 Chassaing B, Koren O, Goodrich JK, *et al.* Dietary emulsifiers impact the mouse gut microbiota promoting colitis and metabolic syndrome. *Nature.* 2015;519:92–6.
- 41 Daniel N, Gewirtz AT, Chassaing B. *Akkermansia muciniphila* counteracts the deleterious effects of dietary emulsifiers on microbiota and host metabolism. *Gut.* 2023;72:906–17.
- 42 Henao-Mejia J, Elinav E, Jin C, *et al.* Inflammasome-mediated dysbiosis regulates progression of NAFLD and obesity. *Nature.* 2012;482:179–85.
- 43 Chen MS, Goodman DW. The structure of catalytically active gold on titania. *Science (1979).* 2004;306:252–5.
- 44 Cani PD, Depommier C, Derrien M, *et al.* *Akkermansia muciniphila*: paradigm for next-generation beneficial microorganisms. *Nat Rev Gastroenterol Hepatol.* 2022;19:625–37.

- 45 Li J, Yang G, Zhang Q, *et al.* Function of *Akkermansia muciniphila* in type 2 diabetes and related diseases. *Front Microbiol.* 2023;14. doi: 10.3389/fmicb.2023.1172400
- 46 Ma Q, Li Y, Wang J, *et al.* Investigation of gut microbiome changes in type 1 diabetic mellitus rats based on high-throughput sequencing. *Biomedicine and Pharmacotherapy.* 2020;124. doi: 10.1016/j.biopha.2020.109873
- 47 Cox LM, Yamanishi S, Sohn J, *et al.* Altering the intestinal microbiota during a critical developmental window has lasting metabolic consequences. *Cell.* 2014;158:705–21.
- 48 Scheithauer TPM, Rampanelli E, Nieuwdorp M, *et al.* Gut microbiota as a trigger for metabolic inflammation in obesity and type 2 diabetes. *Front Immunol.* 2020;11:571731.
- 49 Herrmann E, Young W, Rosendale D, *et al.* RNA-based stable isotope probing suggests *Allobaculum spp.* as particularly active glucose assimilators in a complex murine microbiota cultured in vitro. *Biomed Res Int.* 2017;2017:1829685.
- 50 Pujo J, Petitfils C, Le Faouder P, *et al.* Bacteria-derived long chain fatty acid exhibits anti-inflammatory properties in colitis. *Gut.* 2021;70:1088–97.
- 51 Jiao W, Zhang Z, Xu Y, *et al.* Butyric acid normalizes hyperglycemia caused by the tacrolimus-induced gut microbiota. *American Journal of Transplantation.* 2020;20:2413–24.
- 52 Zhou Z, Lu Y, Li J, *et al.* Astragalus compound oral solution synergistically enhances health-promoting effect of metformin in type 2 diabetes mouse model. *Res Sq.* Published Online First: 2023. doi: 10.21203/rs.3.rs-2505907/v1
- 53 Zhernakova A, Kurilshikov A, Bonder MJ, *et al.* Population-based metagenomics analysis reveals markers for gut microbiome composition and diversity. *Science.* 2016;352:565–9.
- 54 Cai Z, Wang S, Li J. Treatment of Inflammatory Bowel Disease: A Comprehensive Review. *Front Med (Lausanne).* 2021;8. <https://doi.org/10.3389/fmed.2021.765474>
- 55 Lim ES, Zhou Y, Zhao G, *et al.* Early life dynamics of the human gut virome and bacterial microbiome in infants. *Nat Med.* 2015;21:1228–34.
- 56 Shah SA, Deng L, Thorsen J, *et al.* Expanding known viral diversity in the healthy infant gut. *Nat Microbiol.* 2023;8:986–98.
- 57 Doorbar J, Egawa N, Griffin H, *et al.* Human papillomavirus molecular biology and disease association. *Rev Med Virol.* 2015;25:2–23.
- 58 Daniel H, Gholami AM, Berry D, *et al.* High-fat diet alters gut microbiota physiology in mice. *ISME Journal.* 2014;8:295–308.
- 59 Thurman CE, Klores MM, Wolfe AE, *et al.* Effect of housing condition and diet on the gut Microbiota of weanling immunocompromised mice. *Comp Med.* 2021;71:485–91.
- 60 Lundberg R, Bahl MI, Licht TR, *et al.* Microbiota composition of simultaneously colonized mice housed under either a gnotobiotic isolator or individually ventilated cage regime. *Sci Rep.* 2017;7:1–11.

- 61 Ericsson AC, Davis JW, Spollen W, *et al.* Effects of vendor and genetic background on the composition of the fecal microbiota of inbred mice. *PLoS One*. 2015;10:1–19.
- 62 Rao K, Erb-Downward JR, Walk ST, *et al.* The systemic inflammatory response to *Clostridium difficile* infection. *PLoS One*. 2014;9:e92578.
- 63 Chen X, Katchar K, Goldsmith JD, *et al.* A mouse model of *Clostridium difficile*-associated disease. *Gastroenterology*. 2008;135:1984–92.
- 64 Fraulob JC, Ogg-Diamantino R, Fernandes-Santos C, *et al.* A mouse model of metabolic syndrome: Insulin resistance, fatty liver and non-alcoholic fatty pancreas disease (NAFPD) in C57BL/6 mice fed a high fat diet. *J Clin Biochem Nutr*. 2010;46:212–23.
- 65 Feng H, Xiong J, Liang S, *et al.* Fecal virus transplantation has more moderate effect than fecal microbiota transplantation on changing gut microbial structure in broiler chickens. *Poult Sci*. 2023;103282.
- 66 Kao DH, Roach B, Walter J, *et al.* Effect of lyophilized sterile fecal filtrate vs lyophilized donor stool on recurrent *Clostridium difficile* infection (rCDI): Preliminary results from a randomized, double-blind pilot study. *J Can Assoc Gastroenterol*. 2019;2:101–2.
- 67 Wilcox MH, McGovern BH, Hecht GA. The efficacy and safety of fecal microbiota transplant for recurrent *Clostridium difficile* infection: current understanding and gap analysis. *Open Forum Infect Dis*. 2020;7. doi: 10.1093/ofid/ofaa114
- 68 McMurdie PJ, Holmes S. phyloseq: an R package for reproducible interactive analysis and graphics of microbiome census data. *PLoS One*. 2013;8:e61217.
- 69 Dixon P. VEGAN, a package of R functions for community ecology. *Journal of Vegetation Science*. 2003;14:927–30.
- 70 Love MI, Huber W, Anders S. Moderated estimation of fold change and dispersion for RNA-seq data with DESeq2. *Genome Biol*. 2014;15:550.
- 71 Andersen KS, Kirkegaard RH, Karst SM, *et al.* ampvis2: an R package to analyse and visualise 16S rRNA amplicon data. *bioRxiv*. Published Online First: April 2018. doi: 10.1101/299537
- 72 McKnight DT, Huerlimann R, Bower DS, *et al.* microDecon: A highly accurate read-subtraction tool for the post-sequencing removal of contamination in metabarcoding studies. *Environmental DNA*. 2019;1:14–25.
- 73 Percie du Sert N, Hurst V, Ahluwalia A, *et al.* The ARRIVE guidelines 2.0: Updated guidelines for reporting animal research. *Br J Pharmacol*. 2020;177:3617–24.
- 74 Bondarenko V, Løkke CR, Dobrowolski P, *et al.* Controlling the uncontrolled variation in the diet induced obese mouse by microbiomic characterization. *Sci Rep*. 2022;12. doi: 10.1038/s41598-022-17242-8
- 75 MacArthur Clark J. The 3Rs in research: a contemporary approach to replacement, reduction and refinement. *Br J Nutr*. 2018;120:S1–7.

- 76 Wu J, Wang K, Wang X, *et al.* The role of the gut microbiome and its metabolites in metabolic diseases. *Protein Cell.* 2021;12:360–73. <https://doi.org/10.1007/s13238-020-00814-7>
- 77 De Vos WM, Tilg H, Van Hul M, *et al.* Gut microbiome and health: mechanistic insights. *Gut.* 2022;71:1020–32.

REVIEWER COMMENTS

Reviewer #1 (Remarks to the Author):

In their revised submission, Torben Sølbeck Rasmussen and colleagues offered discussions and clarifications addressing our queries. Outstanding concerns remain however. To fortify the robustness of conclusions, it would be beneficial to incorporate additional experiment and discourse. My other major concerns are as follows:

1. The authors provided an explanation on their results in terms of obesity phenotype and immune profile, whilst all of these results which they themselves said were puzzling. I wonder whether this observation is stable and reliable to be discussed? They did not provide reasonable discussion or explanation

2. An essential point to consider is the therapeutic effect of modified FVT. It is crucial that the modified FVT group is not only compared with the obese group but also with the unmodified FMT lean group. However, such comparisons seem to be lacking throughout the manuscript (as seen in Figure 3, Figure 4, and Figure S5).

3.1 In Figure 3A, at the termination of treatment, the Shannon diversity index in the FMT-ChP treatment group decreased compared to the obese control, whilst higher in lean control group. Can you explain this result?

3.2 In Figure 3D, the FVT group (especially the FVT Chp group) does not appear to reshape the bacterial community towards a lean state. Instead, the dysbiosis of the gut microbiota seems to be more severe when compared with the lean control group. This observation leads to some confusion regarding the authors' conclusion in the abstract, which suggests that FVT mediated gut microbiota modulation influences the treatment efficacy of FVT.

Reviewer #2 (Remarks to the Author):

Reviewer #3 (Remarks to the Author):

Reviewer #4 (Remarks to the Author):

We thank the authors for their comments, which I believe have sufficiently addressed our previous concerns.

Studying their bacteriome and virome analyses (Figs. 3 and 4 and associated results) more closely, we have several additional concerns:

1. They claimed there were statistically significant differences in bacteriome (Fig. 3B) and virome (Fig. 4B) compositions between all FVT-treated conditions and obese control by computing beta diversity, presumably between pairs of mice in the different groups. They did not compare this to beta diversity between mice *within* groups at all. The effect they should have checked was a statistically significant difference between within-group and between-group beta diversities. Or else the effect picked up could simply have been due to inter-individual differences, which are large especially for viromes.

2. Their taxonomic composition analyses do not appear to be carefully done (Figs. 3C and 4C). It is unclear how they defined the taxonomic groups to which reads are assigned - some are species, some genera, some families, and some even orders. Many of them even overlap with each other (e.g. Clostridiaceae and Clostridiales in Fig. 3C, and Microviridae and Petitvirales in Fig. 4C - and there is even Unclassified_Viruses there). If the taxonomic groups are not well-defined, then this calls into question any differential abundance analyses done subsequently (Figs. 3D-F and 4E-F and conclusions thereof). For example, assigning reads at the family or order level may conceal a real signal at the species level. Splitting up reads belonging together into 2 groups may also create a spurious signal.
- b. The similar-colored boxes (Figs. 3C and 4C-D) are also hard to interpret. A stacked bar plot may allow easier visual interpretation.

Minor points:

- i. The greenish colors of the lines in Fig. 1A are very similar in print. Please make them more distinctive from each other.
- ii. The scale bars in the Fig. 1C histology images are covered up (apparent in the previous version).

Please find all responses in the rebuttal letter written in red. The text added to the manuscript is indicated in *italics* within the rebuttal letter, with corresponding line references to the tracked changes in the Word document. Also a clean version of the manuscript is uploaded.

Reviewer #1+#2 (Remarks to the Author):

We thanks the reviewers for the additional comments.

In their revised submission, Torben Sølbeck Rasmussen and colleagues offered discussions and clarifications addressing our queries. Outstanding concerns remain however. To fortify the robustness of conclusions, it would be beneficial to incorporate additional experiment and discourse. My other major concerns are as follows:

1. The authors provided an explanation on their results in terms of obesity phenotype and immune profile, whilst all of these results which they themselves said were puzzling. I wonder whether this observation is stable and reliable to be discussed? They did not provide reasonable discussion or explanation

Thank you for the comment. We realized that one of the line references unfortunately was incorrect in our last rebuttal letter, thus part of these responses was only included in the manuscript with tracked changes and not the rebuttal letter. This is now included below. We are sorry for the inconvenience.

As mentioned in our first response, we believe that from a scientific standpoint it's important to also include and discuss data that did not fit into our original hypotheses. We agree that the presentation and interpretation in the first submission had to be improved, but in the 1st revision, we indeed addressed this concern in several paragraphs in lines 309-343. Here we both discuss the potential links between the different cytokines and the observed phenotypes, as well as stating the issues of inconsistency in the differences of analytes levels between the two control groups of lean and obese mice. We also suggest that phage immunogenicity could have been involved in the increased cytokines levels, and in term of the FACS analysis we also highlight the limitations of the results. From our perspective we have included all necessary caveats of these observations and included reasonable discussions and have therefore not changed this part of the manuscript further. However, should the editor request so, we will obviously extend/adjust our discussion of these results further.

See line 309-343:

"The cytokine profile of the blood serum was examined since low-grade systemic inflammation is connected to metabolic syndrome[1,2]. Especially elevated levels of the cytokines MIP-2, IL-15, and TNF- α in the FVT-ChP treated mice indicated elevated inflammation. However, the interpretation was challenged by inconsistent differences in the cytokine levels between the lean and obese control groups (Fig. S3). The recruitment and activation of neutrophils via MIP-2 can prompt the release of diverse inflammatory mediators, which can hasten the onset of liver inflammation[3]. TNF- α is produced by adipose tissue and works as a pro-inflammatory cytokine that has been shown to play a role in the development of metabolic dysfunction-associated fatty liver disease (MAFLD)[4,5]. Overexpression of IL-15 in transgenic mice and IL-15 treatment of NOD mice have been reported to improve glucose tolerance[6]. As a double edge sword, the increased levels of pro-inflammatory cytokines (MIP-2, IL-15, and TNF- α) in the blood serum of FVT-ChP treated mice compared to the obese control may partly explain their improved blood glucose

regulation, as well as the lack of improvement in their associated weight gain measures and histopathology liver score compared to the obese control mice.

Phage immunogenicity has garnered increased interest due to its potential role in regulating our immune system by sharing similar structures and proteins with eukaryotic viruses[7]. Phages may be both passively and actively distributed throughout various parts of our body[8]. The interactions between phages transferred along with the chemostat-propagated virome and the host immune system could help explain our observations of increased levels of three out of ten cytokines in the FVT-ChP treated mice, compared with the obese control. However, we lack evidence of specific viruses that could have triggered this immune response in the blood serum. Although low-grade systemic inflammation is connected to metabolic syndrom[1,2], it does not exclude the possibility of improved glucose regulation alongside increased levels of some cytokines in blood serum.

The inflammatory state of obesity is associated with an accumulation of macrophages in the adipose tissue[9], where the activated macrophages are recruited by CD8⁺ T cells[10]. Additionally, elevated levels of dendritic cells can play a role in the immune response to liver injury[11]. In this study, reductions in the proportions of CD8⁺ T cells, activated macrophages, T helper cells, and dendritic cells were observed in the mice treated with unmodified FVT compared to the obese control, which may be linked to the observed reduction in the liver pathology. Since we do not have the total cell count from adipose tissue and there were no significant differences in cytokine levels (Fig. S3 & S9), it is challenging to draw a general conclusion about the extent to which there was less inflammation in the unmodified FVT group. However, the phenotypical differences present in the liver tissue, adipose tissue, and blood cytokine profile all point in the same direction.”

2. An essential point to consider is the therapeutic effect of modified FVT. It is crucial that the modified FVT group is not only compared with the obese group but also with the unmodified FVT lean group. However, such comparisons seem to be lacking throughout the manuscript (as seen in Figure 3, Figure 4, and Figure S5).

Thanks for the comment. Just to avoid any misunderstandings, the lean group did not receive any FVT or FVT like treatment, solely saline (sham). Despite this, we agree that it would benefit the presentation of our data to also include the statistics of the lean vs the different treatments, this is now included in all manuscript figures and supplemental figures. For box/bar/violin plots, the significant differences compared to the lean control are marked with a red star to differentiate it to significant differences between the obese control and the FVTs. For the PCoA plots, we have extended the table below with comparisons of the lean control group. For the differential abundance testing we have also included the comparisons with the lean control (Fig. S5 and Fig. S6). We initially did not include it to make the figures less busy. However, our initial hypothesis and main scientific questions remain as described in the first revision response:

1. Can modified and unmodified viral transplantation induce a discernible difference in mice phenotype and microbiome compared to obese control mice?
2. Does the high-fat diet-induced obesity mice model effectively work in comparison to lean control mice fed a low-fat diet?

Thus, corrections for multiple comparisons were performed accordingly.

We have added the below text to all relevant figure legends:

“The ★label on top of the boxplot of the treatment represents a $p < 0.05$ between the treatment and the lean control mice (two-side Wilcoxon rank-sum test).”

And added to the result section at line 182-184:

“As expected, the ad libitum low-fat diet provided to the lean control mice, significantly ($p < 0.05$) affected the bacterial diversity and composition compared to the ad libitum high-fat diet fed groups (Fig. 3A & 3B).”

3.1 In Figure 3A, at the termination of treatment, the Shannon diversity index in the FMT-ChP treatment group decreased compared to the obese control, whilst higher in lean control group. Can you explain this result?

Thanks a lot for this comment. We agree that it is important to address this observation in our discussion. Yes, FVT-ChP treated mice showed a significant decrease in the Shannon diversity index compared to the lean and obese control group. First, it is not surprising that the *ad libitum* high-fat diet the FVT-ChP group was provided, significantly impacted the alpha diversity compared with the *ad libitum* low-fat diet the lean group was provided. This is a well known phenomenon [12,13]. The significant decrease in the Shannon diversity index of the FVT-ChP group versus the obese group is indeed interesting. Although the chemostat fermentation setup [14] was set to mimic the conditions in the mouse gut and originating from the same intestinal content as the other treatments, it's unlikely that the process haven't affected the composition and diversity of phages when the fermentation reached a steady state. The composition of phages was indeed clearly differentiated in the chemostat propagated virome compared with the other FVTs (Fig. S10E), which may explain why the Shannon diversity index only were decreased in FVT-ChP group and not the other FVTs, compared to the obese control.

We have therefore added the below text to the discussion at line 414-420:

“The chemostat fermentation setup aimed to mimic mouse gut conditions [14] and utilized the same intestinal content as other FVT treatments. However, the chemostat propagation will inevitably alter phage composition and diversity compared to the original inoculum, as also illustrated by the differences in phage composition in the chemostat-propagated virome compared to other treatments (Fig. S10), also offering a possible explanation of the decreased Shannon diversity index observed in the FVT-ChP group, compared to the obese control.”

3.2 In Figure 3D, the FVT group (especially the FVT Chp group) does not appear to reshape the bacterial community towards a lean state. Instead, the dysbiosis of the gut microbiota seems to be more severe when compared with the lean control group. This observation leads to some confusion regarding the authors' conclusion in the abstract, which suggests that FVT mediated gut microbiota modulation influences the treatment efficacy of FVT.

Thanks for the comment. We agree that the FVT-ChP or other FVTs did not reshape the bacterial community into a “lean state” (something we do not claim either). What we claim was that FVT-ChP and unmodified FVT significantly shifted the bacterial composition (beta-diversity, Figure 3B) at termination compared with the obese control, and that this shift might have influenced the phenotypic outcomes observed. The most important bacterial taxa (e.g. lactobacilli & *Allobaculum*) driving this shift in the gut microbiome are now illustrated in Figure 3D. As earlier mentioned it’s not really surprising that the *ad libitum* high-fat diet, which all FVT groups were provided, greatly affected the gut microbiome compared with the *ad libitum* low-fat diet [12,13], thus, it makes more sense to compare the bacterial community of the FVT groups with the obese control that was provided the same diet, than to compare with the lean control. We therefore remain to believe that our statement in the abstract fits well to Figure 3D and we did not change the manuscript with regard to this comment.

Reviewer #3 + #4 (Remarks to the Author):

We thank the authors for their comments, which I believe have sufficiently addressed our previous concerns.

Thanks a lot for the positive feedback.

Studying their bacteriome and virome analyses (Figs. 3 and 4 and associated results) more closely, we have several additional concerns:

1. They claimed there were statistically significant differences in bacteriome (Fig. 3B) and virome (Fig. 4B) compositions between all FVT-treated conditions and obese control by computing beta diversity, presumably between pairs of mice in the different groups. They did not compare this to beta diversity between mice *within* groups at all. The effect they should have checked was a statistically significant difference between within-group and between-group beta diversities. Or else the effect picked up could simply have been due to inter-individual differences, which are large especially for viromes.

As the reviewer points out there is a cage factor. This is always a precondition in animal studies. Therefore, all animal studies are planned with at least two cages in each group to avoid bias from caging. In our study we had 2 cages per group. Under materials and methods we have now written, at line 542-543:

“Mice were housed with 2 cages per group to account for potential cage effect bias.”

Thus, we agree that this should be investigated and included in the manuscript. We had already included cage-effect analysis of all other parameters except for the gut microbiome in Table S4. We have therefore expanded this part of the supplementary table with a cage effect analysis of the alpha/beta-diversity of the bacteria and virome in Table S4. Looking at the viromes, there were no cage effects associated with the Shannon diversity index in any groups. Similarly, the viral beta-diversity of all FVT groups were also unaffected by cage effects, whereas the lean and obese controls appeared with cage-associated effects (Table S4).

With regard to the bacteria, only the Shannon diversity index of the lean control group displayed cage-associated effects, while all other groups did not. In contrast, the bacterial beta-diversity of most groups appeared with cage-associated effects, which likely is driven by the coprophagic behavior of mice [15] and

is a common issue in mice studies [16]. As already mentioned in the first revision, we will in future studies consider the recent results from Aaron Ericssons lab which argue that decreasing housing density improves statistical power and thereby addresses the cage-effect issue [15]. To address this concern and highlight the limitations we have included the below to the discussion at line 428-435:

“The viral diversity across all groups showed no cage effects for the Shannon diversity index. Similarly, the viral composition in all FVT groups remained unaffected by cage effects, while the lean and obese control groups displayed cage-associated effects. Regarding bacteria, only the Shannon diversity index of the lean control group showed cage-associated effects. However, bacterial composition in most groups exhibited cage-associated effects, likely influenced by coprophagic behavior, a common issue in mouse studies [15,16]. Thus, it cannot be ruled out that cage-associated differences in the bacterial composition may have influenced the treatment efficacy of the different FVTs.”

2. Their taxonomic composition analyses do not appear to be carefully done (Figs. 3C and 4C). It is unclear how they defined the taxonomic groups to which reads are assigned - some are species, some genera, some families, and some even orders. Many of them even overlap with each other (e.g. Clostridiaceae and Clostridiales in Fig. 3C, and Microviridae and Petitvirales in Fig. 4C - and there is even Unclassified_Viruses there). If the taxonomic groups are not well-defined, then this calls into question any differential abundance analyses done subsequently (Figs. 3D-F and 4E-F and conclusions thereof). For example, assigning reads at the family or order level may conceal a real signal at the species level. Splitting up reads belonging together into 2 groups may also create a spurious signal.

Thanks for this helpful comment. The taxonomical groups of the individual zOTUs/vOTUs were collapsed to simplify the figures and ease the interpretation of these results, while in Fig. 4E we had analyzed the viral differential abundance at vOTU level (high resolution version can be found in Fig. S5) and correlated the relative abundance of single zOTUs and vOTUs in Fig. S6 to include this higher taxonomical resolution of the analysis. It should be emphasized that each zOTU/vOTU were expected to constitute one bacterial/viral entity, and the taxonomical assignment was performed after the reads were assembled into individual zOTU/vOTUs, thus no reads were split up. We mixed the different taxonomic levels in the figures since we have collapsed the vOTUs/zOTUs to the lowest possible rank. The current databases do not allow us a better classification of the zOTUs or vOTUs, so we cannot avoid having some vOTUs only classified as *Petitvirales* whereas others can be classified as *Microviridae*. Same goes for the zOTUs. Similarly, we cannot avoid having many vOTUs classified as unknown viruses, since the databases are still insufficient to properly classify especially viruses. The method description can be found at line 643-671 and the script at <https://github.com/frejlarsen/vapline3>.

However, we acknowledge that current taxonomic groups must be specified to avoid any confusion. To address this concern, we have therefore specified that e.g. the collapsed zOTUs belonging to *Clostridiaceae* is belonging to the family level like e.g. *f_Clostridiaceae*, and thereby differentiated from e.g. *g_Bacteroides*. Similar corrections have also been done for the viral contigs (vOTUs), where e.g. merged viral contigs belonging to *Petitvirales* are named as Unclassified *Petitvirales*. This is improved for all main figures and supplementary figures where relevant, but the different levels of order, family, genus etc. remain. It is also emphasized in the figure legends that the zOTUs/vOTUs were collapsed to the best possible taxonomical resolution. If all single zOTUs should be included it would make the figures hard to interpret, since zOTU represent unique sequence variants where only sequence alignments with 100 %

similarity are merged into the same zOTU. Similar issue goes for the vOTUs. We have also added the following to the methods at line 654-659:

"The taxonomical categories of "Other/Remaining taxa", "Unclassified virus", and "Unknown" that are used in the different figures are different entities. "Other/Remaining taxa" encompasses all remaining low abundance taxa not depicted in the plot. "Unknown" refers to contigs that may be viruses but lack specific data records confirming their viral origin, and "Unclassified virus" represents viruses that have been identified as having viral origin but could not be further classified."

b. The similar-colored boxes (Figs. 3C and 4C-D) are also hard to interpret. A stacked bar plot may allow easier visual interpretation.

We included the heatmaps since we believe that they provide a better overview of differences in the relative abundance compared with bar plots. However, we have generated the stacked bar plots as suggested (see below). We prefer to keep the heatmaps, and will let it be up to the editor(s) to decide which type of plots to include.

Fig. 1 Bacteriome abundance barplot based on 16S rRNA gene amplicon sequencing at four time points: at arrival (before diet intervention (5 weeks old), before 1st FVT (11 weeks old), one week after 2nd FVT (14 weeks old), and study termination (23 weeks old).

Fig. 2 Virome abundance barplot based on whole-genome sequencing at three time points: at arrival (before diet intervention (5 weeks old), one week after 2nd FVT (14 weeks old), and study termination (23 weeks old).

Fig. 3 Predicted bacterial hosts (based on the viral contigs) abundance barplot based on whole-genome sequencing at three time points: at arrival (before diet intervention (5 weeks old), one week after 2nd FVT (14 weeks old), and study termination (23 weeks old).

Minor

points:

i. The greenish colors of the lines in Fig. 1A are very similar in print. Please make them more distinctive from each other.

Thank, this is a good point. We have improved the figure to make it easier to interpret by coloring the dots of the timepoints and add shape as a condition, see Fig. 1A.

ii. The scale bars in the Fig. 1C histology images are covered up (apparent in the previous version).

Thanks for finding this mistake, these scale bars should obviously be clear for the reader. The figures are now updated accordingly.

References

1. Guarner V, Rubio-Ruiz ME. Low-grade systemic inflammation connects aging, metabolic syndrome and cardiovascular disease. *Interdiscip Top Gerontol.* 2015;40:99–106.
2. Mansyur MA, Bakri S, Patellongi IJ, Rahman IA. The association between metabolic syndrome components, low-grade systemic inflammation and insulin resistance in non-diabetic Indonesian adolescent male. *Clin Nutr ESPEN.* 2020;35:69–74.
3. Qin CC, Liu YN, Hu Y, Yang Y, Chen Z. Macrophage inflammatory protein-2 as mediator of inflammation in acute liver injury. *World J Gastroenterol.* 2017;23:3043–52.
4. Kakino S, Ohki T, Nakayama H, Yuan X, Otabe S, Hashinaga T, et al. Pivotal role of TNF- α in the development and progression of nonalcoholic fatty liver disease in a murine model. *Hormone and Metabolic Research.* 2018;50:80–7.

5. Stanley TL, Zanni M V., Johnsen S, Rasheed S, Makimura H, Lee H, et al. TNF- α antagonism with etanercept decreases glucose and increases the proportion of high molecular weight adiponectin in obese subjects with features of the metabolic syndrome. *J Clin Endocrinol Metab* [Internet]. 2011 [cited 2023 Mar 9];96:E146–50. Available from: <https://academic.oup.com/jcem/article/96/1/E146/2833882>
6. Quinn le BS, Anderson BG, Conner JD, Pistilli EE, Wolden-Hanson T. Overexpression of interleukin-15 in mice promotes resistance to diet-induced obesity, increased insulin sensitivity, and markers of oxidative skeletal muscle metabolism. *Int J Interferon Cytokine Mediat Res*. 2011;3:29–42.
7. Champagne-Jorgensen K, Luong T, Darby T, Roach DR. Immunogenicity of bacteriophages. *Trends Microbiol* [Internet]. 2023;31:1058–71. Available from: <https://linkinghub.elsevier.com/retrieve/pii/S0966842X23001464>
8. Kan L, Barr JJ. A mammalian cell's guide on how to process a bacteriophage. *Annu Rev Virol* [Internet]. 2023 [cited 2023 Oct 5];10:183–98. Available from: <https://www.annualreviews.org/doi/10.1146/annurev-virology-111821-111322>
9. Weisberg SP, McCann D, Desai M, Rosenbaum M, Leibel RL, Ferrante AW. Obesity is associated with macrophage accumulation in adipose tissue. *J Clin Invest* [Internet]. 2003 [cited 2023 Mar 10];112:1796–808. Available from: <http://www.insightful>.
10. Nishimura S, Manabe I, Nagasaki M, Eto K, Yamashita H, Ohsugi M, et al. CD8⁺ effector T cells contribute to macrophage recruitment and adipose tissue inflammation in obesity. *Nat Med*. 2009;15:914–20.
11. Sutti S, Locatelli I, Bruzzi S, Jindal A, Vacchiano M, Bozzola C, et al. CX3CR1-expressing inflammatory dendritic cells contribute to the progression of steatohepatitis. *Clin Sci*. 2015;129:797–808.
12. Rasmussen TS, Mentzel CMJ, Kot W, Castro-Mejía JL, Zuffa S, Swann JR, et al. Faecal virome transplantation decreases symptoms of type 2 diabetes and obesity in a murine model. *Gut* [Internet]. 2020 [cited 2020 Mar 13];69:2122–30. Available from: <http://www.ncbi.nlm.nih.gov/pubmed/32165408>
13. Murphy EA, Velazquez KT, Herbert KM. Influence of High-Fat-Diet on Gut Microbiota: A Driving Force for Chronic Disease Risk. *Curr Opin Clin Nutr Metab Care*. 2015;18:515–20.
14. Adamberg S, Rasmussen TS, Larsen SB, Nielsen DS, Adamberg K. Reproducible chemostat cultures to eliminate eukaryotic viruses from fecal transplant material. *bioRxiv*. 2023;
15. Russell A, Copio JN, Shi Y, Kang S, Franklin CL, Ericsson AC. Reduced housing density improves statistical power of murine gut microbiota studies. *Cell Rep*. 2022;39.
16. Hildebrand F, Nguyen TLA, Brinkman B, Yunta RG, Cauwe B, Vandenabeele P, et al. Inflammation-associated enterotypes, host genotype, cage and inter-individual effects drive gut microbiota variation in common laboratory mice. *Genome Biol*. 2013;14.